# ImageNet-Hard: The Hardest Images Remaining from a Study of the Power of Zoom and Spatial Biases in Image Classification

**Mohammad Reza Taesiri**
University of Alberta
mtaesiri@gmail.com

**Giang Nguyen**
Auburn University
nguyengiangbkhn@gmail.com

**Sarra Habchi**
Ubisoft
sarra.habchi@ubisoft.com

**Cor-Paul Bezemer**
University of Alberta
bezemer@ualberta.ca

**Anh Nguyen**
Auburn University
anh.ng8@gmail.com

## Abstract

Image classifiers are information-discarding machines, by design. Yet, how these models discard information remains mysterious. We hypothesize that one way for image classifiers to reach high accuracy is to zoom to the most discriminative region in the image and then extract features from there to predict image labels, discarding the rest of the image. Studying six popular networks ranging from AlexNet to CLIP, we find that proper framing of the input image can lead to the correct classification of 98.91% of ImageNet images. Furthermore, we uncover positional biases in various datasets, especially a strong center bias in two popular datasets: ImageNet-A and ObjectNet. Finally, leveraging our insights into the potential of zooming, we propose a test-time augmentation (TTA) technique that improves classification accuracy by forcing models to explicitly perform zoom-in operations before making predictions. Our method is more interpretable, accurate, and faster than MEMO, a state-of-the-art (SOTA) TTA method. We introduce ImageNet-Hard, a new benchmark that challenges SOTA classifiers including large vision-language models even when optimal zooming is allowed.

## 1 Introduction

Since the release of AlexNet in 2012 [38], deep neural networks have set many ImageNet (IN) [59] accuracy records [38, 23]. While many papers reported improved learning algorithms or architectures, little is known about how the inner workings of image classifiers actually evolve. The success is often attributed to a network's ability to detect more objects [9] and a variety of facets of each object (*i.e.*, invariance to style, pose, and form changes) [50, 21]. By aggregating the information from all the visual cues in a scene, a classifier somehow chooses a better label for the image. For example, Figs. 13–14 in [58] show that a model detects both dogs and cats in the same image and only discards the dog features right before the classification layer to arrive at a `tiger cat` prediction.

When processing an image, a network may implicitly **zoom** in or out (defined in Sec. 3) to the most discriminative image region *ignoring* the rest of the image (Fig. 1a), and then extract that localized region's features to predict image labels. We hypothesize that the improved image classification may largely be due to the networks accurately zooming to the discriminative areas (*e.g.*, `junco` and `magpie` birds in Fig. 1a) rather than more accurately describing them (*i.e.* generating better features of these two birds). In this work, we present supporting evidence for our zooming hypothesis.

37th Conference on Neural Information Processing Systems (NeurIPS 2023) Track on Datasets and Benchmarks.

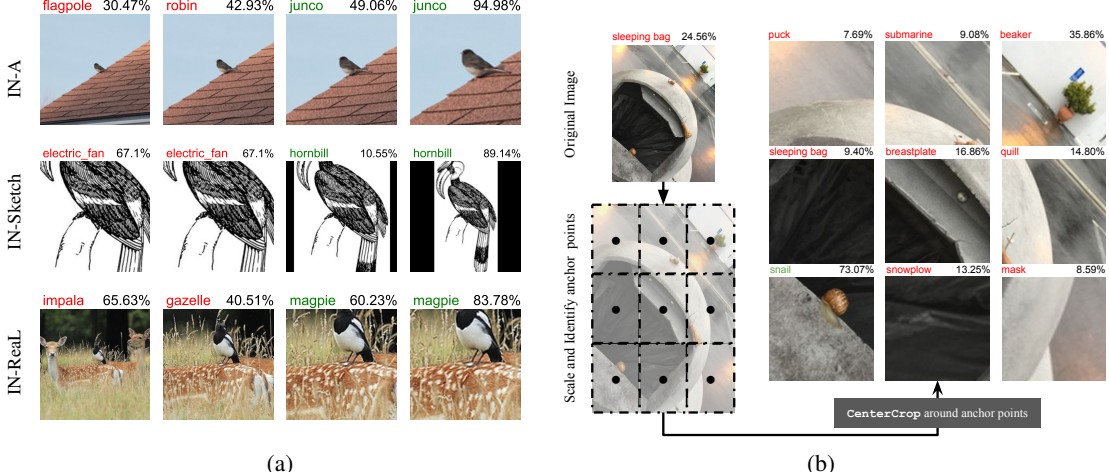

(a)                    (b)

Figure 1: **(a)** Each subfigure shows an input image, predicted label, and confidence score from an ImageNet classifier (top and middle: ResNet-50 [23]; bottom: ViT-B/32 [17]). With the standard center-crop transform, all 3 samples were misclassified (left-most column). Adjusting framing via zooming yields correct predictions. **(b)** The zooming process correctly classifies a `snail` ImageNet-A image. We uniformly adjust the input query image's smaller dimension to match the target scale $S$. We then partition the image into a $3 \times 3$ grid, generating 9 crops centered at grid-cell centers (*i.e.*, ● anchor points) and feed each crop to the original image classifier.

We conduct a thorough study to test the effects of zooming in and out on the classification accuracy of six network architectures on six ImageNet-scale benchmarks. Our main findings also include:

1. A major, surprising finding is that **state-of-the-art, IN-trained models can accurately predict up to $98.91\%$ of ImageNet samples when an optimally-zoomed image is provided**. The remaining few hundred IN images ($0.39\%$) that are never correctly labeled by any model (despite optimal zooming) include mostly *ill-posed* and *rare* images (Sec. 4.1).
2. ImageNet-A [27] and ObjectNet [86] both exhibit a substantial center bias. For example, by only upsampling and center-cropping each ImageNet-A image, ResNet-50's accuracy increases dramatically from $0.09\%$ to $14.58\%$ (Sec. 4.2).
3. Zooming can be leveraged as an inductive bias at test time to improve ImageNet classification accuracy. That is, integrating zoom transformations into MEMO [83], a leading test-time augmentation method, yields consistently higher accuracy than the baseline ResNet-50 models and also MEMO with default transformations on multiple datasets (Sec. 4.3).

Our findings show that the accuracy of image classifiers can be improved by finding an optimal zoom setting first and then classifying that crop alone (Fig. 1a). Motivated by this insight, we build **ImageNet-Hard**[1], a new 1000-way classification benchmark that challenges state-of-the-art (SOTA) classifiers despite the application of optimal zooming (Sec. 4.4). In other words, we collect images from seven existing ImageNet-scale benchmarks where OpenAI's CLIP ViT-L/14 [53] misclassifies even when allowed to try 324 zooming settings. Interestingly, SOTA classifiers that operate at 224×224 resolution perform poorly on ImageNet-Hard (below 19% accuracy). Analyzing misclassifications on ImageNet-Hard reveals a major remaining challenge in the era of SOTA classifiers of Transformers [17], EfficientNets [69, 36], and large vision-language models (Sec. 4.5).

## 2   Related Work

**Learning to Zoom in image classification**   Leveraging zoom-in or crops of an image has a long history of improving fine-grained image classification with approaches varying from combining multiple crops at different resolutions [18, 73], using multiple crops of the object (i.e., part-based classification) [16, 37, 84, 68, 84, 36] to warping the input image [55, 31, 32]. We note that a common

---

[1]Code and data are available on `https://taesiri.github.io/ZoomIsAllYouNeed`.

prior definition of "zoom" [55, 31, 32, 70] is to first divide an input image into a grid and then warp the image, distorting the aspect ratio of the objects in the image. In contrast, our zoom procedure utilizes only two functions: resize and crop, maintaining the original aspect ratio.

Furthermore, to our knowledge, we are the first to perform a zoom study on ImageNet-scale datasets (ImageNet [59], ImageNet-A [27], ObjectNet [8], etc) while prior zoom approaches [16, 37, 84, 68, 84, 36, 32] exclusively focus on non-ImageNet, fine-grained classification (e.g. classifying birds or dogs). Due to such differences in the image distribution of interests, prior works mostly study *zooming in* (which benefits fine-grained classification) while we study *both* zooming in and out.

**Test-time data augmentation (TTA)** is a versatile technique that could help estimate uncertainty [65, 7, 6] and improve classification accuracy [38, 67, 23, 52, 46, 62, 34, 14]. When test inputs are sampled from unseen, non-training distributions, augmenting the data often improve a model's generalization to new domains [74, 83]. A simple TTA method is 10-crop evaluation [38] in which 5 patches of $224 \times 224$ px along with their horizontal reflections (resulting in 10 patches) are extracted from the original image. An alternative way to leverage the marginal output distributions over augmented data is to use them as gradient signals to update the classifier's parameters [75, 83]. We employ this approach to update the model during test time, with patches being zoom-based augmentations.

**Biases in ImageNet and datasets** Resize-then-center-crop has been a pre-processing standard for ImageNet classification since AlexNet [38]. This pre-processing exploits the known center bias of ImageNet. While ImageNet has been shown to contain a variety of biases in image labels [10, 71], object poses [5], image quality [39], we are the first to examine the *positional* biases of the out-of-distribution (OOD) benchmarks for ImageNet classifiers and find a strong center bias in ImageNet-A and ObjectNet that could affect how the community interprets progress on these OOD benchmarks.

Perhaps the closest to us is a preprint by Li et al. [42] that shows that cropping to the main object can improve model accuracy on ImageNet-A [27]. Yet, unlike [42], we study six ImageNet-scale datasets, both zooming in and zooming out, and we propose a new dataset of ImageNet-Hard.

# 3 Method

**Zoom definition** To zoom in or out of the image, we only use *resize* and *crop* operations. Initially, we uniformly resize the test image so that the smaller dimension matches the target scale of $S$. Then, we define a $3 \times 3$ grid on the image to divide it into 9 patches. We perform a *CenterCrop* operation at the center (● in Fig. 1b) of each patch to extract a $224 \times 224$ px crop from each of the nine locations (see Python code in Appendix A.1). In the *CenterCrop* step, zero-padding is used when the content to be cropped is smaller than $224 \times 224$. Overall, at each target scale $S$, we generate 9 crops (Fig. 1b).

We test 36 different values of $S$ ranging from 10 to 1024 px, resulting in a total of $36 \times 9 = 324$ different zoomed versions for each image. Based on initial scale factor $S$, we define three groups: (1) *zoom-out* group contains all augmented crops where $S < 224$; (2) *zoom-in* group contains all augmented crops where $S > 224$; and (3) *zoom-224* group contains the 9 patches where $S = 224$.

**Benchmark datasets** We use the ImageNet (IN) [59] dataset with both the original and ImageNet-ReaL [10] (ReaL) labels. For each IN image, we use the union of the IN and ReaL labels (IN+ReaL) to complement each other to reduce noise in IN labels. We further examine the effects of zoom-based transformations on four popular OOD benchmarks: (a) natural adversarials (ImageNet-A [27]), (b) image renditions (ImageNet-R [26]), (c) black-and-white sketches (ImageNet-Sketch [75]), and (d) viewpoint-and-background-controlled samples (ObjectNet [8]). We refer to these as benchmarks as IN-A, IN-R, IN-S, and ON, respectively, in the rest of the paper.

**Classifiers** We study the effects of zoom-based transformations on six popular image classifiers in the last decade: AlexNet [38], VGG-16 [63], ResNet-18 & ResNet-50 [23], ViT-B/32 [17], and OpenAI's CLIP-ViT-L/14 [53]. The inclusion of the 11-year-old AlexNet provides a baseline for the power of deep features (when given the right region to look at). Predicted labels from CLIP-ViT-L/14 are acquired using its standard zero-shot classification setup (Appendix A.5).

# 4 Experimental Results

## 4.1 Zooming has the potential to substantially improve image classification accuracy

To understand the potential of zooming in improving image classification accuracy, first, we establish an **upper-bound accuracy** (*i.e.* when an "optimal" zoom is given). That is, we apply 36 scales × 9 anchors = 324 zoom transformations (Sec. 3) to each image to generate 324 zoomed versions of the input. We then feed all $N = 324$ versions to each network and label an image "correctly classified given the optimal zoom" if at least 1 of the 324 is correctly labeled. We call such maximum possible top-1 accuracy "upper-bound accuracy" in Tab. 1. Our experiment also informs the community of the type of image that *cannot* be correctly labeled even with an optimal zooming strategy.

Table 1: On in-distribution data (IN & ReaL) there exists a substantial improvement when models are provided with an optimal zoom, either selected from 36 (b) or 324 pre-defined zoom crops (c). In contrast, OOD benchmarks still pose a significant challenge to IN-trained models even with optimal zooming (i.e., all upper-bound accuracy scores $< 80\%$).

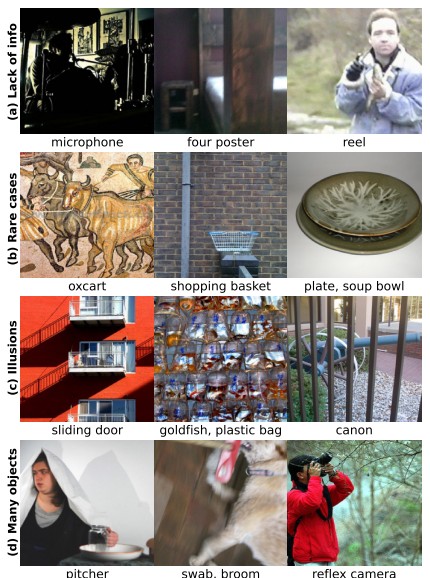

| | IN | ReaL | IN+ReaL | IN-A | IN-R | IN-S | ON |
|---|---|---|---|---|---|---|---|
| *(a) Standard top-1 accuracy based on N = 1 crop* | | | | | | | |
| AlexNet | 56.16 | 62.67 | 61.76 | 1.75 | 21.10 | 10.05 | 14.23 |
| VGG-16 | 71.37 | 78.90 | 78.52 | 2.69 | 26.98 | 16.78 | 28.32 |
| ResNet-18 | 69.45 | 76.94 | 76.47 | 1.37 | 32.14 | 19.41 | 27.59 |
| ResNet-50 | 75.75 | 82.63 | 82.97 | 0.21 | 35.39 | 22.91 | 36.18 |
| ViT-B/32 | 75.75 | 81.89 | 82.59 | 9.64 | 41.29 | 26.83 | 30.89 |
| **CLIP**-ViT-L/14 | 75.03 | 80.68 | 81.95 | 71.28 | 87.74 | 58.23 | 66.32 |
| *(b) Upper-bound accuracy using N = 36 crops* | | | | | | | |
| Random | 3.60 | 3.60 | 3.60 | 18.00 | 18.00 | 3.60 | 31.85 |
| AlexNet | 85.19 | 90.30 | 89.74 | 31.37 | 47.04 | 24.40 | 49.17 |
| VGG-16 | 92.30 | 96.08 | 95.81 | 46.69 | 52.86 | 34.34 | 62.94 |
| ResNet-18 | 92.08 | 95.97 | 95.73 | 47.48 | 58.85 | 37.91 | 63.08 |
| ResNet-50 | 94.46 | 97.36 | 97.40 | 55.68 | 61.42 | 41.71 | 69.60 |
| ViT-B/32 | 95.05 | 97.61 | 97.88 | 68.43 | 68.77 | 49.10 | 70.30 |
| **CLIP**-ViT-L/14 | 94.19 | 97.32 | 97.56 | 97.16 | 98.60 | 83.77 | 89.59 |
| *(c) Upper-bound accuracy using N = 324 crops* | | | | | | | |
| Random | 32.40 | 32.40 | 32.40 | 100.00 | 100.00 | 32.40 | 100.00 |
| AlexNet | 90.03 | 93.85 | 93.48 | 42.23 | 55.52 | 29.53 | 59.65 |
| VGG-16 | 95.30 | 97.90 | 97.66 | 58.27 | 60.88 | 39.90 | 71.85 |
| ResNet-18 | 95.15 | 97.76 | 97.55 | 58.87 | 66.89 | 43.68 | 71.44 |
| ResNet-50 | 96.78 | 98.62 | 98.57 | 66.68 | 68.84 | 47.64 | 76.83 |
| ViT-B/32 | **97.19** | **98.75** | **98.91** | 78.03 | 75.58 | 55.99 | 79.28 |
| **CLIP**-ViT-L/14 | 96.78 | 98.69 | 98.80 | **98.45** | **99.20** | **89.00** | **93.13** |

Figure 2: IN+Real samples that are not correctly classifiable by IN-trained models using any of the 324 zoom transforms.

**Results** Table 1 shows upper-bound accuracy over different values of $N = \{1, 36, 324\}$. First, the random baselines (given $N = 324$ attempts per image) are at $32.4\%$ for 1,000 classes (IN, ReaL, IN+ReaL, and IN-Sketch), and $100\%$ for 200 and 313 classes (IN-A and ON, respectively). Yet, the accuracy of models with optimal zooming is far from random—*e.g.* ResNet-50 largely outperforms not only the random baseline but also the 1-crop baseline on IN ($96.78\%$ vs. $75.75\%$; Tab. 1a vs. c).

On the IN, ReaL, and IN+ReaL datasets, there is a substantial gap for all models (around +20 to +35 points) between the 1-crop and the optimal zooming setting (Tab. 1a vs. c). Surprisingly, given optimal zooming, the 11-year-old AlexNet actually can correctly label over $90\%$ of IN images, which is roughly the 1-crop accuracy ($87.8\%$) of the 2022 state-of-the-art ConvNexts [44]. This result is consistent with our hypothesis: One way for state-of-the-art classifiers to obtain their current accuracy is to simply learn *how to zoom* on top of the same, old feature extractors (*e.g.* that of AlexNet).

**Unclassifiable IN images** Interestingly, even with optimal zooming, no model reaches $100\%$ on IN images. We find that $0.39\%$ of the IN+ReaL images were not classified correctly by any of the IN-trained classifiers and these images are similar to natural adversarial images (Fig. 2) and can be categorized into four groups:

1. **Lack of information** (Fig. 2a): Images lack adequate signals for classification due to low light, occlusion, blurriness, or noise.

2. **Rare cases** (Fig. 2b): Images depict the primary object but in an uncommon form, pose, or rendition.

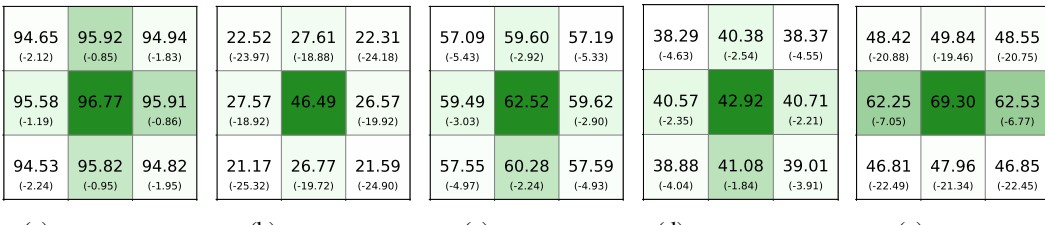

| | | | | | | | | | | | | | | |
|---|---|---|---|---|---|---|---|---|---|---|---|---|---|---|
| 94.65 (-2.12) | 95.92 (-0.85) | 94.94 (-1.83) | 22.52 (-23.97) | 27.61 (-18.88) | 22.31 (-24.18) | 57.09 (-5.43) | 59.60 (-2.92) | 57.19 (-5.33) | 38.29 (-4.63) | 40.38 (-2.54) | 38.37 (-4.55) | 48.42 (-20.88) | 49.84 (-19.46) | 48.55 (-20.75) |
| 95.58 (-1.19) | 96.77 | 95.91 (-0.86) | 27.57 (-18.92) | 46.49 | 26.57 (-19.92) | 59.49 (-3.03) | 62.52 | 59.62 (-2.90) | 40.57 (-2.35) | 42.92 | 40.71 (-2.21) | 62.25 (-7.05) | 69.30 | 62.53 (-6.77) |
| 94.53 (-2.24) | 95.82 (-0.95) | 94.82 (-1.95) | 21.17 (-25.32) | 26.77 (-19.72) | 21.59 (-24.90) | 57.55 (-4.97) | 60.28 (-2.24) | 57.59 (-4.93) | 38.88 (-4.04) | 41.08 (-1.84) | 39.01 (-3.91) | 46.81 (-22.49) | 47.96 (-21.34) | 46.85 (-22.45) |
| (a) ImageNet-ReaL | | | (b) ImageNet-A | | | (c) ImageNet-R | | | (d) ImageNet-Sketch | | | (e) ObjectNet | | |

Figure 3: Upper-bound accuracy (%) of ResNet-50 at each of the 9 zoom locations. The large gaps between the center and eight off-center locations on IN-A and ON demonstrate a center bias, which is much smaller in IN (Appendix B.2) and IN-R (b). The values in parentheses indicate the delta with respect to the center crop.

3. **Illusions** (Fig. 2c): Images have misleading elements, like a shadow appearing as a staircase, leading to misclassification.

4. **Many objects** (Fig. 2d): Images displaying several classes of objects but not all classes are listed in the set of groundtruth labels.

**OOD datasets pose a significant challenge to IN-trained models despite optimal zooming.** Across IN-A, IN-R, and ON, all IN-trained models perform far below the 324-crop random baseline (100%) with the highest score being 79.28% (Tab. 1). In contrast, CLIP reaches far better scores than IN-trained models (98.45% on IN-A and 99.20% on IN-R; Tab. 1). Our result suggests that OOD images (*e.g.* objects in unusual poses or renditions) require a more robust feature extractor to recognize besides zooming. And that CLIP was trained on an Internet-scale dataset [53] and thus is much more familiar with a variety of poses, styles, and shapes of objects [21].

Among the 324 zoom transformations, for each (classifier, dataset) pair, we initially construct a bipartite graph connecting transforms to images based on their correct classification. With this graph, we employ the iterative, greedy minimum-set cover algorithm [64, 35] to compute the minimum set of transforms required to achieve the upper-bound accuracy detailed in Sec. 4.1. Through this process, we discover that, on average, only 70% of the transforms are essential. Furthermore, we identify the **top-36 zoom transforms most important to classification** (see visualizations in Appendix D.1). More details on this process can be found in Appendix B.4.

The upper-bound accuracy using 36 crops (Tab. 1b) is only *slightly lower* than that when using all 324 crops but is substantially higher than (1) the standard 1-crop, *e.g.* 85.19% vs. 56.16% for AlexNet on IN (Tab. 1b); and (2) the random baseline (*i.e.* 3.6% for IN). Our result confirms that these 36 zoom transforms are indeed important to classification (not because models are given 36 random trials per image) and that studying them might reveal interesting insights into the datasets.

As our 324 transforms include both zoom-in and zoom-out, we further analyze the contribution of each zoom type to each dataset. We find that, across 7 datasets, zoom-in is more useful than zoom-out. And that **zoom-out is the most important to abstract images** *i.e.*, of IN-R and IN-S (Appendix B.6).

### 4.2   ImageNet-A and ObjectNet suffer from a severe center bias

The standard image pre-processing for IN-trained models involves resizing the image so its smaller dimension is 256, then taking the center $224 \times 224$ crop of the resized image [2, 38]. While suitable for ImageNet, this pre-processing may not be optimal for every OOD dataset, not allowing a model to fully utilize off-center visual cues (which optimal zooming could). Leveraging the minimum set of transforms obtained in Appendix B.4, we quantify which spatial locations (out of 9 anchors; Fig. 1b) contain the most discriminative features in each dataset. That is, we compute the upper-bound accuracy for each of the 9 anchor points per dataset and discover biases in some benchmarks.

**Experiment**   For each image, we have 9 anchors (Fig. 1b) and the originally $K = 36$ zoomed versions per anchor as defined in Sec. 3. Yet, after reducing to the minimum set (Appendix B.4), $K$ averages at 25, over all datasets, and $10 \leq K \leq 31$. Here, we count the probability that the $K$ zoomed versions per anchor lead to a correct prediction. In other words, we compute the upper-bound accuracy as in Sec. 4.1 but for each anchor separately.

**Results** First, as expected, the upper-bound accuracy for each anchor (Fig. 3) is consistently lower than when all 9 anchors are allowed (Tab. 1c). Second, across all 6 datasets, the center anchor consistently achieves the highest upper-bound accuracy versus the other 8 locations (Fig. 3 and Appendix B.2), indicating a center bias in all datasets. However, we find this bias is small in IN, IN-R, and IN-S but large in IN-A and ON (*i.e.* the largest difference between center accuracy and the lowest off-center accuracy is around -25 and -23 points, respectively; whereas for other datasets, it is around (-1) to (-5) points, as shown in Fig. 3).

The center bias in ObjectNet can be explained by the fact that the images were captured using smartphones with aspect ratios of 2:3 or 9:16 (Appendix D.3.3). Overall, such strong center bias in IN-A and ON may not be desirable since improvements on these two benchmarks may be attributed to learning to zoom to the center as opposed to the intended quest of recognizing objects in unusual forms (IN-A) or poses (ON). By merely upscaling the image and center cropping, we can achieve higher accuracy using nearly all the same models on these two datasets (Figs. A14 and A17).

We also find that, during test time, center-zooming (Appendix B.5) increases the top-1 accuracy of all IN-trained models but not CLIP, even on IN-A and ON images. This observation is intriguing considering these OOD datasets contain more distracting objects than ImageNet images (Appendix C.2) and therefore, center-zooming *should* de-clutter the scene for more accurate classification. However, CLIP prefers a specific zoom scale that provides sufficient background for object recognition—it struggles to identify a single object in a tightly-cropped image [85]. Future research should examine whether this "zoom bias" of CLIP is due to its image- or text-encoder, or both.

## 4.3 Test-time augmentation of MEMO with *only* zoom-in transforms improves accuracy

Aggregating model predictions over zoom-in versions of the input image during test time leads to higher top-1 accuracy on IN, IN-ReaL, IN-A and ON, but lower accuracy on IN-R and IN-S (Appendix B.7). However, interestingly, always zooming out on IN-R and IN-S abstract images also hurts accuracy, suggesting that an adaptive zooming strategy might be a better approach.

Here, we test building such an adaptive test-time zooming strategy by modifying MEMO [83], a SOTA test-time augmentation method that finetunes a pre-trained classifier at *test* time to achieve a more accurate prediction. Specifically, MEMO finds a network that produces a low-entropy predicted label over a set of $K = 16$ augmented versions of the test image $I$ and then runs this finetuned model on $I$ again to produce the final prediction. It does this by applying different augmentations to the test point $I$ to get augmented points $I_1, \ldots, I_K$, passing these through the model to obtain predictive distributions, and updating the model parameters by minimizing the entropy of the averaged marginal distribution over predictions. While improving accuracy, MEMO requires a pre-defined set of diverse augmentation transforms (*e.g.* sheer, rotate, and solarize in AugMix [26]). Yet, the hyperparameters for each type of transform are hard-coded, and the contribution of each transform to improved classification accuracy is unknown.

We improve MEMO's accuracy and interpretability by replacing AugMix transforms with only zoom-in functions. Intuitively, a model first looks at all zoomed-in frames of the input image (at different zoom scales and locations) and then decides to achieve the most confident prediction.

**Experiment** MEMO relies on AugMix [25], which applies a set of 13 image transforms, such as translation, rotation, and color distortion, to an original image at varying intensities, and then *chains* them together to create $K = 16$ new augmented images (examples in Appendix D.5).

We replace AugMix with `RandomResizedCrop` [4] (RRC), which takes a random crop of the input image (*i.e.* at a random location, random rectangular area, and a random aspect ratio) and then resizes it to the fixed 224×224 (*i.e.* the network input size). RRC basically implements a random zoom-in function (examples in Appendix D.5).

We compare the original MEMO [83] (which uses AugMix) and our version that uses RRC on five benchmarks (IN, IN-A, IN-R, IN-S, and ON). We follow the same experimental setup as in [83] (*e.g.* $K = 16$). Specifically, we test three ResNet-50 variants that were pre-trained using distinct augmentation techniques.[2]

---

[2]The ResNet-50 model used as a baseline in this Sec. 4.3 is different from that in our other (non-MEMO) sections of the paper.

We utilize Grad-Cam [58] to understand the impact of MIMO on the network's attentions within the final layer, both before and after modification. Specifically, our investigation seeks test our hypothesis concerning the model's focus on the regions of interest within an image.

**Results** Both our MEMO + `RRC` and the original MEMO + AugMix [83] consistently outperform the baseline models, which do not use MEMO, on all five datasets (Tab. 2). That is, when combined with MEMO, zoom-in transforms implemented via `RRC` are also helpful in classifying IN-S and IN-R images—where we previously find zoom-in to *not* help in mean/max aggregation (Appendix B.7).

On average, over all three models and five datasets, our `RRC` outperforms AugMix by +0.28 points, with a larger impact on IN-A, where it achieves a mean improvement of +1.10 points (Tab. 2). Our results show that zoom-in alone can be a useful inductive bias, helping improve downstream image classification. In contrast, some of the transformations among the 13 transform functions in AugMix may not be essential to the results of Zhang et al. [83] (no ablation studies of transformations were provided in [83]) and are less effective than our zoom-in.

Table 2: MEMO + `RRC` (*i.e.* random zoom-in transforms) **outperforms** baselines and MEMO [83].

| | Baseline (1-crop) | | | | | MEMO + AugMix [83] | | | | | MEMO + `RRC` (**Ours**) | | | | |
|---|---|---|---|---|---|---|---|---|---|---|---|---|---|---|---|
| | IN | IN-A | IN-R | IN-S | ON | IN | IN-A | IN-R | IN-S | ON | IN | IN-A | IN-R | IN-S | ON |
| ResNet-50 [23] | 76.13 | 0.00 | 36.17 | 24.09 | 35.92 | 77.27 | 0.83 | **41.28** | **27.63** | 38.38 | **77.50** | **1.31** | 40.81 | 27.53 | **38.85** |
| DeepAug+AugMix [26] | 75.82 | 3.87 | 46.77 | 32.62 | 34.81 | 76.27 | 5.35 | 50.79 | **35.70** | 36.42 | **76.38** | **5.76** | **50.88** | 35.65 | **36.64** |
| MoEx+CutMix [40] | 79.04 | 7.97 | 35.52 | 23.96 | 38.59 | 79.38 | 11.21 | **40.65** | **27.07** | 40.62 | **79.49** | **13.61** | 40.41 | 26.80 | **41.43** |
| *mean ± std* | 36.75 ± 24.75 | | | | | 39.26 (+2.51) ± 24.32 | | | | | 39.54 (+2.79) ± **24.10** | | | | |

Figure 4 shows Grad-CAM visualizations for three samples, providing evidence of how the network's behavior changes before and after the MEMO update. For an image of a `pug`, the network initially focused on a kitchen appliance, failing to detect the object correctly. After applying the MEMO modification, it refocused on the dog, classifying it accurately. Similarly, in an image of a `fox squirrel`, the network initially had a diffuse focus but refocused on the fox squirrel after the update. These results demonstrate the effectiveness of the MEMO modification in guiding the network's attention or encouraging the model to perform an implicit zoom on the regions of interest, thereby improving its classification performance.

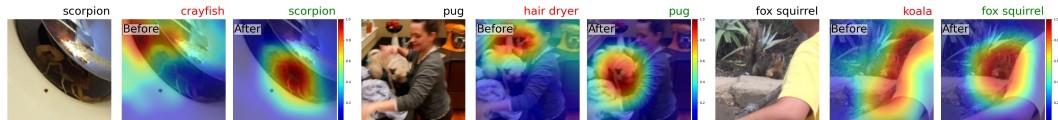

Figure 4: Grad-CAM for the activation of the last convolutional layer of a ResNet-50 before and after the MEMO update suggests that the network attends to the object of interest after the update.

### 4.4 ImageNet-Hard: A benchmark with images that remain unclassifiable, even after 324 zoom attempts

Existing ImageNet-scale benchmarks followed one of the following three construction methods: (1) perturbing real images with the aim of making them harder for models to classify (*e.g.*, ImageNet-C [24] and DAmageNet [13]); (2) collecting the real images that models misclassify (*e.g.*, IN-A, ImageNet-O [27]); or (3) setting up a highly-controlled data collection process (*e.g.*, IN-S and ON). Yet, none of such benchmarks explicitly challenge models on the ability to recognize a well-framed object in an image (*i.e.*, no zooming required). For example, ON is supposed to test the recognition of objects in unusual poses but the cluttered background in ON images is actually a major reason for misclassification (Sec. 4.2). Furthermore, the results in Tab. 1 suggest that given optimal zooming, these existing benchmarks only challenge IN-trained models but not the Internet-scale vision-language models (*e.g.* CLIP) anymore. We propose ImageNet-Hard, a novel ImageNet-scale benchmark that challenges existing and future SOTA models. ImageNet-Hard is a collection of images that the SOTA CLIP-ViT-L/14 fails to correctly classify even when 324 zooming attempts are provided.

### 4.4.1 ImageNet-Hard construction

**Initial data collection** We take CLIP-ViT-L/14 (the highest-performing model in Tab. 1) and run the zooming procedure to find "Unclassifiable images" (defined in Sec. 4.1) from the following six datasets: IN-V2 [56], IN-ReaL, IN-A, IN-R, IN-S, and ON. That is, for each image $x$, we generate 324 zoomed versions of $x$ and feed them into CLIP-ViT-L/14. We add $x$ to ImageNet-Hard only if none of the 324 versions are correctly classified.

**Adding ImageNet-C** The original IN-C [24] are the original IN images but center-cropped to 224 × 224 px, which significantly makes the classification task unnecessarily more ill-posed (*e.g.*, by adding Gaussian noise to a crop where the main object is already removed).

To find a subset of IN-C images for adding into ImageNet-Hard, we first re-generate ImageNet-C by adding the 19 types of corruption noise to IN without resizing the original IN images. Second, we run CLIP-ViT-L/14 on all 19 corruption types and manually select a subset of six diverse and lowest-accuracy corruption groups: Impulse Noise, Frost, Fog, Snow, Brightness, and Zoom Blur. We repeat the initial data collection process for these 6 image sets of IN-C.

**Groundtruth labels** After the above procedure, our dataset contains 13,925 images collected from IN+ReaL, IN-V2, IN-A, IN-C, IN-R, IN-S, and ON (see the distribution in Appendix E.1). ImageNet-Hard presents a 1000-way classification task where the 1000 classes are from ImageNet. We manually inspect all images and remove 295 samples that are obviously ill-posed (e.g. an entirely black image but labeled `great white shark` in IN-S Fig. A60), arriving at a total of 13,630 ImageNet-Hard images. A sample contains only one groundtruth label from its original datasets except for IN and IN-C images, which have a set of IN+ReaL labels. Each IN or IN-C image is considered correctly labeled by a model if its top-1 predicted label is among the groundtruth labels.

**Refining groundtruth labels via human feedback** Label noise is still present in IN and OOD benchmarks despite cleaning efforts [10, 56, 81]. Since ImageNet-Hard contains images misclassified by CLIP-ViT-L/14, our manual inspection confirms many misclassified images have debatable labels.

To ameliorate the issue, we orchestrate a human feedback study for eliminating images with inaccurate labels. First, the first author examine every image and flag 3,133 images as ambiguous and needs verification. Then, we have two groups of annotators to help verify the labels (by choosing Accept, Reject, or Not Sure). Group A is composed of three students, each examine all 3,133 images where Group B is composed of 38 students, each examine 50 randomly-selected images. Our inter-annotator aggregation procedure merges labels from both groups and results in 2,280 images removed (out of 3,133 originally flagged), leaving ImageNet-Hard at a total of 11,350 images.

That is, we accept an image $x$ if one of the two conditions is satisfied: (1) when all 3/3 group-A annotators accept $x$; or (2) when 2/3 group-A annotators accept $x$ and all group-B reviewers of $x$ accept $x$ (assuming at least 1 group-B annotator reviews $x$; otherwise $x$ will be rejected).

Inspired by IN-ReaL [10], we further clean up the labels by eliminating 370 images associated with the labels `sunglass`, `sunglasses`, `tub`, `bathtub`, `cradle`, `bassinet`, `projectile`, and `missile`, *i.e.*, the classes that often contain similar images that belong to more than one class. After this refinement, the final ImageNet-Hard dataset contains a total of **10,980 images**.

**4K version** We utilize GigaGAN [33] to upscale every image in our final dataset and construct ImageNet-Hard-4K, which is aimed to facilitate future research into how a super-resolution step may improve image classification results (*e.g.*, to classify an object when the image is blurry).

**Release** ImageNet-Hard and ImageNet-Hard-4K are released on HuggingFace (see samples in Fig. A49) under MIT License. Code for evaluating models on ImageNet-Hard is on GitHub.

### 4.4.2 ImageNet-Hard challenges SOTA classifiers, especially those operating at 224×224

Here, we evaluate the standard 1-crop, top-1 accuracy of SOTA classifiers on ImageNet-Hard. We use the image pre-processing function defined by each classifier. In addition to the 6 models in Sec. 4.1, we also test CLIP-ViT-L/14@336px [53], EfficientNet (B0@224px and B7@600px) [69], and EfficientNet-L2@800px [1]. CLIP-ViT-L/14@336px, EfficientNet-B7@600px, and EfficientNet-L2@800px are state-of-the-art models that operate at high resolutions of 336×336, 600×600, and 800×800 respectively. In addition, our evaluation includes models from the OpenCLIP family [30].

**Results** Tab. 3 shows fairly low top-1 accuracy by various classifiers on ImageNet-Hard. First, all well-known IN-trained classifiers that operate at 224×224 perform poorly between 7.34% (AlexNet) and 18.52% accuracy (ViT-B/32).

Since ImageNet-Hard is based on a collection of images that OpenAI's CLIP ViT-L/14@224px mislabels, this classifier's accuracy on our dataset is only 1.86%. Yet, interestingly, CLIP-ViT-L/14@336px also performs poorly at 2.02% (Tab. 3). Furthermore, all 68 tested OpenCLIP models perform poorly, with an accuracy below 16% (see details in Appendix E.6).

Separately, we observe a trend that models operating at a higher resolution tend to perform better on ImageNet-Hard with EfficientNet-L2@800px scoring highest at 39.00% (compared to 88.40% [79] on the original ImageNet). Overall, all models perform substantially worse on ImageNet-Hard (Tab. 3) than on other ImageNet-scale datasets (see Tab. A6; 1-crop). This result is expected because ImageNet-Hard is a set of hard cases collected from those OOD benchmarks.

**ImageNet-Hard-4K** We find that when upsampling images to 4K using GigaGAN [33] and downsampling them back to the resolution of each classifier does not help but even hurt the accuracy slightly (Tab. A13). Given that GigaGAN performs remarkably well, this result suggests ImageNet-Hard is different from typical fine-grained animal classification where improving the texture details increases classification accuracy [76]. The next section (Sec. 4.5) sheds light on model failures on ImageNet-Hard, revealing challenges posed to future SOTA models.

Table 3: Top-1 accuracy (%) on ImageNet-Hard of IN-trained models and those trained on larger, non-ImageNet datasets (black). All models operate at 224×224 unless otherwise specify.

| Classifier | Accuracy | Classifier | Accuracy | Classifier | Accuracy |
|---|---|---|---|---|---|
| AlexNet | 7.34 | ViT-B/32 | 18.52 | CLIP-ViT-L/14@224px | 1.86 |
| VGG-16 | 12.00 | EfficientNet-B0@224px | 16.57 | CLIP-ViT-L/14@336px | 2.02 |
| ResNet-18 | 10.86 | EfficientNet-B7@600px | 23.20 | OpenCLIP-ViT-bigG-14 | 15.93 |
| ResNet-50 | 14.74 | EfficientNet-L2@800px | 39.00 | OpenCLIP-ViT-L-14 | 15.60 |

## 4.5 Analysis of Image Classification Errors

Motivated by the fact that EfficientNet-L2 is the best classifier on ImageNet-Hard, we qualitatively analyze its failure cases to characterize the challenge posed by our benchmark. Specifically, we provide `gpt-3.5-turbo` [51] with a pair of EfficientNet-L2's top-1 (incorrect) label and the groundtruth label and ask it to categorize the error into "common" or "rare" based on the labels' semantic similarity (see Appendix E.3 for full details). For instance, mislabeling `bucket` into `barrel` is common (as two objects are quite related) while mislabeling `cloak` into `jigsaw puzzle` is rare.

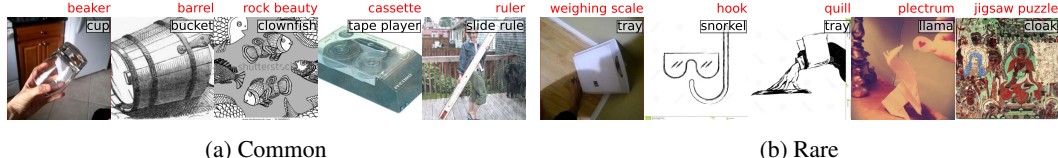

    (a) Common            (b) Rare

Figure 5: ImageNet-Hard samples misclassified by EfficientNet-L2@800px can be categorized into two groups: (a) **Common**: the top-1 label is related to the groundtruth label; and (b) **Rare**: the top-1 label is semantically far from the groundtruth label. See Figs. A56 and A57 for more samples.

**Results** See Appendix E.4 for samples of wrong labels that EfficientNet-L2 most frequently misclassifies into. We find that 39.4% of EfficientNet-L2's misclassifications on the ImageNet-Hard dataset are "common", while 60.6% are "rare".

**A. Common** group captures model confusion between two related classes (e.g. two fish species: `clownfish` and `rock beauty`; Fig. 5a). Yet, another source of problem for these "errors" is the debatable groundtruth labels, which may require domain-expert annotators to verify and rectify [45].

**B. Rare** group captures errors where the model confusion is between two semantically distant classes (e.g., `llama` → `plectrum`; Fig. 5b). This often happens with abstract images or objects in unusual

poses [5] or forms [15]. Classifying this group of images is challenging and sometimes requires a strong understanding of context and reasoning capabilities.

# 5    Discussion and Conclusion

**Limitations**   By manual inspection, we estimate 14.7% of labeling noise, which ImageNet-Hard inherits from the source datasets.

Our study rigorously analyzed the zooming effect on six known classifiers and image classification benchmarks. We first demonstrate that previous state-of-the-art classifiers, as old as AlexNet [38], could potentially achieve near 90% accuracy with optimal zooming. This sparks the intriguing question of whether image classifiers' evolution over the past ten years is about mastering where and at what scale to zoom (instead of enhancing feature extractors, a.k.a. representation learning [3]). Through another lens, we probe the evolution by analyzing the implicit zooming mechanisms that deep classifiers apply to input images. This perspective diverges from [54], which studied the progression of representation learning from CNNs to ViTs.

We are the first to document the spatial biases of existing benchmarks. Notably, IN-A and ON contain a large center bias and simply zooming to the center will de-clutter the scene and yield a high accuracy (24.69% for ViT-B/32 on IN-A; Tab. A5), which is competitive with state-of-the-art trained models (*e.g.* 24.1% of Robust ViT [11]) and much higher than state-of-the-art TTA techniques (*e.g.* 11.21% of MEMO [83]; Tab. 2). Our simple, but strong zoom-in baselines on IN-A and ON motivate future research into better-controlled benchmarks that more explicitly test models on a set of pre-defined properties. Our proposed TTA method with zoom-in transforms (MEMO + RRC) is not only more accurate but also more interpretable and faster to run (Tab. A7) than the original MEMO.

Finally, we introduce ImageNet-Hard (Sec. 4.4), a new challenging dataset for SOTA IN-trained and vision-language classifiers.

# Acknowledgement

GN is supported by Auburn University PGRF Fellowship. AN was supported by a NSF CAREER award No. 2145767, and donations from NaphCare Foundation, and Adobe Research. We greatly appreciate David Seunghyun Yoon's help in generating the ImageNet-Hard-4K (Sec. 4.4.2) version using GigaGAN. We also thank those students at Auburn University and Alberta University who participated in our experiment for cleaning up the labels of ImageNet-Hard images.

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

# Appendix for:
# ImageNet-Hard: The Hardest Images Remaining from a Study of the Power of Zoom and Spatial Biases in Image Classification

## A   Implementation details

In this section, we provide a detailed description of our experimental setup, including the Python code for our zoom transform, the classifiers we employed, and the setup we used for zero-shot classification.

### A.1   Sample Python code for zoom-based transform

```python
from PIL import Image
import torchvision.transforms.functional as fv
import torchvision.transforms as transforms
from functools import partial

def crop_at(size, slice_x, slice_y):
    def slice_crop(image, size, slice_x, slice_y):
        width, height = image.size
        tile_size_x = width // 3
        tile_size_y = height // 3
        anchor_x = (slice_y * tile_size_x) + (tile_size_x // 2)
        anchor_y = (slice_x * tile_size_y) + (tile_size_y // 2)
        return fv.crop(
            image,
            anchor_y - (size // 2),
            anchor_x - (size // 2),
            size,
            size,
        )
    return partial(slice_crop, size=size, slice_x=slice_x, slice_y=
    slice_y)

zoom_scale = 255
zoom_transform = transforms.Compose(
                [
                        transforms.Resize(
                            zoom_scale,
                            interpolation=transforms.InterpolationMode
    .BICUBIC,
                            max_size=None,
                            antialias=None,
                        ),
                        crop_at(224, i, j),
                ]
            )
```

Figure A1: Sample python code.

## A.2   Datasets' licenses

| Dataset Name | License |
|---|---|
| ImageNet | Custom license, non-commercial |
| ImageNet-A | License |
| ImageNet-R | MIT License |
| ImageNet-Sketch | MIT License |
| ImageNet-C | MIT License |
| ObjectNet | Custom license derived from Creative Commons Attribution 4.0 |
| ImageNet-V2 | MIT License |

Table A1: Dataset Licenses

## A.3   Zoom Scales used

In our experiments, we tried the following zoom scales:

$10, 16, 32, 48, 64, 96, 122, 128, 192, 224, 235, 240, 256, 288, 320, 348, 384, 448, 460, 512,$
$573, 576, 640, 664, 672, 680, 686, 690, 700, 720, 768, 798, 832, 896, 911, 1024.$

## A.4   Model selection

We use the official OpenAI's official CLIP for all CLIP-related experiments. All IN-trained models are retrieved from the `torchvision` [48] library. For models from the OpenCLIP family, we utilize the OpenCLIP library version `2.20.0`. In the case of the EfficientNet-B family, we use the Hugging Face Transformers library. Lastly, for EfficientNet-L2, we use the implementation from the timm library.

### A.5 Zero-shot classification using CLIP

For CLIP, we follow the standard zero-shot classification. This involves creating a text template for each class in the dataset, which contains a generic description of an image featuring an object from that class. Then, we use CLIP's text encoder to obtain embeddings for these templates and then average them to obtain a final vector that represents the class. To classify an image, we calculate the cosine similarity between its embedding and the text vectors for each class and then select the class with the highest value.

### A.6 Zoom-based transform

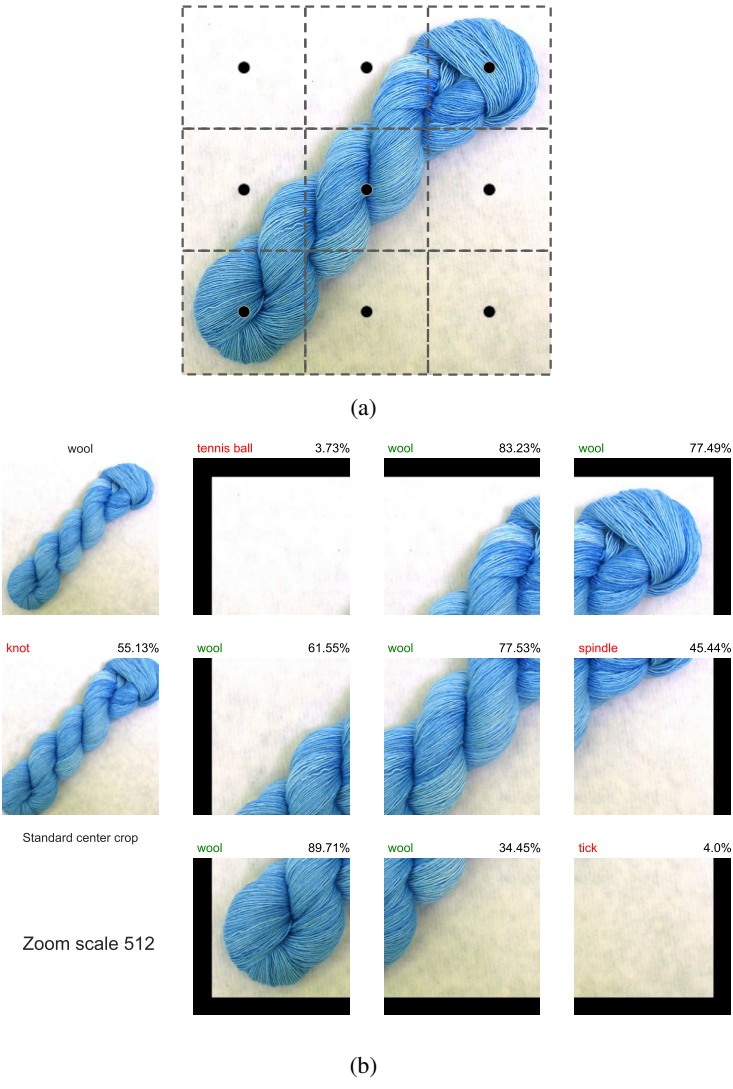

(a)

(b)

Figure A2: (a) Making a 3-by-3 uniform grid out of the image. We pick the center point in each region as the anchor. (b) Sample image showing how our zoom transform is applied to an image.

# B  Additional Results

In this section, we provide additional results for our experiments.

## B.1  Zooming out is needed for a small portion of the datasets

In our approach, we leverage the power of both zoom-in and zoom-out transforms, and Tab. 1 results indicate that this combined zooming approach can be effective in classifying images from diverse datasets. Zooming in enhances texture patterns while zooming out provides a better perspective of the object's shape. The question we aim to answer is which dataset and model pairs require which type of zoom, and whether zooming is always necessary. Additionally, we investigate which types of networks are less reliant on explicit zooming, as they implicitly focus on the main object in the image.

**Experiment**   We separate zoom transforms into three groups and report the maximum possible accuracy as defined in Sec. 3. We use transforms in the minimum set covers (as shown in Fig. A10) for each dataset and classifier pair. We then report the number of images that can only be classified using transforms in each group separately.

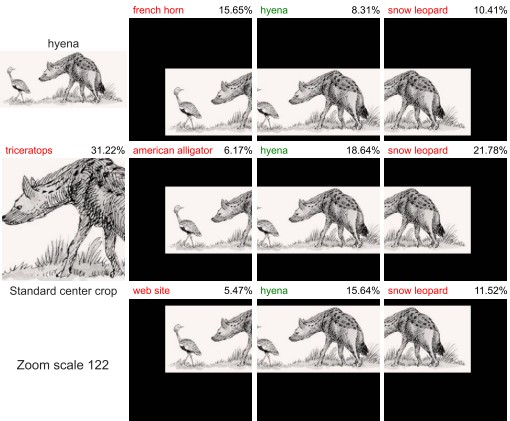

Figure A3: A sample image from the ImageNet-Sketch dataset that can only be solved by zooming out. For this image, with the standard ImageNet transform, the entire body of the animal is not visible. Instead, zooming out of the image helps you see the whole body of the animal. More samples can be found in Appendix D.3.

**Results**   In general, we find that zooming in is more effective than zooming out. Zooming in provides two benefits: (1) it helps the model to focus on the key region where the target object is located, and (2) the model can extract features from the target object at a higher resolution. Across all methods and datasets, we can see a certain percentage of images are only classifiable using transforms of the *zoom-out* group. In particular, for ImageNet-R and ImageNet-Sketch, between $1.2\% - 3\%$ (Table A2) of the entire dataset can only be solved using a transform in the *zoom-out* group. This is especially true for drawings, where the texture may lack distinguishable features, and zooming out allows us to better perceive the shape.

Table A2: Breakdown of maximum possible accuracy by different zoom groups. In each dataset, certain images necessitate a specific zoom group for correct classification regardless of the model being used. However, CLIP performs well overall without depending heavily on a particular zoom level. On average, the percentage of datasets that can only be solved with a specific zoom group is very small for this model.

| Dataset | Model | *zoom-in* Solve | *zoom-out* Solves | *zoom-224* Solves | Only *zoom-in* Solves | Only *zoom-out* Solves | Only *zoom-224* Solves |
|---|---|---|---|---|---|---|---|
| ImageNet | ResNet-18 | 94.57 | 79.49 | 81.16 | 10.59 | 0.43 | 0.08 |
| | ResNet-50 | 96.30 | 85.84 | 86.39 | 7.59 | 0.40 | 0.04 |
| | ViT-B/32 | 96.83 | 86.18 | 85.12 | 7.59 | 0.30 | 0.02 |
| | VGG-16 | 94.60 | 82.11 | 83.08 | 8.92 | 0.58 | 0.07 |
| | AlexNet | 89.17 | 62.92 | 67.98 | 18.01 | 0.65 | 0.18 |
| | CLIP-ViT-L/14 | 95.82 | 90.80 | 87.04 | 4.81 | 0.83 | 0.05 |
| ImageNet ReaL | ResNet-18 | 97.37 | 86.10 | 87.62 | 7.38 | 0.27 | 0.07 |
| | ResNet-50 | 98.22 | 91.07 | 91.87 | 4.65 | 0.25 | 0.04 |
| | ViT-B/32 | 98.50 | 90.79 | 88.06 | 4.92 | 0.18 | 0.03 |
| | VGG-16 | 97.38 | 88.43 | 89.40 | 6.02 | 0.38 | 0.07 |
| | AlexNet | 93.15 | 69.58 | 74.85 | 15.47 | 0.45 | 0.19 |
| | CLIP-ViT-L/14 | 98.05 | 94.44 | 91.69 | 3.20 | 0.55 | 0.04 |
| ImageNet+ReaL | ResNet-18 | 97.16 | 85.51 | 86.77 | 7.72 | 0.28 | 0.05 |
| | ResNet-50 | 98.25 | 91.10 | 91.77 | 4.60 | 0.24 | 0.03 |
| | ViT-B/32 | 98.70 | 91.00 | 90.95 | 4.92 | 0.14 | 0.02 |
| | VGG-16 | 97.12 | 87.88 | 89.09 | 6.25 | 0.42 | 0.06 |
| | AlexNet | 92.79 | 68.65 | 73.93 | 16.25 | 0.47 | 0.16 |
| | CLIP-ViT-L/14 | 98.24 | 95.09 | 92.41 | 2.75 | 0.47 | 0.04 |
| ImageNet-A | ResNet-18 | 63.66 | 47.95 | 45.37 | 13.97 | 2.75 | 0.21 |
| | ResNet-50 | 65.28 | 52.36 | 48.59 | 12.05 | 3.13 | 0.22 |
| | ViT-B/32 | 73.07 | 56.34 | 54.84 | 14.20 | 2.04 | 0.27 |
| | VGG-16 | 56.67 | 44.95 | 39.35 | 11.80 | 3.85 | 0.24 |
| | AlexNet | 52.69 | 32.86 | 31.95 | 17.15 | 2.34 | 0.30 |
| | CLIP-ViT-L/14 | 98.35 | 96.71 | 93.57 | 1.70 | 0.69 | 0.04 |
| ImageNet-R | ResNet-18 | 57.07 | 12.19 | 10.07 | 40.67 | 0.92 | 0.19 |
| | ResNet-50 | 64.52 | 12.95 | 10.36 | 48.72 | 1.00 | 0.23 |
| | ViT-B/32 | 76.71 | 18.57 | 21.92 | 51.75 | 0.85 | 0.15 |
| | VGG-16 | 56.59 | 13.15 | 13.27 | 38.24 | 0.93 | 0.29 |
| | AlexNet | 39.91 | 10.39 | 9.11 | 26.27 | 1.08 | 0.36 |
| | CLIP-ViT-L/14 | 97.99 | 81.32 | 77.03 | 12.01 | 0.44 | 0.05 |
| ImageNet-Sketch | ResNet-18 | 41.14 | 27.06 | 27.41 | 11.83 | 1.77 | 0.36 |
| | ResNet-50 | 44.72 | 32.80 | 31.45 | 10.99 | 2.23 | 0.24 |
| | ViT-B/32 | 53.45 | 37.43 | 37.38 | 13.11 | 1.83 | 0.36 |
| | VGG-16 | 36.20 | 27.20 | 24.59 | 9.47 | 2.97 | 0.28 |
| | AlexNet | 27.71 | 13.84 | 15.11 | 11.26 | 1.22 | 0.33 |
| | CLIP-ViT-L/14 | 86.20 | 80.67 | 73.94 | 6.64 | 2.38 | 0.12 |
| ObjectNet | ResNet-18 | 68.98 | 38.52 | 37.23 | 25.76 | 1.93 | 0.25 |
| | ResNet-50 | 74.16 | 51.56 | 47.79 | 19.68 | 2.16 | 0.30 |
| | ViT-B/32 | 77.66 | 44.49 | 42.65 | 27.43 | 1.34 | 0.20 |
| | VGG-16 | 69.19 | 41.72 | 39.49 | 23.34 | 2.27 | 0.31 |
| | AlexNet | 56.76 | 23.45 | 22.59 | 28.85 | 2.27 | 0.33 |
| | CLIP-ViT-L/14 | 91.28 | 82.22 | 77.60 | 8.37 | 1.38 | 0.15 |
| Average | ResNet-18 | 74.28 | 53.83 | 53.66 | 16.85 | 1.19 | 0.17 |
| | ResNet-50 | 77.35 | 59.67 | 58.32 | 15.47 | 1.34 | 0.16 |
| | ViT-B/32 | 82.13 | 60.69 | 60.13 | 17.70 | 0.95 | 0.15 |
| | VGG-16 | 72.54 | 55.06 | 54.04 | 14.86 | 1.63 | 0.19 |
| | AlexNet | 64.60 | 40.24 | 42.22 | 19.04 | 1.21 | 0.26 |
| | CLIP-ViT-L/14 | 95.13 | 88.75 | 84.75 | 5.64 | 0.95 | 0.07 |

## B.2 Anchor-based analysis of Center bias in ImageNet and OOD datasets

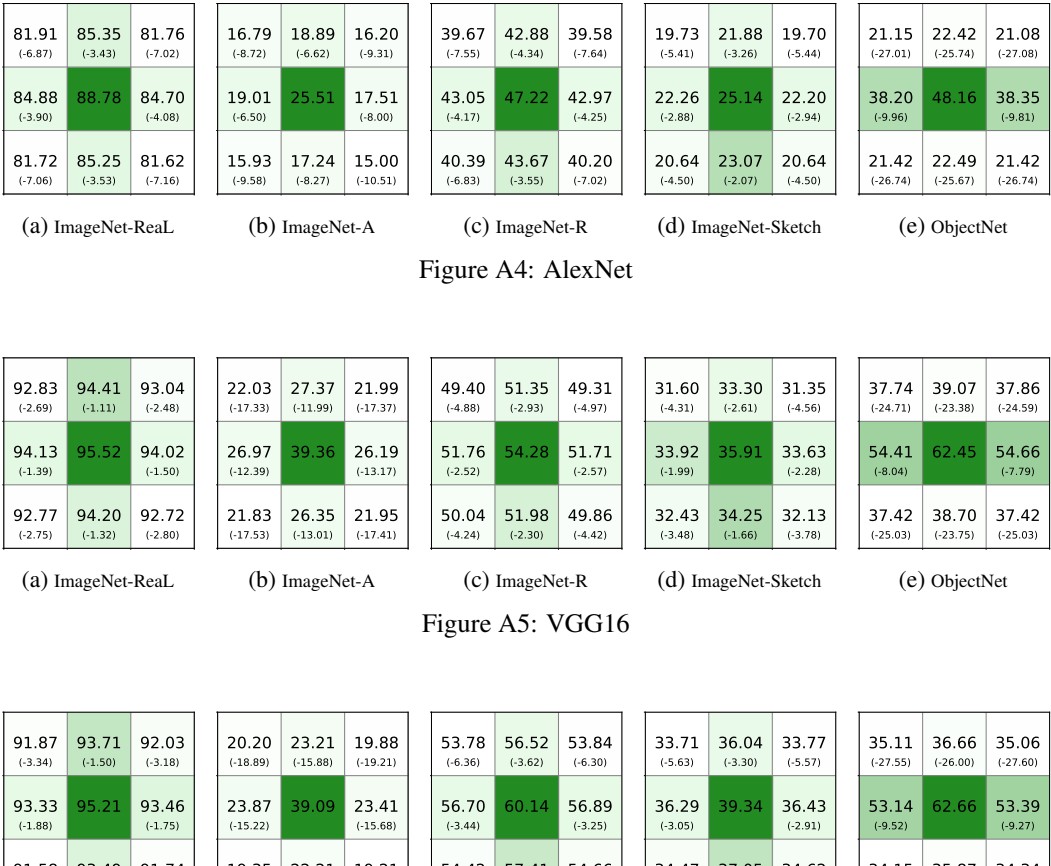

(a) ImageNet-ReaL  (b) ImageNet-A  (c) ImageNet-R  (d) ImageNet-Sketch  (e) ObjectNet

Figure A4: AlexNet

(a) ImageNet-ReaL  (b) ImageNet-A  (c) ImageNet-R  (d) ImageNet-Sketch  (e) ObjectNet

Figure A5: VGG16

(a) ImageNet-ReaL  (b) ImageNet-A  (c) ImageNet-R  (d) ImageNet-Sketch  (e) ObjectNet

Figure A6: ResNet-18

(a) ImageNet-ReaL  (b) ImageNet-A  (c) ImageNet-R  (d) ImageNet-Sketch  (e) ObjectNet

Figure A7: ResNet-50

## B.3 Distribution of the minimum set cover per classifier and dataset

In this section, we provide details on the distribution of minimum set cover size.

## B.4 Only 70% of all transforms are needed to reach maximum possible accuracy

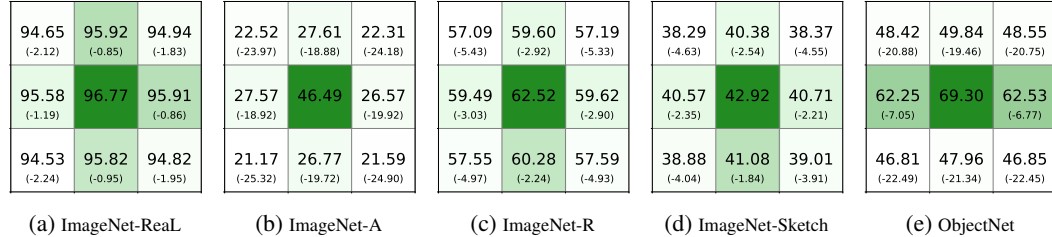

In Sec. 4.1, we first pre-define all 324 zoom transforms and then compute the *maximum* possible accuracy to ensure the predicted labels were the results of models looking at a controlled zoomed region (*i.e.* not because a model was given 324 arbitrary trials per image). Here, we aim to compute the minimum number of zoom settings required for a model to reach the same upper-bound accuracy.

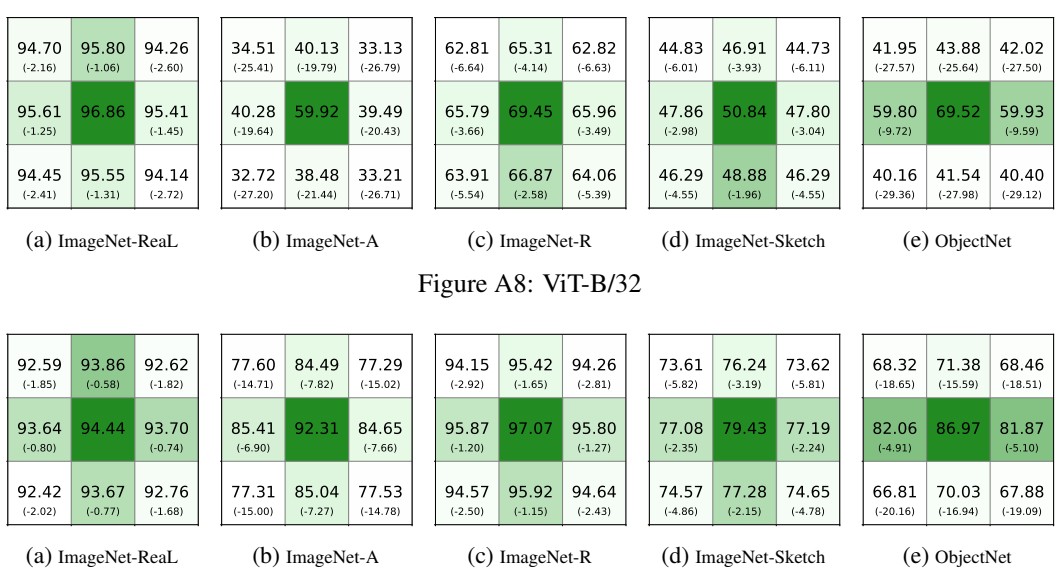

(a) ImageNet-ReaL  (b) ImageNet-A  (c) ImageNet-R  (d) ImageNet-Sketch  (e) ObjectNet

Figure A8: ViT-B/32

**(a) ImageNet-ReaL**

| 92.59 (-1.85) | 93.86 (-0.58) | 92.62 (-1.82) |
|---|---|---|
| 93.64 (-0.80) | 94.44 | 93.70 (-0.74) |
| 92.42 (-2.02) | 93.67 (-0.77) | 92.76 (-1.68) |

**(b) ImageNet-A**

| 77.60 (-14.71) | 84.49 (-7.82) | 77.29 (-15.02) |
|---|---|---|
| 85.41 (-6.90) | 92.31 | 84.65 (-7.66) |
| 77.31 (-15.00) | 85.04 (-7.27) | 77.53 (-14.78) |

**(c) ImageNet-R**

| 94.15 (-2.92) | 95.42 (-1.65) | 94.26 (-2.81) |
|---|---|---|
| 95.87 (-1.20) | 97.07 | 95.80 (-1.27) |
| 94.57 (-2.50) | 95.92 (-1.15) | 94.64 (-2.43) |

**(d) ImageNet-Sketch**

| 73.61 (-5.82) | 76.24 (-3.19) | 73.62 (-5.81) |
|---|---|---|
| 77.08 (-2.35) | 79.43 | 77.19 (-2.24) |
| 74.57 (-4.86) | 77.28 (-2.15) | 74.65 (-4.78) |

**(e) ObjectNet**

| 68.32 (-18.65) | 71.38 (-15.59) | 68.46 (-18.51) |
|---|---|---|
| 82.06 (-4.91) | 86.97 | 81.87 (-5.10) |
| 66.81 (-20.16) | 70.03 (-16.94) | 67.88 (-19.09) |

Figure A9: CLIP-ViT-L/14

Table A3: Distribution of the minimum set cover per classifier and dataset. (ZI: *zoom-in*, ZO: *zoom-out*, ZL: *zoom-224*)

| | ReaL | | | | IN-A | | | | IN-R | | | | IN-Sketch | | | | ON | | | |
|---|---|---|---|---|---|---|---|---|---|---|---|---|---|---|---|---|---|---|---|---|
| | ZI | ZO | ZL | Total | ZI | ZO | ZL | Total | ZI | ZO | ZL | Total | ZI | ZO | ZL | Total | ZI | ZO | ZL | Total |
| ResNet-18 | 160 | 33 | 8 | 201 | 174 | 31 | 6 | 211 | 204 | 65 | 9 | 278 | 209 | 51 | 9 | 269 | 191 | 54 | 9 | 254 |
| ResNet-50 | 136 | 33 | 9 | 178 | 165 | 42 | 7 | 214 | 200 | 62 | 9 | 271 | 216 | 56 | 9 | 281 | 187 | 63 | 9 | 259 |
| ViT-B/32 | 134 | 30 | 4 | 168 | 167 | 19 | 7 | 193 | 196 | 52 | 9 | 257 | 218 | 46 | 9 | 273 | 206 | 58 | 9 | 273 |
| VGG-16 | 158 | 34 | 9 | 201 | 181 | 33 | 8 | 222 | 214 | 66 | 9 | 289 | 210 | 54 | 9 | 273 | 198 | 52 | 9 | 259 |
| AlexNet | 191 | 40 | 8 | 239 | 170 | 33 | 9 | 212 | 212 | 51 | 9 | 272 | 217 | 49 | 9 | 275 | 201 | 58 | 9 | 268 |
| CLIP-ViT-L/14 | 141 | 48 | 8 | 197 | 75 | 14 | 4 | 93 | 76 | 33 | 5 | 114 | 142 | 61 | 9 | 212 | 205 | 66 | 9 | 280 |

Evaluating this minimum set may reveal spatial biases of a dataset (Sec. 4.2) as well as the implicit zoom operation that a state-of-the-art model (*e.g.* CLIP) may have learned.

**Experiment** Given a (dataset, classifier) pair, we constructed a bipartite graph $G = (N, E)$, where $N = A \cup B$, $A$ represents the set of transforms, and $B$ represents the set of images. The edges $E$ are defined as follows:

$$E = \{(n_i, n_j) \mid n_i \in A, n_j \in B, \text{ and transform } n_i \text{ leads to the correct classification of image } n_j\}$$

We aim to find a minimum set cover [64, 35] in this graph, synonymous with finding a minimum subset of transforms among the 324 that lead to the correct prediction for all classifiable images in Sec. 4 (*i.e.* those that make up the accuracy scores in Tab. 1c), without unnecessary transforms.

The resulting subset of transforms from the process leads to the correct prediction for all classifiable images without sacrificing accuracy. During each iteration of the greedy minimum set cover algorithm, the transform that yields the highest number of correct classifications for the remaining images is selected. This process continues until all of the images have been "covered," *i.e.* all images have connected to a transform with at least one edge. The result aligns with our initial goal to remove unnecessary zoom transforms while maintaining the maximum possible accuracy, as outlined in Sec. 4 (i.e., those that make up the accuracy scores in Tab. 1c). The outline of the algorithm can be seen in Algorithm 1.

**Results** Fig. A10 shows the minimum number of transforms per dataset required to reach the maximum possible accuracy. Although this number varies depending on the dataset and classifier, on average, the size of the minimum cover is 229, which is ~70% of all 324 pre-defined transforms.

We evaluate the maximum possible accuracy using the top 36 transforms, the same number as the number of zoom scales and report the results in Tab. 1b. This set of transforms is achieved by stopping the algorithm after 36 iterations, which provided us with 36 high-performing transforms. The maximum possible accuracy using only 36 crops is only slightly lower than that when using all 324 crops but is substantially higher than the standard 1-crop, *e.g.* 85.19% vs. 56.16% for AlexNet on IN (Tab. 1b). Also, the upper-bound accuracy for 36 crops being much higher than the

**Algorithm 1** Greedy Minimum Set Cover for Transforms

---
1: **Initialization:** $C = \emptyset$ (Covered set of images), $T = \emptyset$ (Selected transforms)
2: **while** $C \neq B$ **do**
3:     Find $n_i \in A \setminus T$ that maximizes $|n_j \in B \setminus C \mid (n_i, n_j) \in E|$
4:     $C = C \cup \{n_j \mid (n_i, n_j) \in E\}$
5:     $T = T \cup \{n_i\}$
6: **end while**
7: **Result:** The subset of transforms corresponding to $T$ can classify images without sacrificing accuracy.

---

Figure A10: The minimum number of zoom transforms (out of 324) required to achieve the maximum possible accuracy scores reported in Tab. 1c.

|  | IN | ReaL | IN+ReaL | IN-A | IN-R | IN-S | ON | $\mu$ |
|---|---|---|---|---|---|---|---|---|
| AlexNet | 255 | 239 | 246 | 212 | 272 | 275 | 268 | 252 |
| VGG-16 | 242 | 201 | 201 | 222 | 289 | 273 | 259 | 241 |
| ResNet-18 | 250 | 201 | 208 | 211 | 278 | 269 | 254 | 239 |
| ResNet-50 | 234 | 178 | 183 | 214 | 271 | 281 | 259 | 231 |
| ViT-B/32 | 233 | 168 | 173 | 193 | 257 | 273 | 273 | 224 |
| **CLIP**-ViT-L/14 | 251 | 197 | 186 | 93 | 114 | 280 | 212 | 190 |

random baseline (*i.e.* $3.6\%$ for IN) confirms that the pre-defined zoom transforms are important to classification (not because models are given 36 random trials per image). The top-36 zoom transforms for ResNet-50 on ImageNet contain zooms at various locations in the image (see the visualizations in Appendix D.1).

Remarkably, CLIP requires 190 transforms on average, which is fewer than every other model (Fig. A10; $\mu$ column). This can be attributed to either the implicit zoom power of CLIP or the fact it has a stronger feature extractor.

## B.5   Center-zooming increases the accuracy of all ImageNet-trained models but not CLIP

Previously, we have found that CLIP obtains the best accuracy on all six datasets (Tab. 1a) and also requires the smallest minimum set of zoom transforms to obtain the upper-bound accuracy (Appendix B.4). It is important to understand what classification strategy a CLIP classifier internally performs to classify better. Here, we test the hypothesis that the state-of-the-art CLIP is already performing an implicit zoom on images. If that is true, directly zooming to the center, exploiting the strong center bias of ImageNet-A and ObjectNet, will not improve CLIP accuracy.

**Experiment** We evaluate the accuracy of all models when center-zooming on IN-A and ON images at 11 different scales $S \in \{128, 160, 192, ..., 448\}$ (Fig. A11). That is, center-zooming at $S$ first resizes the input image so that the smaller dimension becomes $S$ and then takes a $224 \times 224$ center crop (zero-padding is applied when necessary).

**Results** In Fig. A11, we show the changes in the top-1 accuracy (1-crop) when varying the center-zoom scales away from the default ImageNet transform scale ($S = 256$) for both ImageNet-A and ObjectNet. While IN-trained networks exhibit consistent improvement as the zoom scale increases, CLIP shows a monotonic decrease in performance (Fig. A11; yellow curves decreasing on both sides of $S = 256$). This result is surprising but consistent with our hypothesis that CLIP internally performs implicit zooming to reach its peak accuracy and therefore manually zooming (either in or out) at the center mostly ruins its performance.

## B.6   Zoom-in is more useful than zoom-out, which is most important to abstract images

Zooming in enhances texture patterns while zooming out provides a better perspective of the object's shape, which is known to be useful to image classification [12, 19]. Results in Sec. 4.1 and Appendix B.4 indicate that this combined zooming approach can be effective in classifying images from diverse datasets. Here, we test which dataset and model pairs require which type of zoom, and whether zooming in or out is always necessary.

**Experiment** To better understand the effectiveness of each zoom group, we calculate the maximum possible accuracy using all nine locations and different zoom scales $S$ to show per-dataset trends. Additionally, we examined the percentage of images within each dataset that required a specific zoom group to be accurately classified. This analysis allowed us to gain a more comprehensive

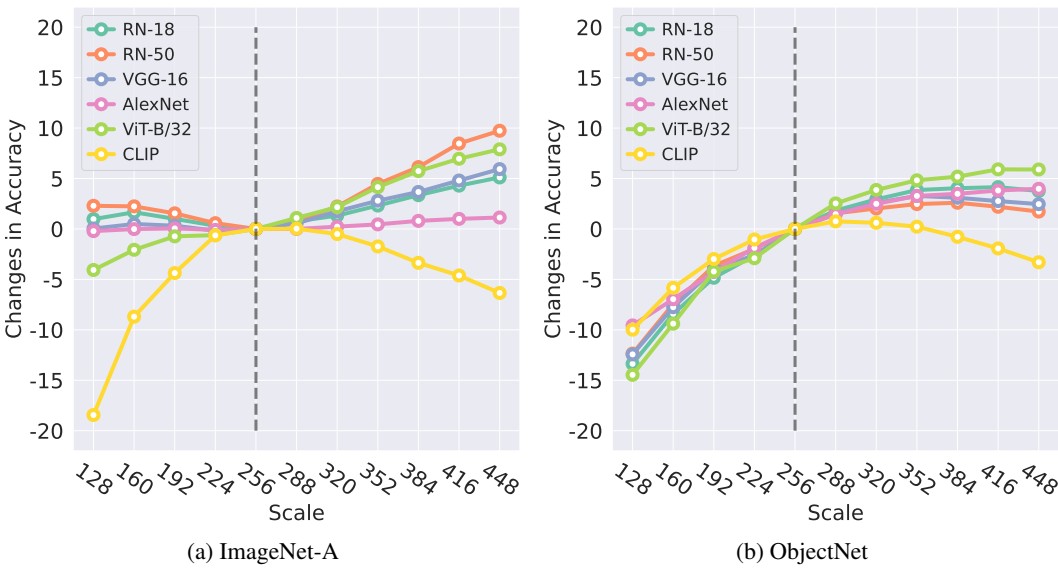

| (a) ImageNet-A | (b) ObjectNet |

Figure A11: Absolute changes in the top-1 accuracy (%) of 6 models on ImageNet-A (a) and ObjectNet (b) when center-zooming images at various scales. Interestingly, center-zooming helps IN-trained networks but hurts CLIP.

understanding of the role that each zoom group played in reaching the maximum possible accuracy reported in Tab. 1.

**Results** The maximum possible accuracy for different zoom scales reveals a clear trend for each dataset. For instance, a slight zoom-**out** enhances accuracy for abstract image datasets like IN-Sketch (Fig. A12a). Conversely, for adversarial image datasets such as IN-A, zooming **in** improves accuracy (Fig. A12b) This pattern is also evident in evaluations using standard 1-crop accuracy (Appendix B.9). Furthermore, the percentage of images that are *exclusively* classifiable with the *zoom-in* group is consistently higher than the other two groups, *i.e.* using ViT-B/32 51.75% on IN-A, and 13.11% on IN-S (Tab. A4a). This shows that most datasets necessitate focusing on the object of interest in the image to both see texture patterns better and reduce background clutter (see Tab. A2 for full results). However, we also find that the *zoom-out* group is also necessary for the correct classification of a small portion of each dataset. For instance, $1.22\% - 2.97\%$ of IN-S images (Tab. A4b) require a *zoom-out* transform to be correctly labeled (*i.e. zoom-in* does not help at all).

Table A4: % of images in the entire dataset that require a particular zoom group to be classified correctly. See Tab. A2 for full results.

|  | *zoom-in* (a) | | *zoom-out* (b) | | *zoom-224* (d) | |
|---|---|---|---|---|---|---|
|  | IN-A | IN-S | IN-A | IN-S | IN-A | IN-S |
| ResNet-18 | 40.67 | 11.83 | 0.92 | 1.77 | 0.19 | 0.36 |
| ResNet-50 | 48.72 | 10.99 | 1.00 | 2.23 | 0.23 | 0.24 |
| ViT-B/32 | 51.75 | 13.11 | 0.85 | 1.83 | 0.15 | 0.36 |
| VGG-16 | 38.24 | 9.47 | 0.93 | 2.97 | 0.29 | 0.28 |
| AlexNet | 26.27 | 11.26 | 1.08 | 1.22 | 0.36 | 0.33 |
| CLIP-ViT-L/14 | 12.01 | 6.64 | 0.44 | 2.38 | 0.05 | 0.12 |

## B.7 Simple aggregation of the zoom transforms can improve accuracy on some datasets but not all

Sec. 4.1 and Appendix B.5 show that using the same feature extractors (even as old as AlexNet), it is possible to achieve higher image classification accuracy if we know where to zoom and at which scale. A practical follow-up question is: How to build a classifier that knows how to zoom given

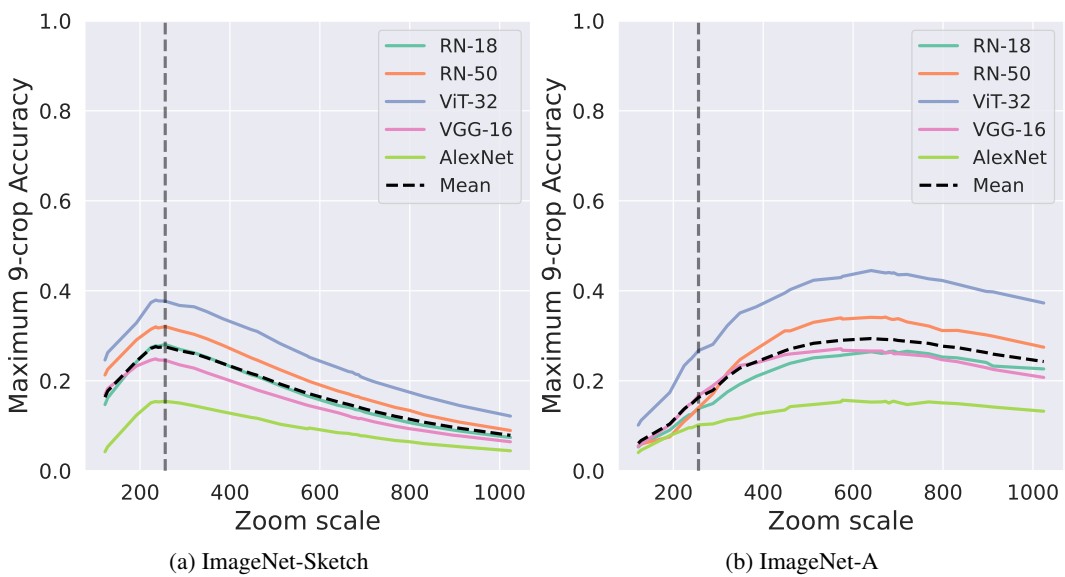

(a) ImageNet-Sketch          (b) ImageNet-A

Figure A12: Maximum possible accuracy using nine crops at varying scales. The vertical line represents the standard ImageNet zoom scale ($S = 256$). While for ImageNet-Sketch (a), zooming out marginally improves the accuracy, for scale factors larger than 256, ImageNet-A (b) exhibits an increase in accuracy. See Appendix B.9 for details.

a test image? In this section, we establish simple baselines that aggregate predictions over a set of zoom transforms.

**Experiment** We employ the mean method from prior work [61, 46], and the max method to aggregate output marginal distributions. For a given image, we get $N$ output distributions over classes from a classifier, in which $N$ is the total number of used transforms. The aggregation process combines these $N$ distributions and outputs a final prediction for the given image. In the aggregation step, we use the mean or max method to infer the final confidence for each class along $N$ distributions. Finally, we select the class that has the highest confidence score. Additionally, we test 5-crop and 10-crop evaluation [38, 63, 23] and compare them with our methods. We use the transforms in the minimum set found for IN-ReaL to evaluate the remaining datasets. The purpose is to reduce the number of augmentations and prevent training on OOD benchmarks.

**Results** max aggregation of zoom-in transforms results in the largest improvements on ImageNet-A. That is, on IN-A, ViT-B/32 reaches a top-1 accuracy of 24.69% (+15.05) (Tabs. A5 and A6) and a ResNet-50 accuracy increases by +13.03 points from 16.62% to 29.65% (Appendix C.3)–a surprisingly strong baseline for future studies. On ObjectNet, max aggregation of zoom-in transforms also yields +1.99 improvement over the 1-crop ViT-B/32 baseline.

On the other hand, mean aggregation results in smaller but more consistent improvements over the 1-crop baseline for many datasets (+3.56 on IN, +4.08 on ReaL, +4.65 on IN-A, and +3.03 on ON; Tab. A5). mean aggregation (Tab. A5b) also outperforms the standard 5-crop and 10-crop [38, 23] aggregation on these four datasets (Tab. A5e–f).

In contrast, for all 6 datasets, aggregating zoom-out and *zoom-224* transforms consistently worsen the performance over the 1-crop baseline (Tab. A5c–d). That is, we find that for a few dozen images (*e.g.* sketches and abstract visuals; Fig. 1ac), interestingly, only zooming out can lead to a correct classification (Appendix B.6), yet for most images in these 6 benchmarks, zooming out hurts the accuracy.

In summary, based on the insights from Sec. 4.1, showing that zooming could help classification, we find that simple methods for aggregating zoom-in transforms at test-time can directly improve model accuracy over the 1-crop and *zoom-224* baselines on four benchmarks, *i.e.* all except IN-R and IN-S, which contain abstract images.

Table A5: Top-1 accuracy (%) of aggregation methods on an IN-trained ViT-B/32 model. Compared to the 1-crop baseline, aggregating zoom-in transforms consistently yields improved accuracy on IN-A, ON but worse accuracy on IN-R and IN-S. *zoom-224* refers to the set of zoom transforms at $S = 224$. See Tab. A6 for more results.

| | (a) | (b) *zoom-in* 🔍 | | (c) *zoom-out* 🔍 | | (d) *zoom-224* | | (e) 5-crop | | (f) 10-crop [38] | |
|---|---|---|---|---|---|---|---|---|---|---|---|
| **Dataset** | 1-crop | max | mean | max | mean | max | mean | max | mean | max | mean |
| IN | 75.75 | 74.35 (-1.40) | **79.31** (+3.56) | 71.48 | 69.47 | 72.66 | 73.67 | 77.33 | 77.73 | 77.30 | 77.87 |
| ReaL | 81.89 | 80.22 (-1.67) | **85.97** (+4.08) | 77.95 | 76.28 | 79.25 | 80.31 | 83.24 | 83.80 | 83.17 | 83.87 |
| IN-A | 9.64 | **24.69** (+15.05) | 14.29 (+4.65) | 7.79 | 5.48 | 8.12 | 7.39 | 12.19 | 9.88 | 12.32 | 9.67 |
| IN-R | 41.29 | 39.90 (-1.39) | 40.06 (-1.23) | 39.05 | 36.21 | 39.52 | 39.28 | 43.90 | 43.17 | **44.31** | 43.28 |
| IN-S | 26.83 | 19.74 (-7.09) | 20.89 (-5.94) | 22.37 | 19.25 | 25.06 | 25.21 | 28.72 | 28.66 | **28.94** | 28.76 |
| ON | 30.89 | 32.88 (+1.99) | **33.92** (+3.03) | 22.56 | 19.51 | 22.75 | 22.72 | 26.96 | 24.98 | 27.14 | 24.97 |

Table A6: Performance of various aggregating methods (%) – The bold numbers show maximum accuracy per model/dataset. CLIP strongly and consistently favors 10-crop over other settings.

| | | (a) | (b) *zoom-in* 🔍 | | (c) *zoom-out* 🔍 | | (d) *zoom-224* | | (e) 5-crop | | (f) 10-crop [38] | |
|---|---|---|---|---|---|---|---|---|---|---|---|---|
| | **Dataset** | **1-crop** | **Max** | **Mean** | **Max** | **Mean** | **Max** | **Mean** | **Max** | **Mean** | **Max** | **Mean** |
| **ResNet-18** | IN | 69.45 | 68.45 (-1.00) | 71.45 (+2.00) | 60.33 | 56.79 | 67.85 | 68.70 | 70.61 | 71.32 | 70.83 | **71.85** |
| | ReaL | 76.94 | 76.33 (-0.61) | 79.94 (+3.00) | 67.64 | 63.92 | 75.73 | 76.74 | 78.26 | 79.01 | 78.42 | 79.46 |
| | IN-A | 1.37 | **11.68** (+10.31) | 5.48 (+4.11) | 2.44 | 2.19 | 3.41 | 2.69 | 3.16 | 2.13 | 3.28 | 1.87 |
| | IN-R | 32.14 | 30.60 (-1.54) | 28.95 (-3.19) | 29.08 | 27.28 | 32.29 | 32.54 | 33.99 | 33.38 | **34.59** | 33.83 |
| | IN-S | 19.41 | 14.86 (-4.55) | 14.34 (-5.07) | 14.48 | 11.49 | 17.80 | 17.83 | 20.83 | 20.70 | **21.39** | 21.06 |
| | ON | 27.59 | **28.21** (+0.62) | 25.92 (-1.67) | 16.11 | 14.10 | 22.82 | 22.86 | 24.77 | 20.91 | 25.47 | 21.03 |
| **ResNet-50** | IN | 75.75 | 73.24 (-2.51) | 77.30 (+1.55) | 69.06 | 66.42 | 74.45 | 75.39 | 76.67 | 77.13 | 76.89 | **77.43** |
| | ReaL | 82.63 | 80.36 (-2.27) | 84.68 (+2.05) | 76.35 | 73.85 | 81.96 | 82.85 | 83.67 | 84.06 | 83.82 | 84.31 |
| | IN-A | 0.21 | **16.11** (+15.9) | 6.23 (+6.02) | 2.79 | 2.19 | 3.04 | 2.11 | 2.28 | 0.95 | 2.43 | 1.00 |
| | IN-R | 35.39 | 33.58 (-1.81) | 32.73 (-2.66) | 35.85 | 33.22 | 36.64 | 36.44 | 37.47 | 36.50 | **38.23** | 36.86 |
| | IN-S | 22.91 | 16.89 (-6.02) | 17.80 (-5.11) | 19.51 | 17.12 | 21.60 | 21.66 | 24.71 | 24.51 | **24.94** | 24.74 |
| | ON | **36.18** | 34.56 (-1.62) | 34.22 (-1.96) | 27.10 | 25.32 | 31.78 | 31.98 | 33.34 | 29.58 | 33.93 | 29.86 |
| **ViT-B/32** | IN | 75.75 | 74.35 (-1.40) | **79.31** (+3.56) | 71.48 | 69.47 | 72.66 | 73.67 | 77.33 | 77.73 | 77.30 | 77.87 |
| | ReaL | 81.89 | 80.22 (-1.67) | **85.97** (+4.08) | 77.95 | 76.28 | 79.25 | 80.31 | 83.24 | 83.80 | 83.17 | 83.87 |
| | IN-A | 9.64 | **24.69** (+15.05) | 14.29 (+4.65) | 7.79 | 5.48 | 8.12 | 7.39 | 12.19 | 9.88 | 12.32 | 9.67 |
| | IN-R | 41.29 | 39.90 (-1.39) | 40.06 (-1.23) | 39.05 | 36.21 | 39.52 | 39.28 | 43.90 | 43.17 | **44.31** | 43.28 |
| | IN-S | 26.83 | 19.74 (-7.09) | 20.89 (-5.94) | 22.37 | 19.25 | 25.06 | 25.21 | 28.72 | 28.66 | **28.94** | 28.76 |
| | ON | 30.89 | 32.88 (+1.99) | **33.92** (+3.03) | 22.56 | 19.51 | 22.75 | 22.72 | 26.96 | 24.98 | 27.14 | 24.97 |
| **VGG-16** | IN | 71.37 | 69.60 (-1.77) | 72.46 (+1.09) | 64.75 | 59.95 | 69.51 | 70.48 | 72.31 | 73.09 | 72.67 | **73.53** |
| | ReaL | 78.90 | 77.23 (-1.67) | 80.59 (+1.69) | 72.55 | 67.68 | 77.48 | 78.58 | 79.80 | 80.42 | 80.13 | **80.80** |
| | IN-A | 2.69 | **11.55** (+8.86) | 6.24 (+3.55) | 3.33 | 2.77 | 4.69 | 3.87 | 4.87 | 3.19 | 5.09 | 3.19 |
| | IN-R | 26.98 | 26.18 (-0.80) | 24.74 (-2.24) | 28.01 | 25.62 | 27.76 | 27.78 | 28.75 | 27.95 | **29.23** | 28.35 |
| | IN-S | 16.78 | 13.30 (-3.48) | 13.05 (-3.73) | 15.18 | 13.37 | 15.82 | 15.97 | 17.80 | 17.63 | **18.28** | 17.92 |
| | ON | **28.32** | 26.96 (-1.36) | 26.15 (-2.17) | 19.88 | 16.42 | 23.47 | 23.60 | 26.21 | 21.65 | 26.52 | 21.80 |
| **AlexNet** | IN | 56.16 | 54.74 (-1.42) | 56.98 (+0.82) | 40.78 | 27.09 | 51.80 | 51.50 | 57.86 | 58.60 | 58.26 | **59.11** |
| | ReaL | 62.67 | 61.46 (-1.21) | 64.35 (+1.68) | 45.84 | 30.58 | 58.25 | 58.16 | 64.53 | 65.39 | 64.98 | **65.94** |
| | IN-A | 1.75 | **4.65** (+2.90) | 3.27 (+1.52) | 1.56 | 1.23 | 2.31 | 1.97 | 2.53 | 2.04 | 2.64 | 2.03 |
| | IN-R | 21.10 | 20.65 (-0.45) | 17.97 (-3.13) | 15.72 | 11.25 | 19.91 | 19.55 | 22.79 | 21.86 | **23.26** | 22.16 |
| | IN-S | 10.05 | 7.94 (-2.11) | 6.54 (-3.51) | 5.82 | 2.72 | 8.29 | 7.39 | 10.84 | 10.65 | **11.20** | 10.80 |
| | ON | 14.23 | **14.91** (+0.68) | 11.80 (-2.43) | 6.11 | 3.75 | 9.65 | 9.01 | 12.63 | 9.57 | 12.84 | 9.58 |
| **CLIP-ViT-L/14** | IN | 75.03 | 70.01 (-5.00) | 74.45 (-0.58) | 72.01 | 72.21 | 74.45 | 76.04 | 76.77 | 76.91 | 76.72 | **77.00** |
| | ReaL | 80.68 | 76.37 (-4.31) | 81.31 (+0.63) | 78.28 | 78.93 | 81.45 | 82.05 | 82.26 | **82.55** | 82.26 | **82.55** |
| | IN-A | 71.28 | 76.57 (+5.29) | 68.16 (-3.12) | 60.71 | 49.51 | 71.69 | 70.04 | 77.80 | 76.61 | **78.25** | 76.83 |
| | IN-R | 87.74 | 84.12 (-3.62) | 83.54 (-4.20) | 86.84 | 86.29 | 88.12 | 88.24 | 89.64 | 89.66 | **90.01** | 89.94 |
| | IN-S | 58.23 | 51.88 (-6.35) | 56.06 (-2.17) | 57.14 | 57.43 | 59.00 | 59.90 | 61.28 | 61.61 | 61.59 | **62.07** |
| | ON | 66.32 | 60.20 (-6.12) | 58.10 (-8.22) | 56.57 | 58.11 | 62.44 | 62.65 | 66.70 | 64.88 | **66.87** | 64.97 |

## B.8 Runtime analysis of MEMO

Another benefit of RRC compared to AugMix is faster inference time. Table A7 shows the runtime analysis of MEMO. Typically, TTA methods suffer from slow runtime due to augmentation and test-time training processes. We find that MEMO + RRC consistently leads to an average $1.6\times$ speed-up compared to MEMO + AugMix (Tab. A7; 0.65s / image vs. 1.15s / image), providing more evidence to support this transformation as a viable option for test-time augmentations.

## B.9 1-crop accuracy with different zoom scales

In this section, we demonstrate the performance of various models when zooming in or out of an image. In other words, we utilize the standard 1-crop ImageNet transform while altering the initial scale of the image.

Table A7: Average runtime per query image (in seconds). Using `RandomResizedCrop` in MEMO speed ups the runtime by an average factor of 1.6×.

| Runtime (in seconds) | IN | IN-A | IN-R | IN-S | ON |
|---|---|---|---|---|---|
| MEMO + AugMix [83] | | | | | |
| ResNet-50 [23] | 1.24 | 1.12 | 1.12 | 1.32 | 1.51 |
| DeepAug+AugMix [26] | 1.19 | 1.07 | 1.12 | 1.23 | 1.55 |
| MoEx+CutMix [40] | 1.15 | 1.16 | 1.11 | 1.31 | 1.53 |
| MEMO + `RRC` (**Ours**) | | | | | |
| ResNet-50 [23] | 0.64 | 0.60 | 0.65 | 0.88 | 1.19 |
| DeepAug+AugMix [26] | 0.62 | 0.62 | 0.64 | 0.87 | 1.18 |
| MoEx+CutMix [40] | 0.65 | 0.62 | 0.66 | 0.88 | 1.19 |

In this section, we are conducting experiments using the following models: AlexNet [38], ConvNext (Base, Large, Small, Tiny) [44], DenseNet-161 [29], EfficientNet-B7 [69], MobileNet (V2, V3 Large) [60, 28], ResNet (50, 101) [23], ResNeXt-50 (32x4d) [80], ShuffleNet V2 x1.0 [47], VGG-19 [63], Vision Transformer (ViT-B/16, ViT-B/32, ViT-L/16, ViT-L/32) [17], and Wide ResNet-50-2 [82].

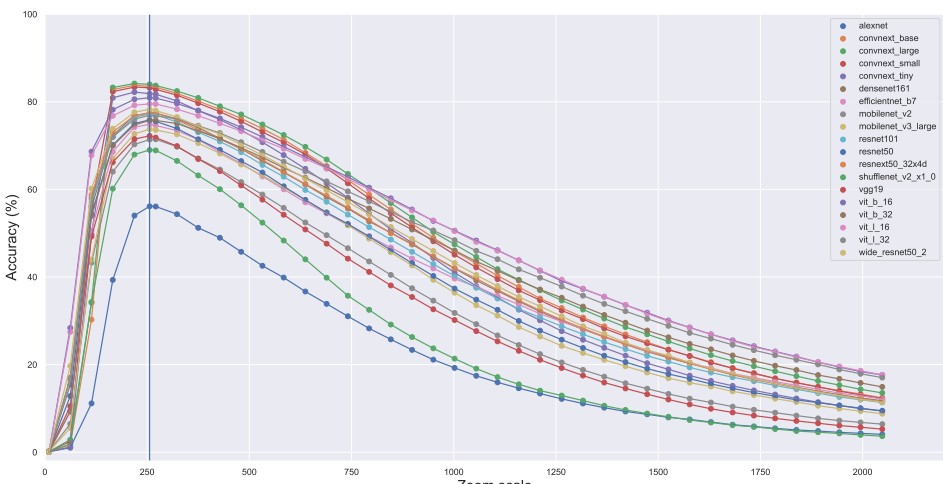

Figure A13: ImageNet accuracy using a 1-crop transform (the vertical line represents the standard ImageNet transform scale factor).

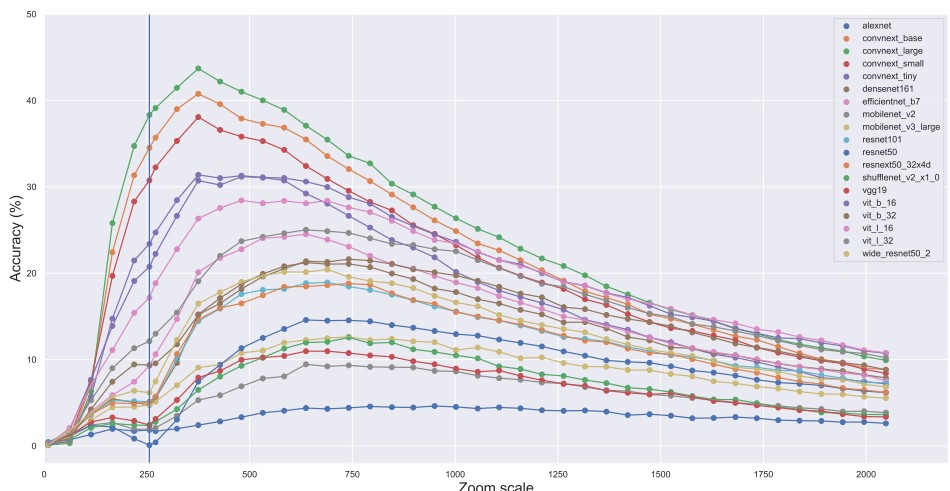

Figure A14: ImageNet-A accuracy using a 1-crop transform (the vertical line represents the standard ImageNet transform scale factor).

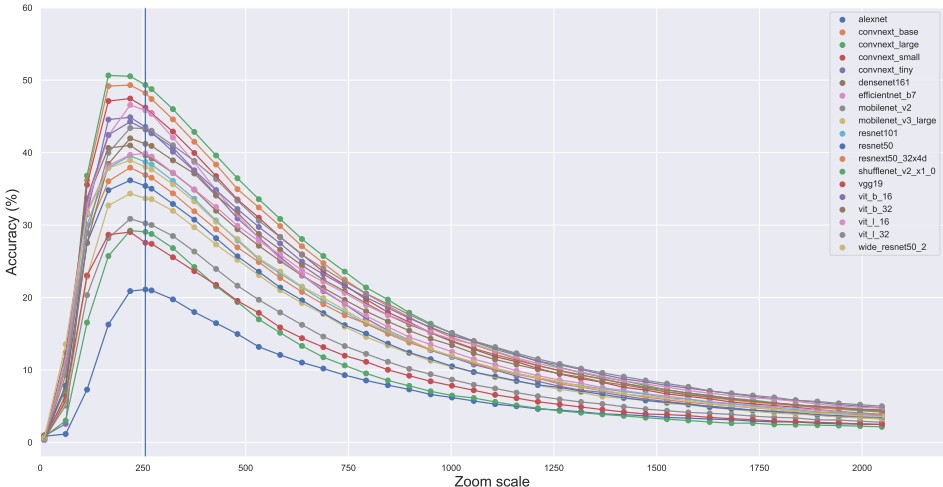

Figure A15: ImageNet-R accuracy using a 1-crop transform (the vertical line represents the standard ImageNet transform scale factor).

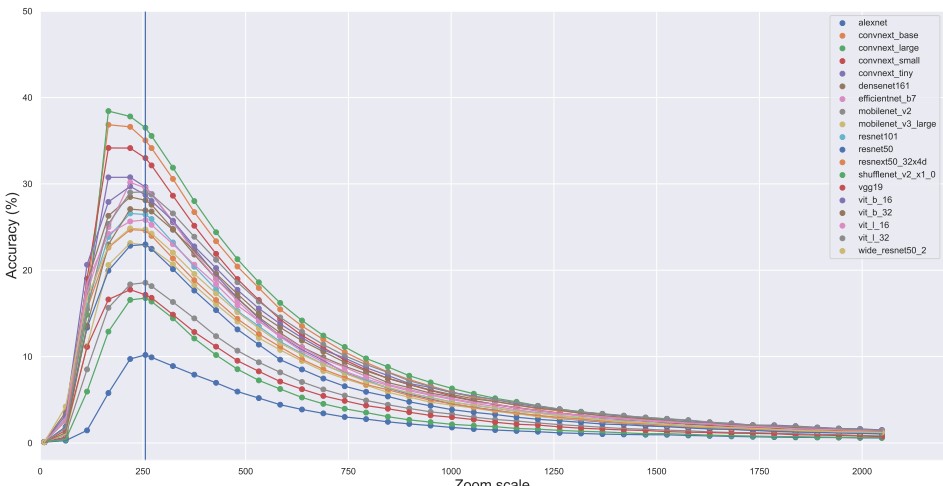

Figure A16: ImageNet-Sketch accuracy using a 1-crop transform (the vertical line represents the standard ImageNet transform scale factor).

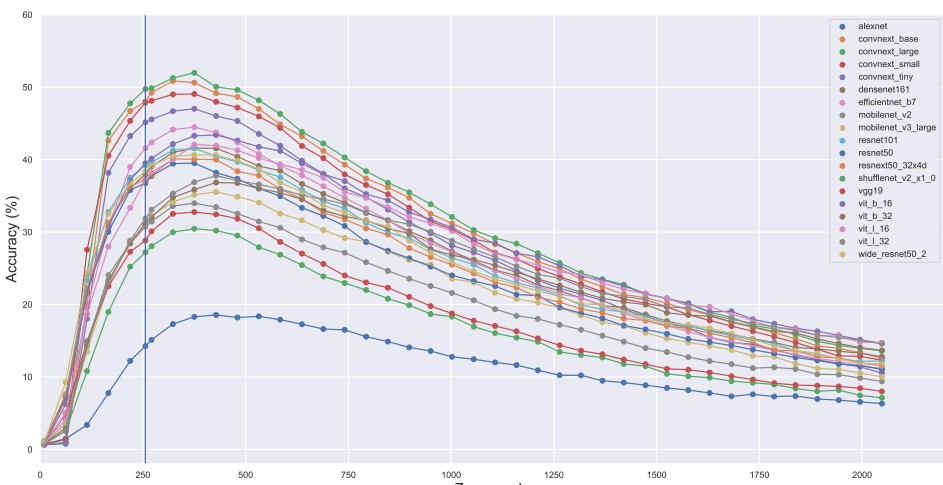

Figure A17: Accuracy using a 1-crop transform on 5K random images of the ObjectNet dataset (the vertical line represents the standard ImageNet transform scale factor).

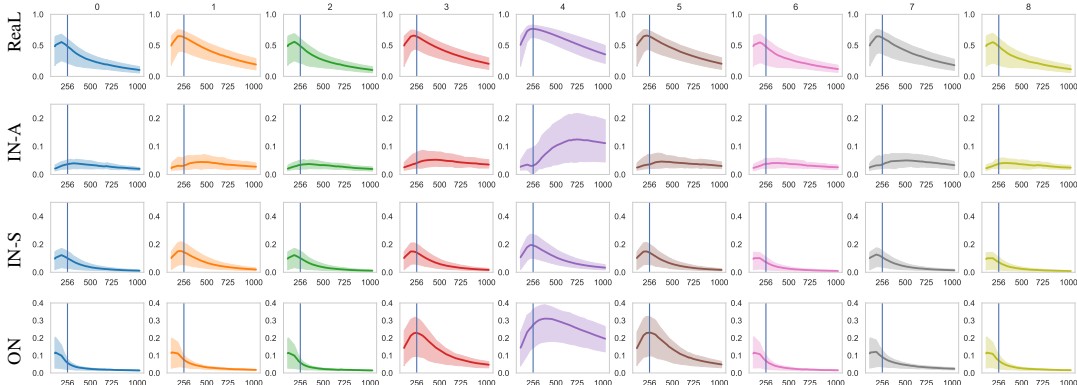

Figure A18: Breakdown of the accuracy of IN-trained models at different crop locations and scale size – Analysis of accuracy across various crop locations and scale sizes reveals that different datasets exhibit distinct optimal conditions. For instance, the IN-A dataset experiences a considerable increase in accuracy when zoomed in, while ImageNet-R yields better results when zoomed out.

## B.10   Distribution of the Top 36 performing transforms.

In this section, we provide more details about the distribution of the top-36 performing transforms. Our results suggest that, on average, 26.65% of all top-36 performing transforms belong to the center at varying scales.

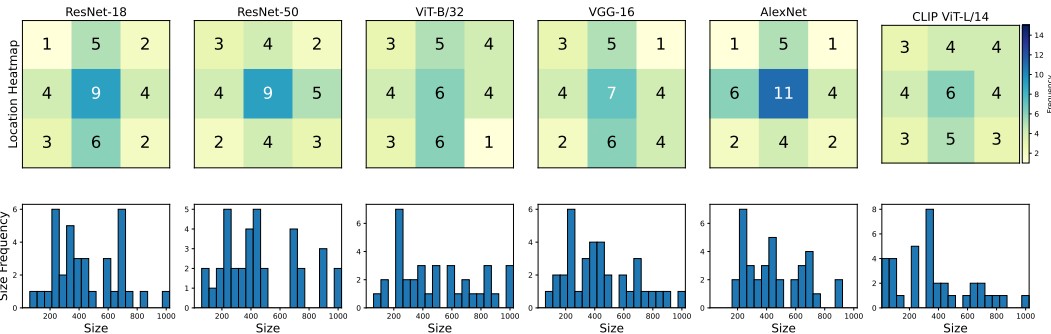

Figure A19: Distribution of Top-36 performing transforms for ImageNet-ReaL

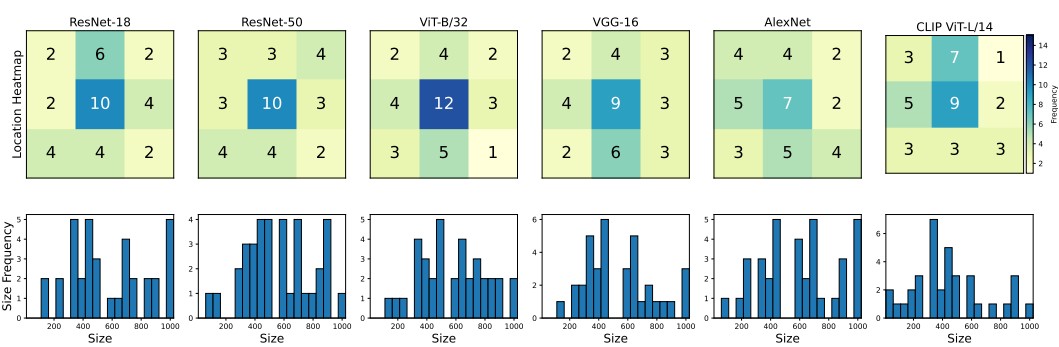

Figure A20: Distribution of Top-36 performing transforms for ImageNet-A

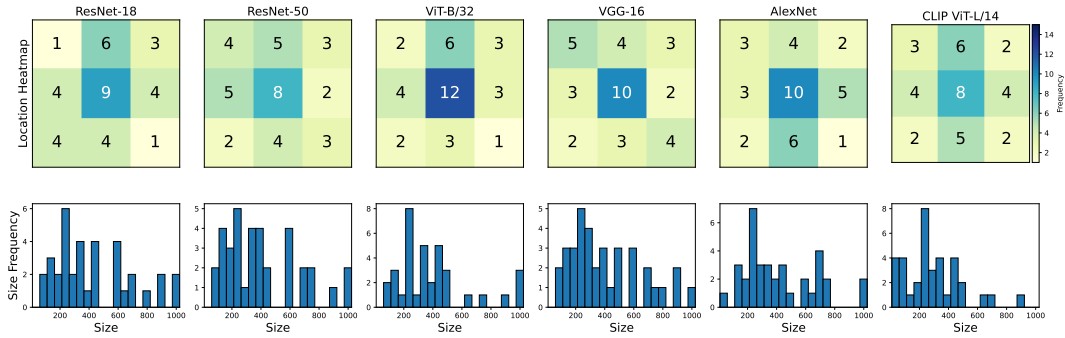

Figure A21: Distribution of Top-36 performing transforms for ImageNet-R

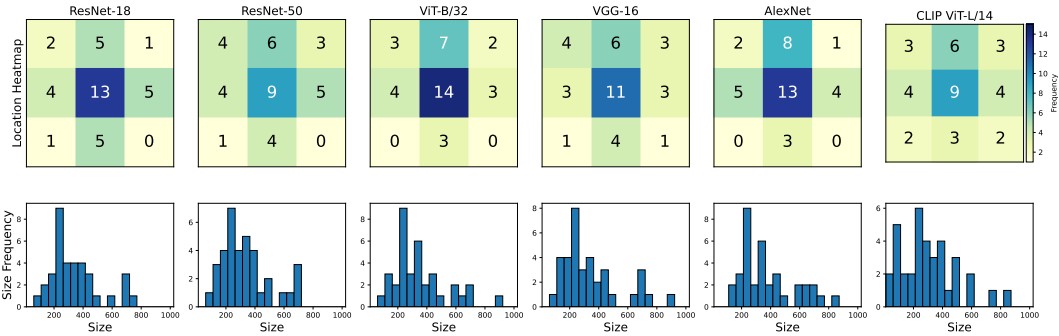

Figure A22: Distribution of Top-36 performing transforms for ImageNet-Sketch

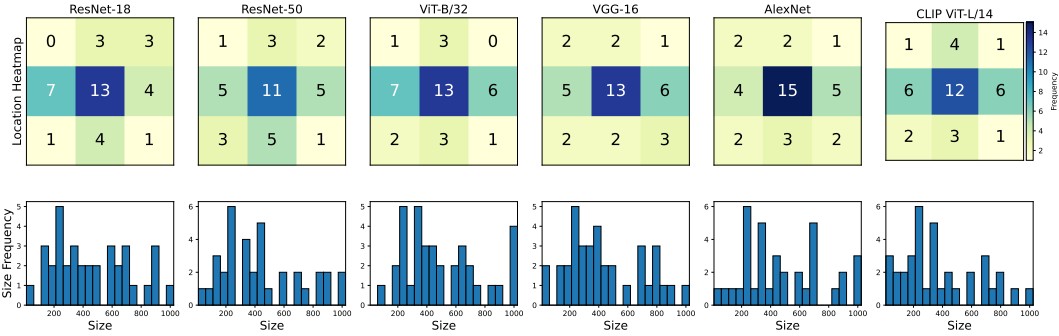

Figure A23: Distribution of Top-36 performing transforms for ObjectNet

## B.11 Background occlusion in ImageNet dataset

Sample images for images with and without occlusion.

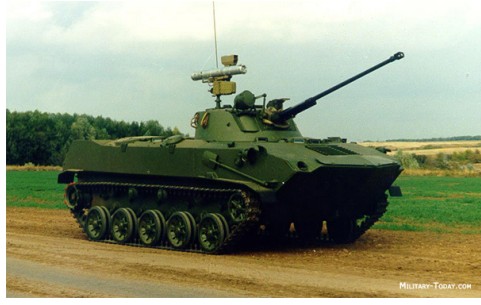

(a) A sample image of the `Tank` class without occlusion.

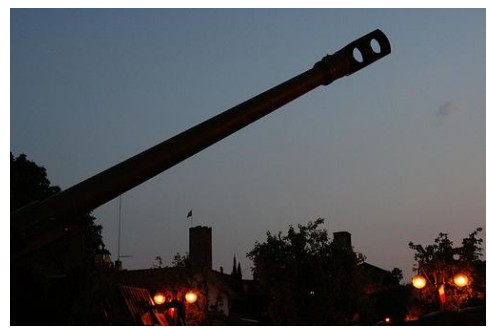

(b) Image with heavy background occlusion.

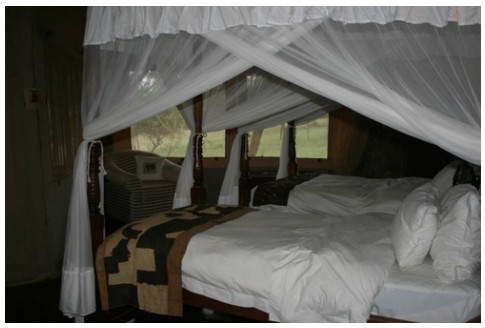

(c) A clean sample image of the `Four Poster` class.

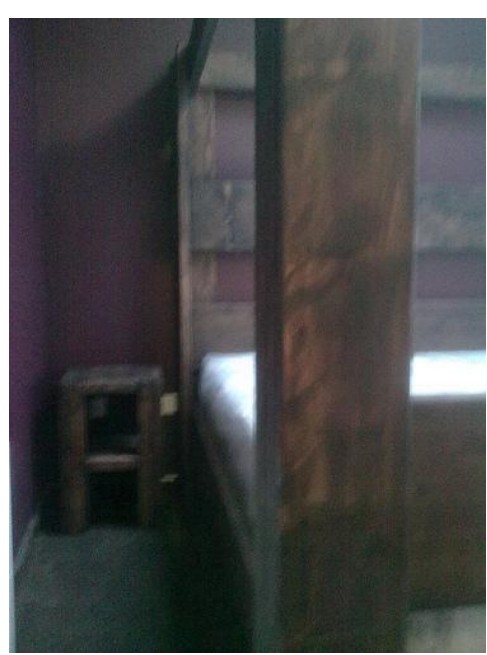

(d) A low-quality image with background occlusion.

Figure A24: Background occlusion examples.

# C    Additional Experiments

In this section, we provide additional experiments with the proposed zoom-based transform.

## C.1    Zooming is similarly important to the foreground and background contents

Background pixels, despite often being neglected in image classification, can contain predictive signals [86, 78, 20, 57]. It has remained largely unknown how much the image context (background) could contribute to the model performance. While Zhu et al. [86] disentangle the predictiveness of background (BG) and foreground (FG) via model training, we directly measure how pretrained models perceive these two signals.

**Experiment**    Using bounding-box annotations provided by Russakovsky et al. [59], we create two dataset variations of ImageNet: *FGSet* and *BGSet*, following Zhu et al. [86]. We mask all the background for *FGSet* as in Fig. A25b, and for *BGSet* we mask all the main objects, as depicted in Fig. A25d & Fig. A25f. After that, we compute the accuracy of these two sets with all tested classifiers using ImageNet and ImageNet-ReaL labels as in Tab. A8.

**Results**    Our results suggest that zooming is important to ImageNet regardless of whether foreground or background features are used, with the difference for *FGSet* and *BGSet* on average being similar (Tab. A8). Additionally, when only the background features were available, almost half of ImageNet images (45.23%) could be correctly classified if optimal Zoom was used. Finally, we found that with only foreground information, ViT-B/32 could achieve a maximum possible accuracy of 95.50% given an optimal zooming method, suggesting that only $98.75\% - 95.50\% = 3.25\%$ of images (Tab. 1) required the background information. These findings suggest that both foreground and background features are important for ImageNet classification, but that an optimal zooming method can considerably improve performance even in the absence of one of these feature sets.

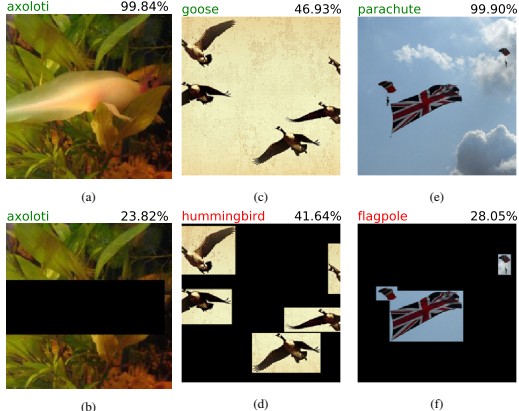

Figure A25: The foreground and the background both contain predictive signals. A ResNet-50 classifier can detect `axolotl` (a), even when the main object is masked (b). Removing the background from images of 'goose' (c) and 'parachute' (e) causes misclassification (d, f).

## C.2    Adversarial datasets contain more objects compared to ImageNet

So far, our findings indicate that if we apply the zoom-in operation to the two datasets of ImageNet-A and ObjectNet, the performance of conventional vision models improves consistently up to a certain threshold (Sec. 4 and Appendix B.5). This suggests that the initial images contain distracting elements that impede the model from correctly identifying the object of interest. Both ImageNet-A and ObjectNet are considered out-of-distribution datasets, which are specifically designed to evaluate a vision model's ability to withstand natural adversarial and pose attacks. We hypothesize that the primary reason that these datasets are hard can be attributed to background clutter, multiple objects, and the presence of a positional bias in these images.

**Experiment**    We use OWL-ViT [49], an open vocabulary object detection model, to quantify the number of objects present in three datasets of ImageNet, ImageNet-A, and ObjecNet. The OWL-ViT expects an input image with a set of object names and will determine if any object instances are present in the image. To specify object names, we use LVIS vocabulary [22], which encompasses a

Table A8: ImageNet classification from object-only and background-only signals. Numbers show the maximum possible top-1 accuracy (%) using zoom-based transforms for minimum set covers in Appendix B.4. We discover that background signals potentially hold significance for image classification. The bold numbers show the highest possible accuracy per dataset and group.

| | 1-crop | | | | Max possible using zooming | | | |
| | FGSet | | BGSet | | FGSet | | BGSet | |
| | IN | ReaL | IN | ReaL | IN | ReaL | IN | ReaL |
|---|---|---|---|---|---|---|---|---|
| ResNet-18 | 59.77 | 64.97 | 4.91 | 7.84 | 89.89(+30.12) | 92.04(+27.07) | 25.81(+20.90) | 31.33(+23.49) |
| ResNet-50 | 68.02 | 72.90 | 6.18 | 9.83 | 93.45(+25.43) | 94.89(+21.99) | 30.30(+24.12) | 35.98(+26.15) |
| ViT-B/32 | 67.46 | 71.78 | **9.72** | 13.38 | 94.40(+26.94) | 95.50(+23.72) | **39.70**(+29.98) | **45.23**(+31.85) |
| VGG-16 | 63.78 | 69.09 | 5.36 | 8.59 | 91.01(+27.23) | 92.91(+23.82) | 26.98(+21.62) | 32.62(+24.03) |
| AlexNet | 42.38 | 46.54 | 3.66 | 5.46 | 80.20(+37.82) | 83.25(+36.71) | 22.02(+18.36) | 27.04(+21.58) |
| CLIP-ViT-L/14 | **74.46** | **78.62** | 9.49 | **13.80** | **96.14**(+21.68) | **97.35**(+18.73) | 36.85(+27.36) | 42.51(+28.71) |
| mean | 62.65 | 67.32 | 6.55 | 9.82 | 90.85 (+28.20) | 92.66 (+25.34) | 30.28 (+23.73) | 35.79 (+25.97) |

comprehensive list of 1203 distinct objects. The OWL-ViT model includes a threshold parameter that reflects its confidence level in its predictions. To assess whether different threshold values would affect our results, we conducted our experiment using both 0.1 and 0.05 as threshold values.

After calculating the distribution of the number of objects in images, we perform a Mann-Whitney U test to determine whether there is a statistically significant difference in this distribution between datasets. As each dataset has a different number of classes, we limited our analysis to shared classes between any two datasets.

**Results** The results of our study reveal a contrast between ImageNet and ImageNet-A, as well as ImageNet and ObjectNet. This finding implies a dissimilarity between the images in the original ImageNet dataset and its OOD datasets that might arise from the presence of background clutter. Specifically, on average, images in ImageNet-A and ObjectNet datasets tend to feature more objects, which can pose more significant distractions for image classification models.

The results of the Mann-Whitney U test also reflect this finding, the p-value for both thresholds was found to be less than 0.05, which is statistically significant at the 95% confidence level (Tab. A9).

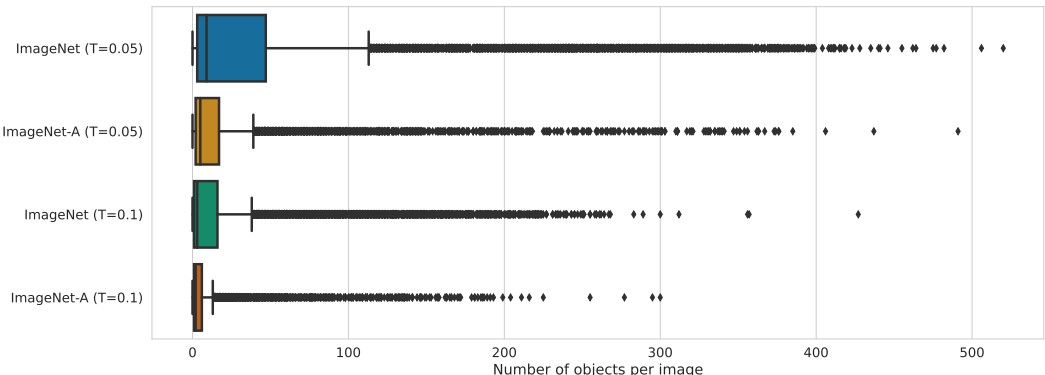

Figure A26: Comparison of the number of objects in two datasets of ImageNet and ImageNet-A using OWL-ViT [49] – $T$ denotes the classification's threshold

### C.2.1 $p$-values for Mann Whitney U test

Table A9: The result of the Mann-Whitney U test to compare ImageNet with ImageNet-A and ObjectNet

| | $T = 0.05$ | $T = 0.01$ |
|---|---|---|
| **ImageNet-A** | 6.27E-265 | 1.71E-235 |
| **ObjectNet** | 1.80E-02 | 3.66E-02 |

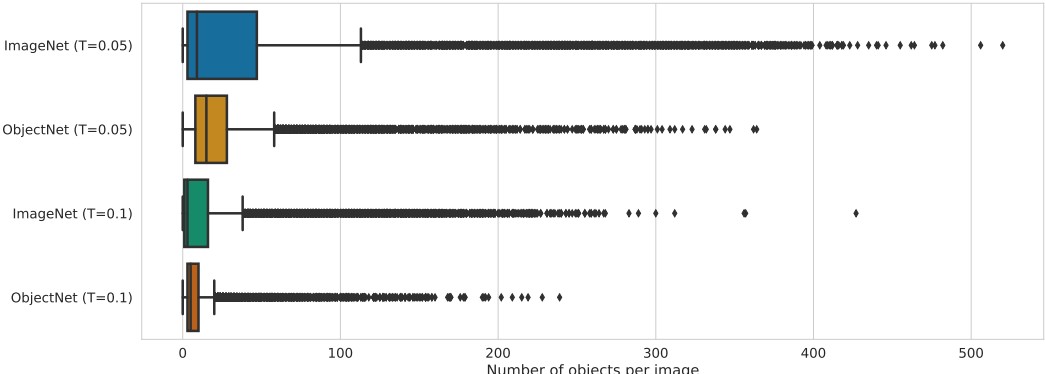

Figure A27: Comparison of the number of objects in two datasets of ImageNet and ObjectNet using OWL-ViT [49] – $T$ denotes the classification's threshold

## C.3  Zooming further improves robustified models on ImageNet-A

Intensive data augmentations have been proven to significantly boost CNNs' performance [77, 66] on ImageNet. Motivated by these previous successes and the fact that neural networks trained on diverse augmentations are able to learn robust representations [41], we want to know if robustified pretrained models (*i.e.* trained with intensive augmentations) could reach higher accuracy on ImageNet-A using zooming in.

**Experiment**   We test 4 different ResNet-50 classifier versions that have been trained with different data augmentation procedures. From the the `torchvision` library, we select two sets of model weights; trained with (V2[3]) and without (V1[4]) data augmentations. We also take two other models trained with DeepAugmentation+AugMix [26] and MoEx+CutMix [40]. The second column in Tab. A10 represents the accuracy of models using 1-crop.

**Results**   Zooming in consistently helps ResNet-50 networks, with improvements varying from +13 to +24 points. The best-performing network is `torchvision`-V2 which uses the max aggregator and achieves 29.65%. These results suggest that simple aggregation over the proposed zoom transform is effective for datasets that have dominant center bias.

Table A10: The results of different aggregation functions on four ResNet-50 variants when tested on ImageNet-A (%). Each model has been trained using different training-time augmentation techniques. Improvements values in parentheses are with respect to the 1-crop baseline.

| ResNet-50 | Baseline | *Max* | *Mean* |
|---|---|---|---|
| `torchvision` V1 | 0.21 | 16.11 (+15.90) | 6.23 |
| MoEx+CutMix [40] | 8.60 | 24.72 (+16.12) | 15.32 |
| DeepAug+AugMix [26] | 3.94 | 27.93 (+23.99) | 13.16 |
| `torchvision` V2 | 16.62 | 29.65 (+13.03) | 22.08 |

---

[3]`ResNet50_Weights.IMAGENET1K_V2`
[4]`ResNet50_Weights.IMAGENET1K_V1`

# D Visualization

In this section, we provide several visualizations of zooming transforms.

## D.1 Visualizations for 36 top performing zoom transforms

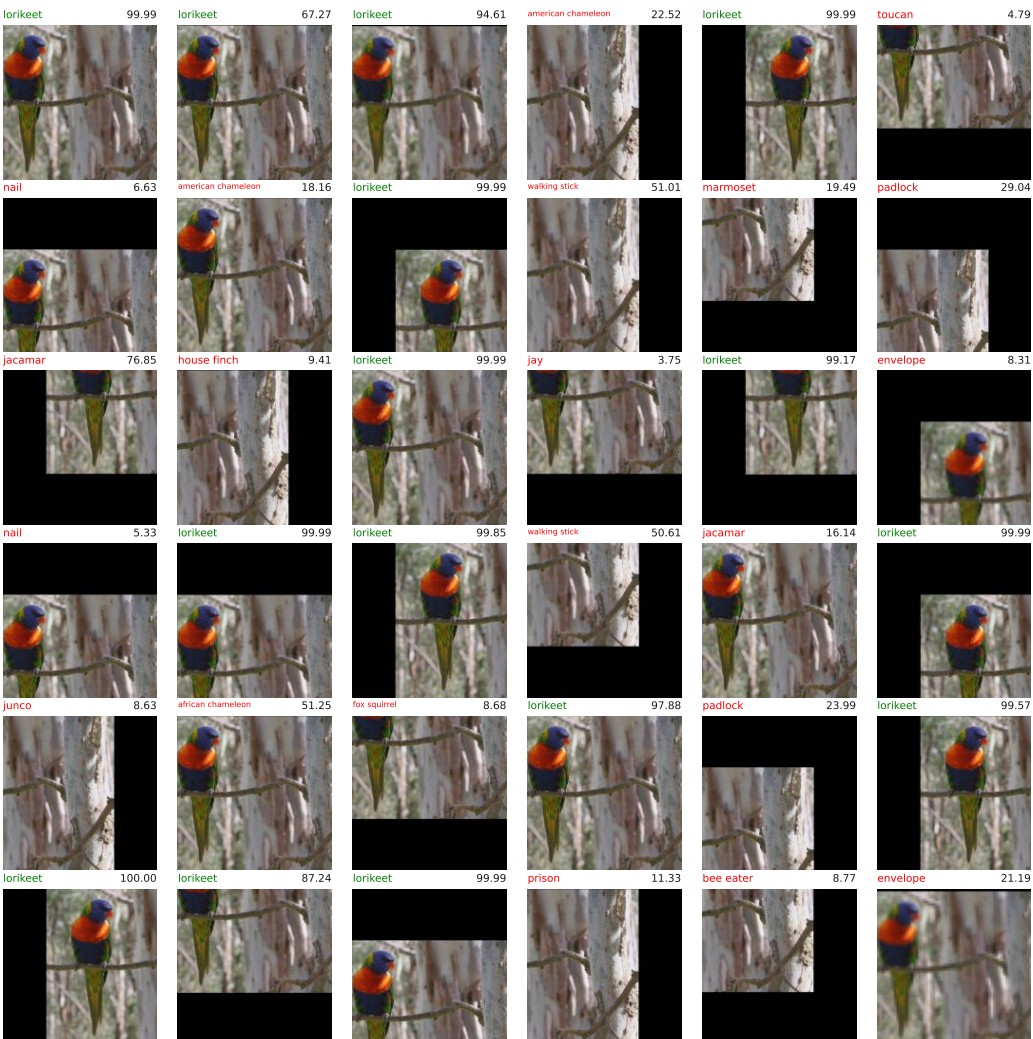

Figure A28: Different framing of an image of a `lorikeet` according to 36 high-performing transforms of a ResNet-50 model

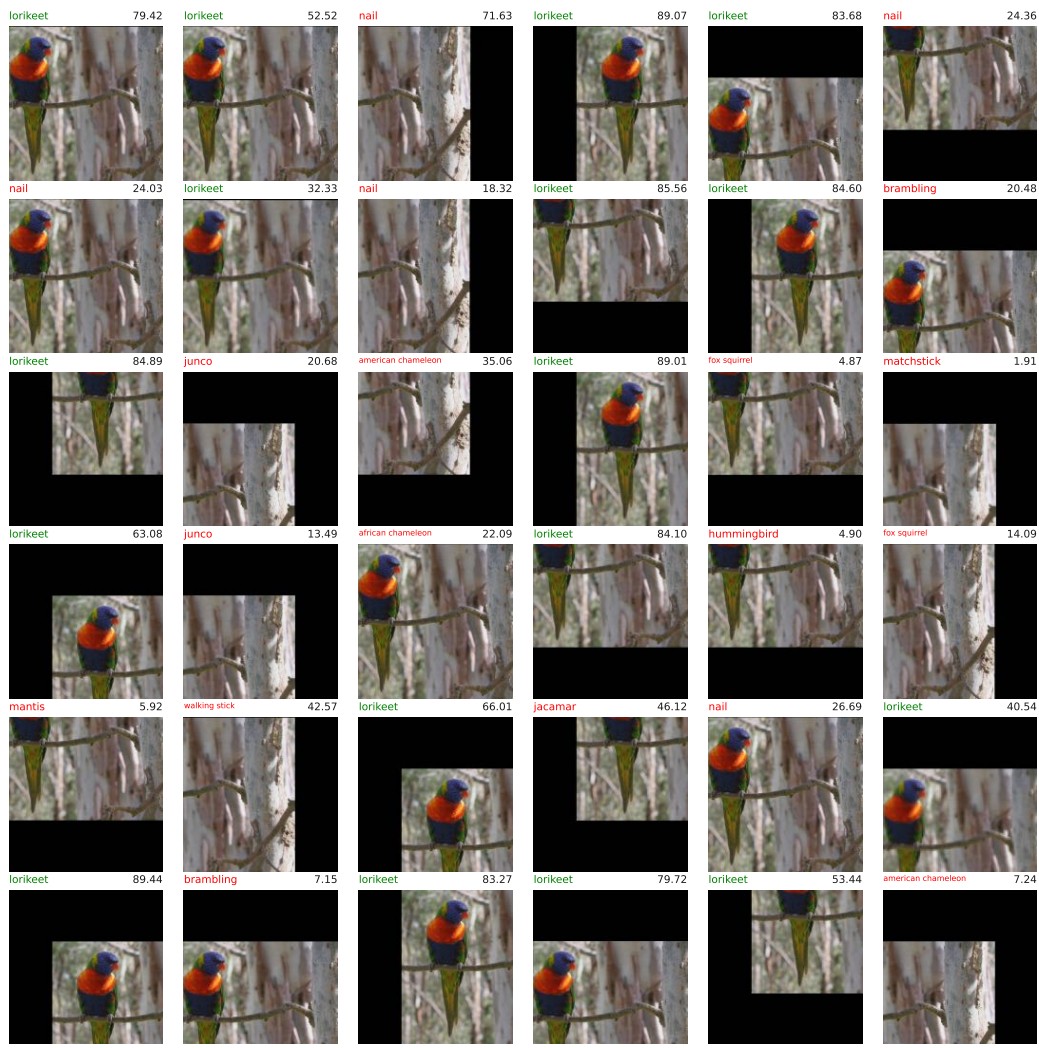

Figure A29: Different framing of an image of a `lorikeet` according to 36 high-performing transforms of a ViT/B-32 model

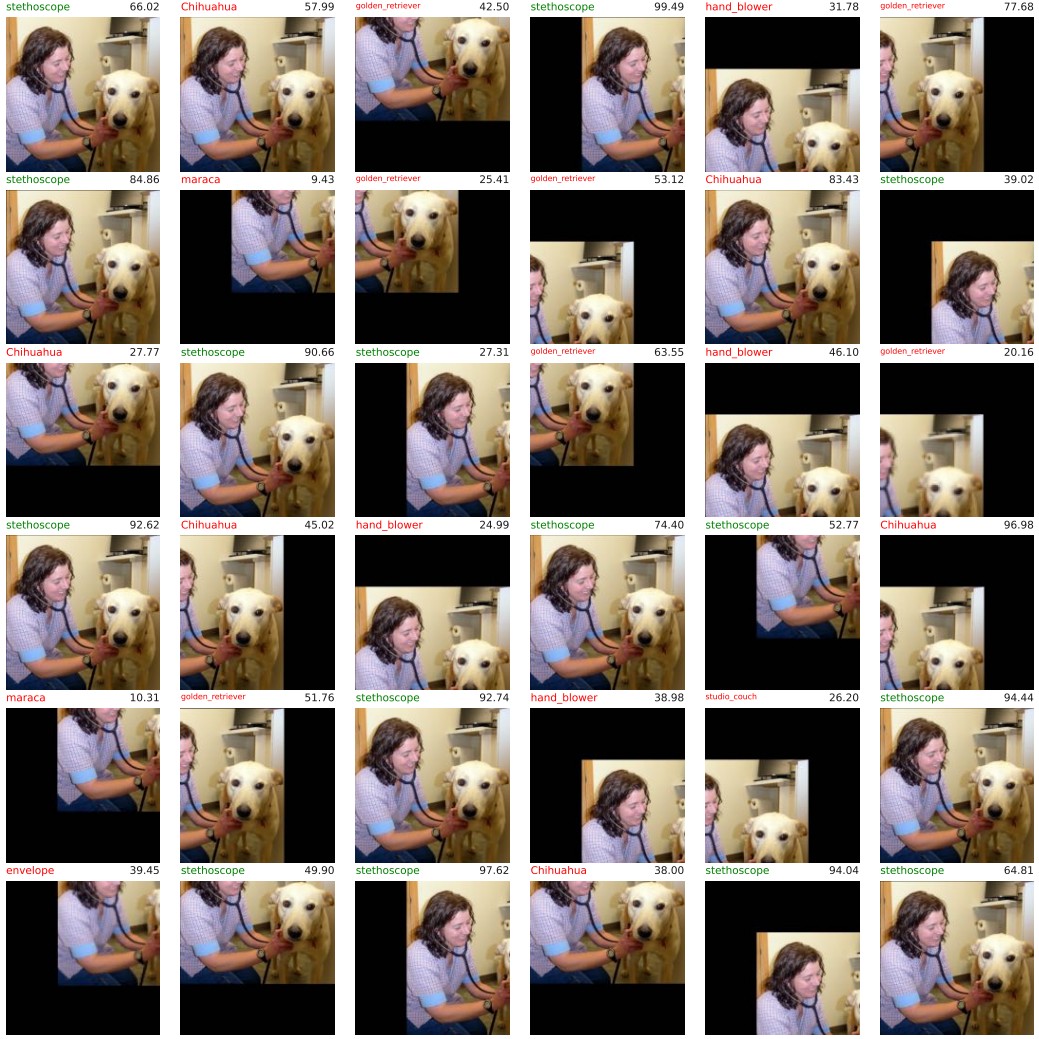

Figure A30: Different framing of an image of a `stethoscope` according to 36 high-performing transforms of a ResNet-50 model

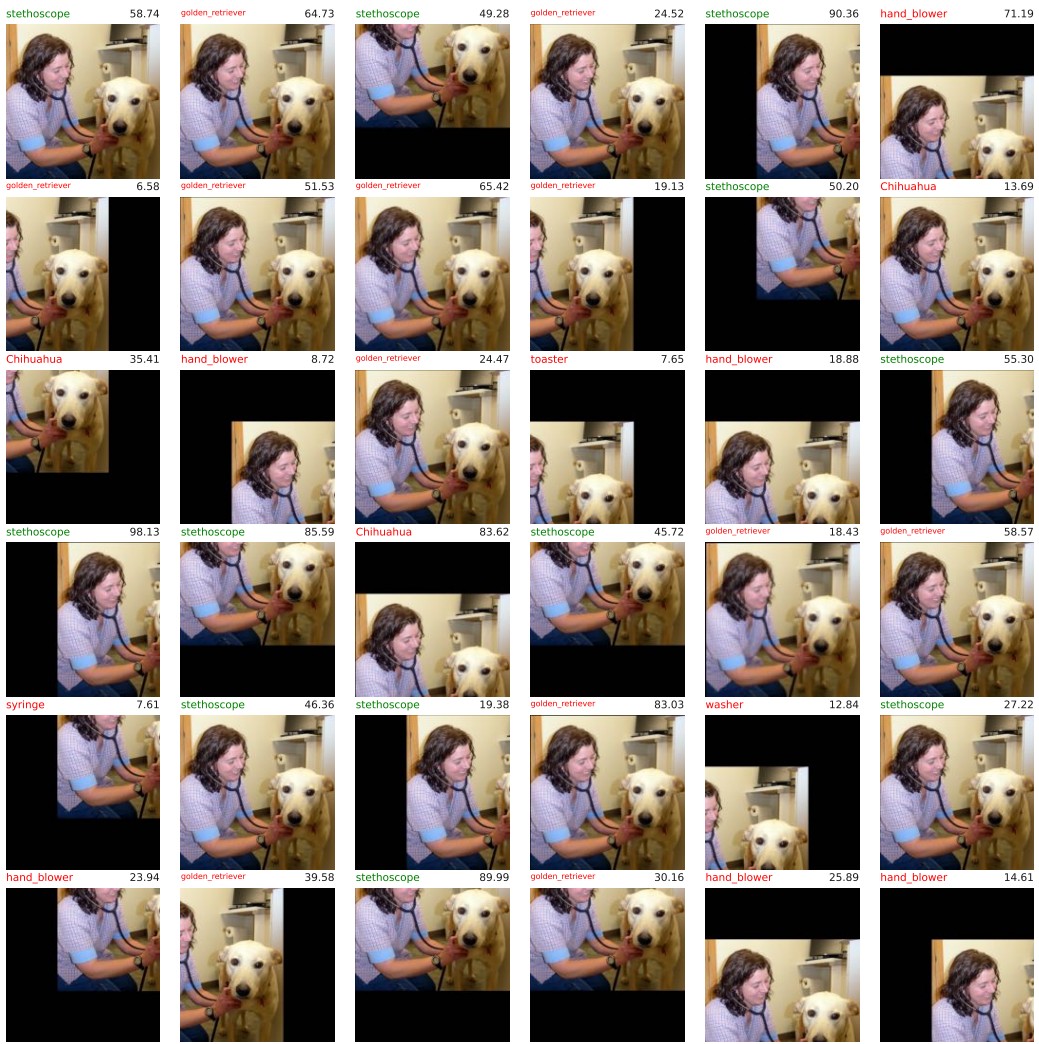

Figure A31: Different framing of an image of a `stethoscope` according to 36 high-performing transforms of a ViT/B-32 model

## D.2 Overview of 324 transforms

The visualizations below illustrate the transforms that result in the correct prediction of the query image, using ViT-B/32 [17] and CLIP-ViT-L/14 [53]. Each circle represents a transform, with the initial zoom scale indicated in the accompanying text. The green circles represent the transformations that lead to correct classification, while the red circles indicate incorrect ones.

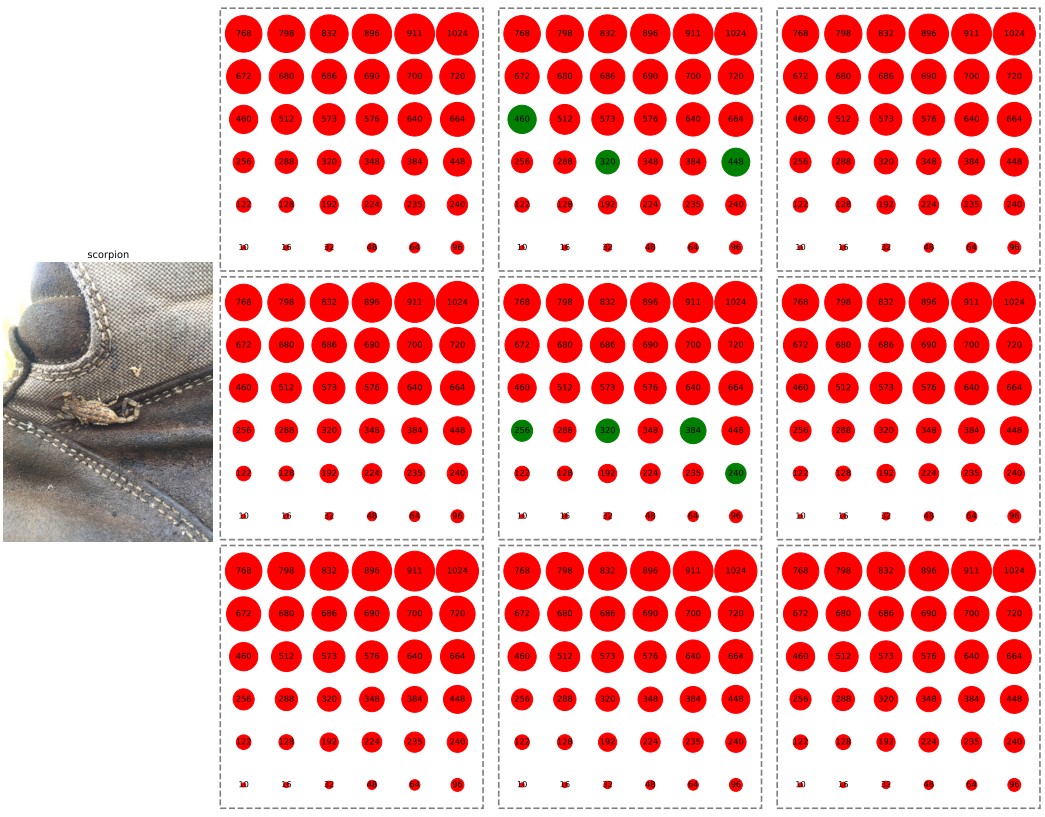

Figure A32: Visualization of effective transforms that lead to the correct classification of an image containing `scorpion`, using a ViT-B/32 model.

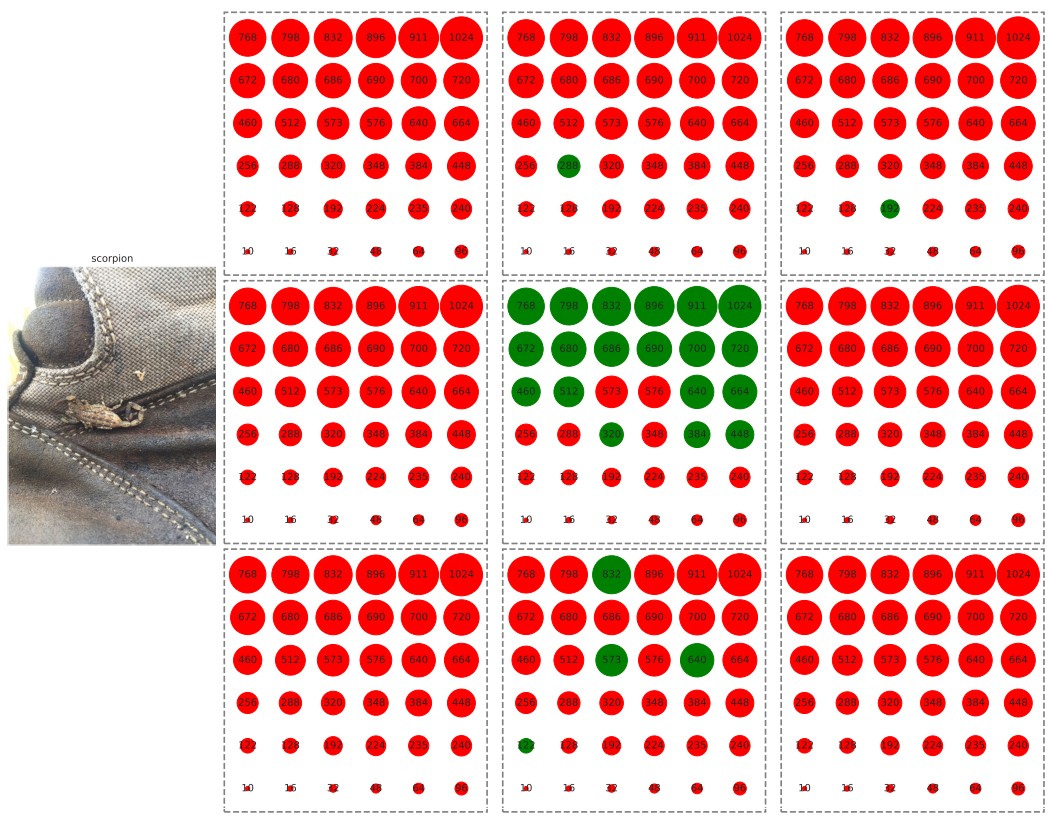

Figure A33: Visualization of effective transforms that lead to the correct classification of an image containing `scorpion`, using a CLIP-ViT-L/14 model.

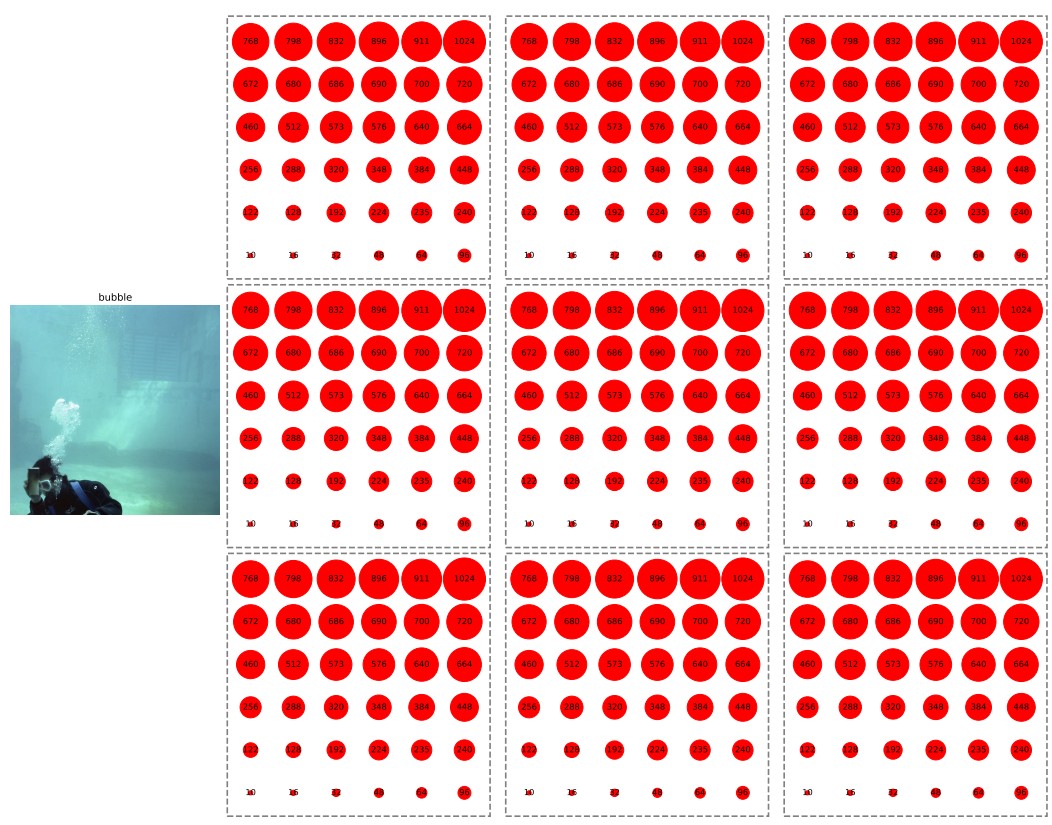

Figure A34: Visualization of effective transforms that lead to the correct classification of an image containing `bubble`, using a ViT-B/32 model.

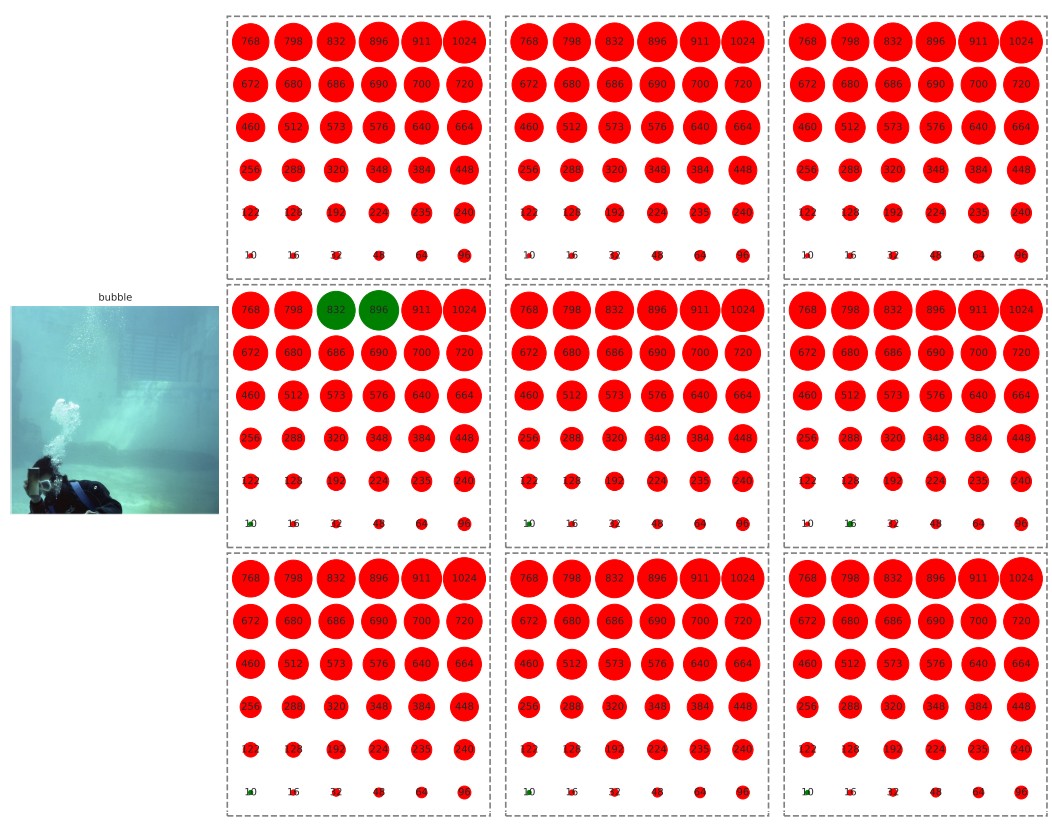

Figure A35: Visualization of effective transforms that lead to the correct classification of an image containing `bubble`, using a CLIP-ViT-L/14 model.

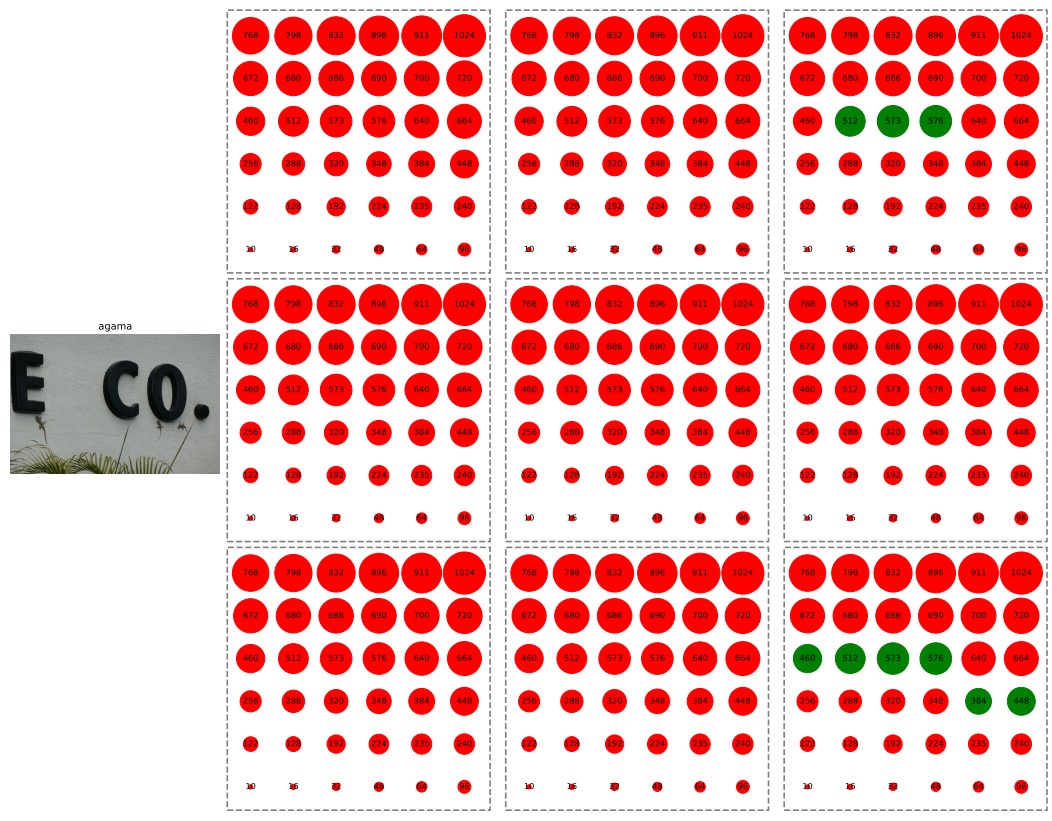

Figure A36: Visualization of effective transforms that lead to the correct classification of an image containing agama, using a ViT-B/32 model.

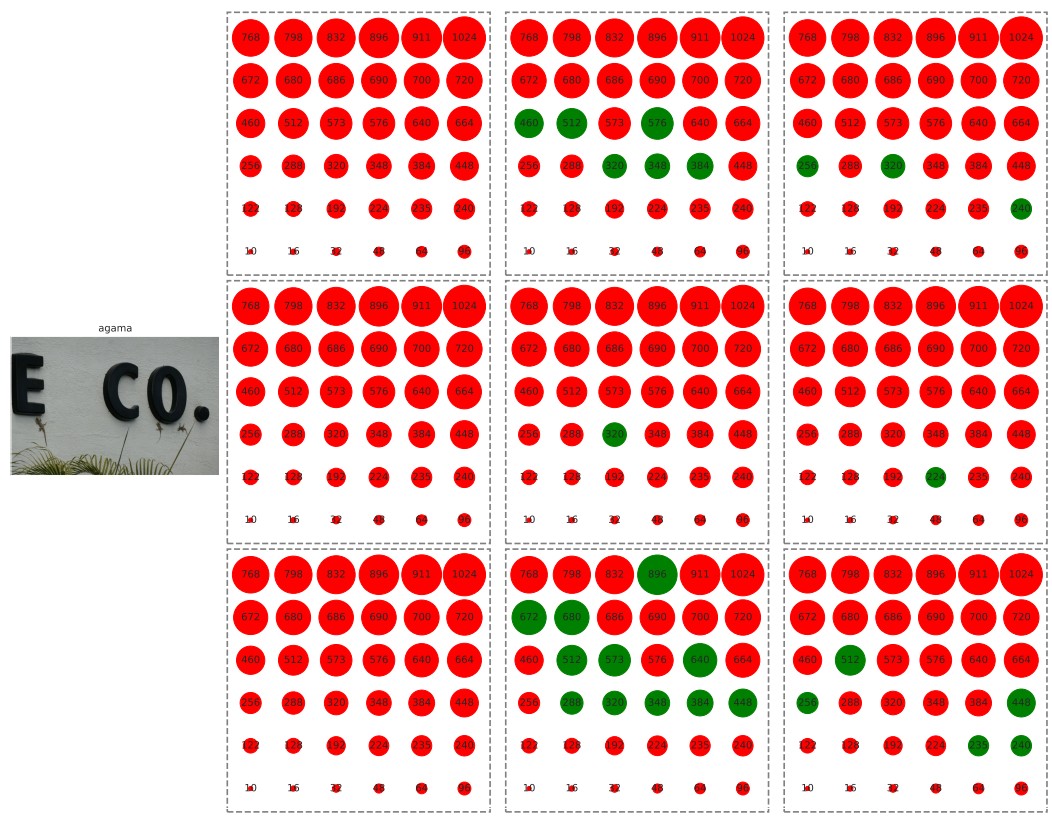

Figure A37: Visualization of effective transforms that lead to the correct classification of an image containing `agama`, using a CLIP-ViT-L/14 model.

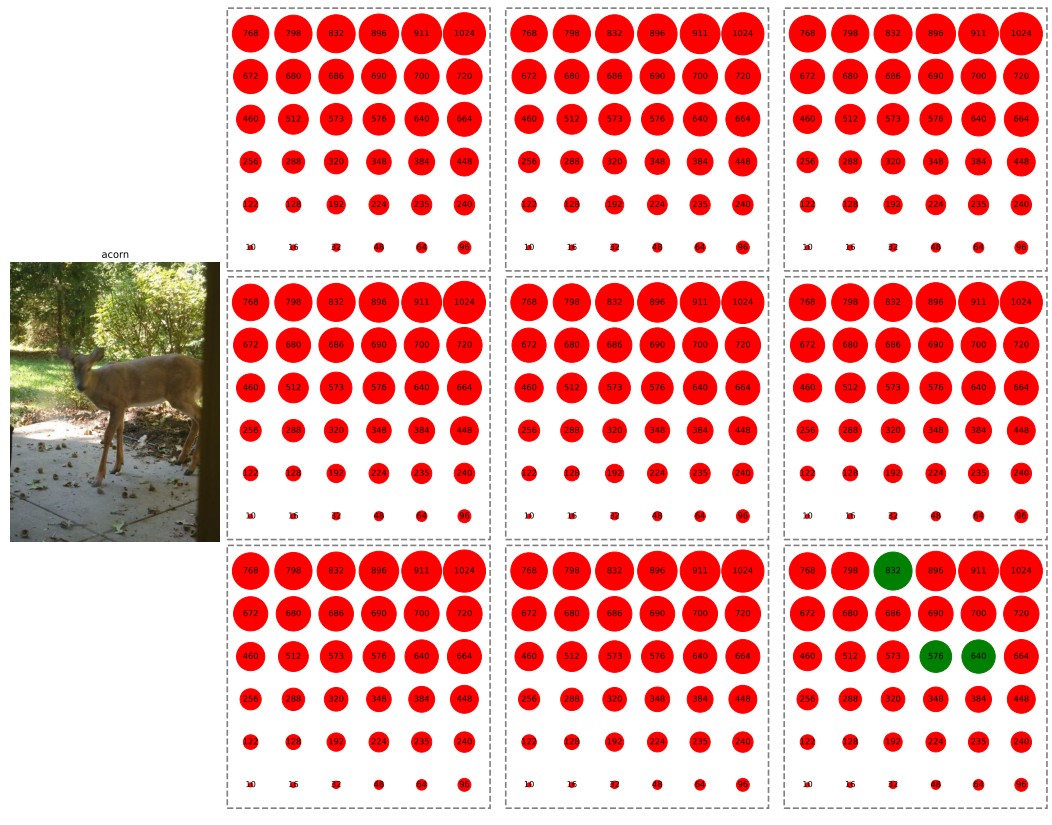

Figure A38: Visualization of effective transforms that lead to the correct classification of an image containing `acorn`, using a ViT-B/32 model.

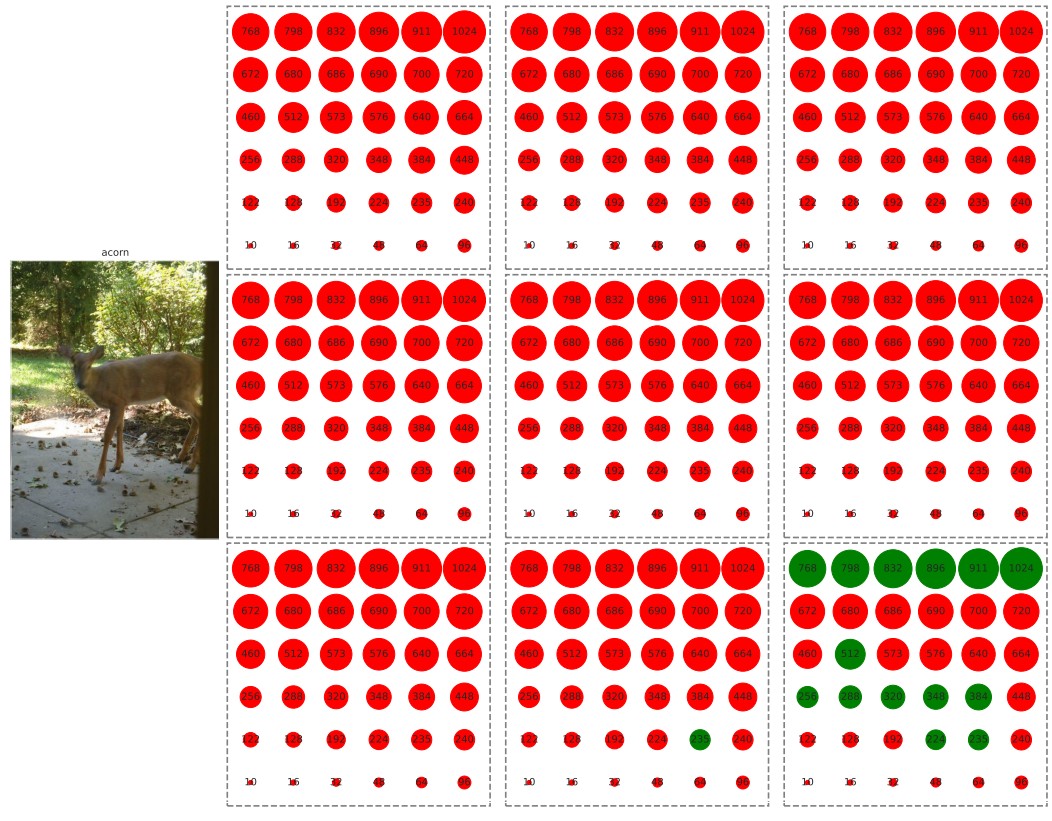

Figure A39: Visualization of effective transforms that lead to the correct classification of an image containing acorn, using a CLIP-ViT-L/14 model.

## D.3 Only *zoom-out* solves

Sample images that required zooming out to be classified correctly.

### D.3.1 ImageNet-Sketch

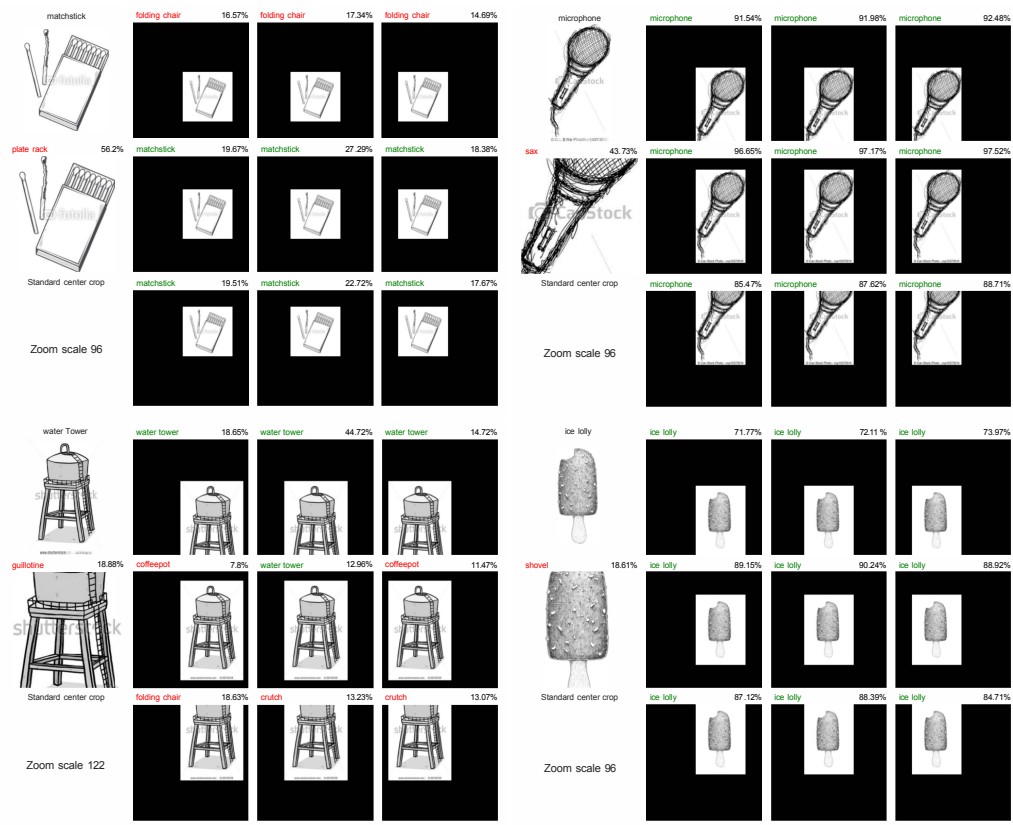

Figure A40: ImageNet-Sketch images that can only be solved using *zoom-out*. Predictions are from a ResNet-50 classifier.

### D.3.2 ImageNet-R

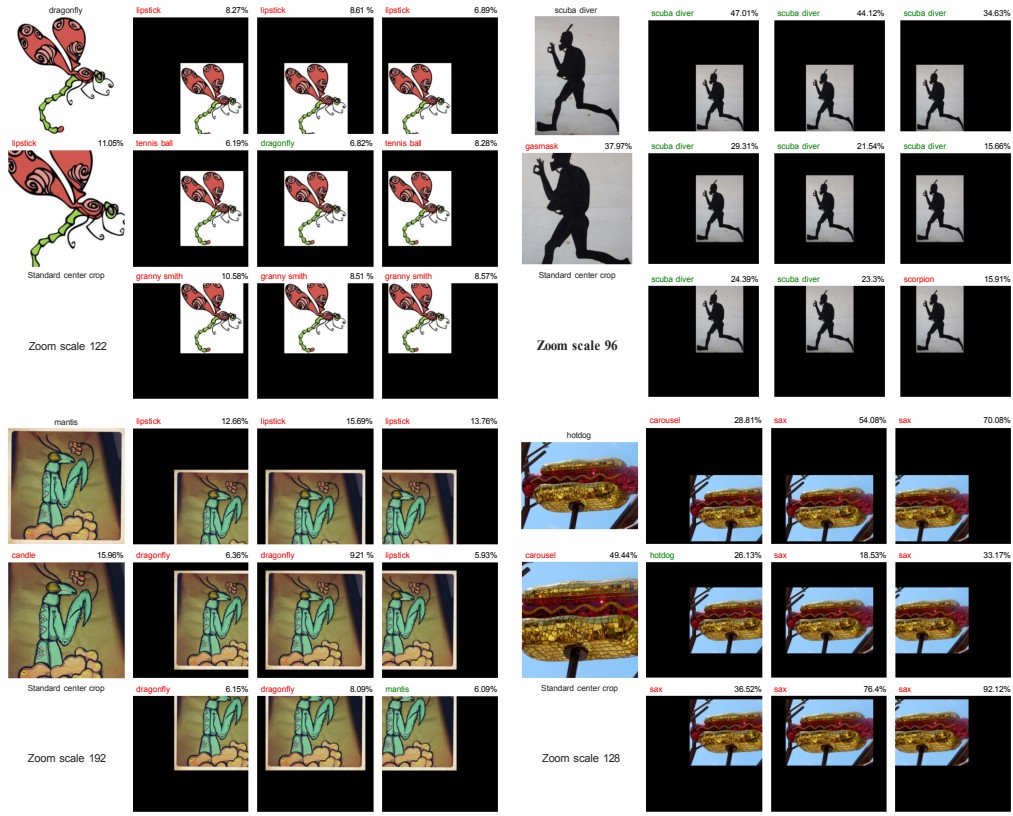

Figure A41: ImageNet-R images that can only be solved using *zoom-out*. Predictions are from a ResNet-50 classifier.

### D.3.3 ObjectNet

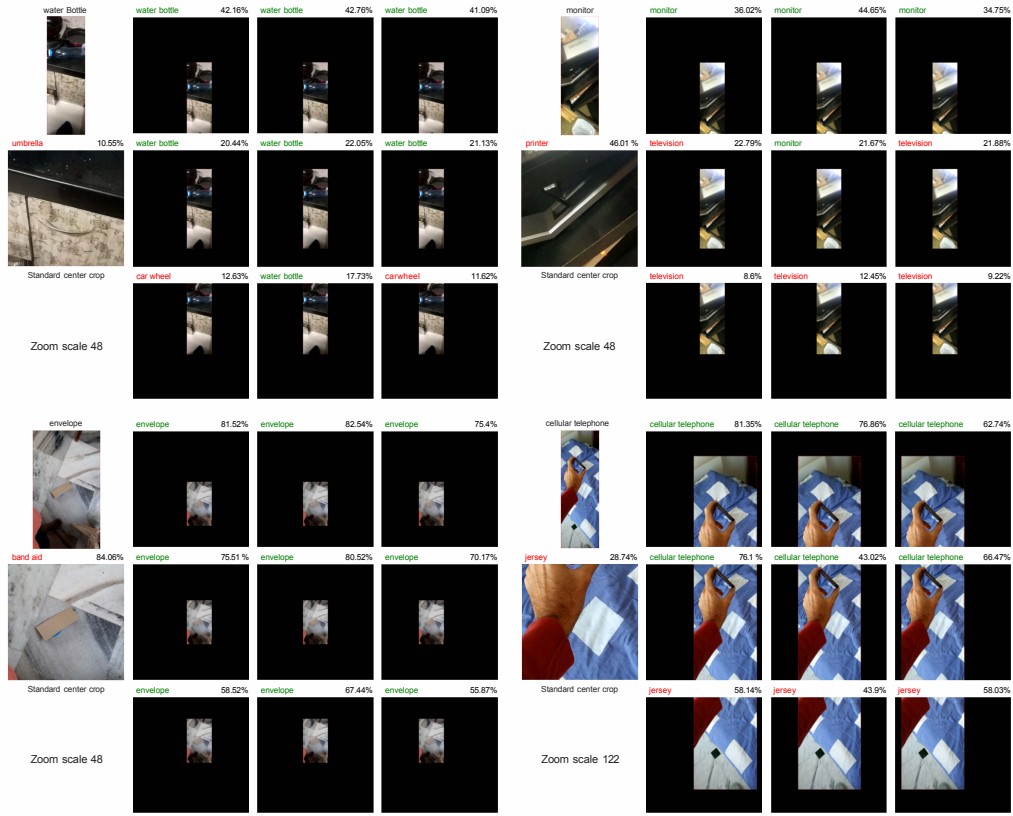

Figure A42: ObjectNet images that can only be solved using *zoom-out*. Predictions are from a ResNet-50 classifier.

## D.4 Only *zoom-in* solves

Sample images that required zooming in to be classified correctly.

### D.4.1 ObjectNet

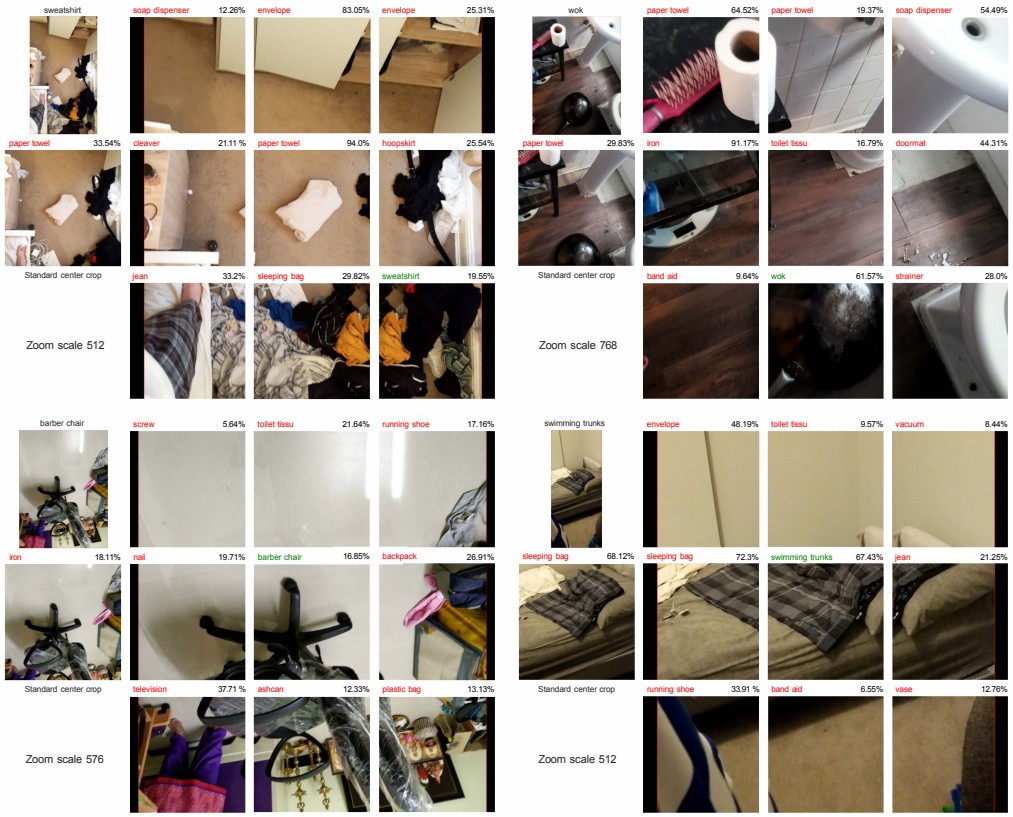

Figure A43: ObjectNet images that can only be solved using *zoom-in*. Predictions are from a ResNet-50 classifier.

## D.5 AugMix and `RandomResizedCrop`

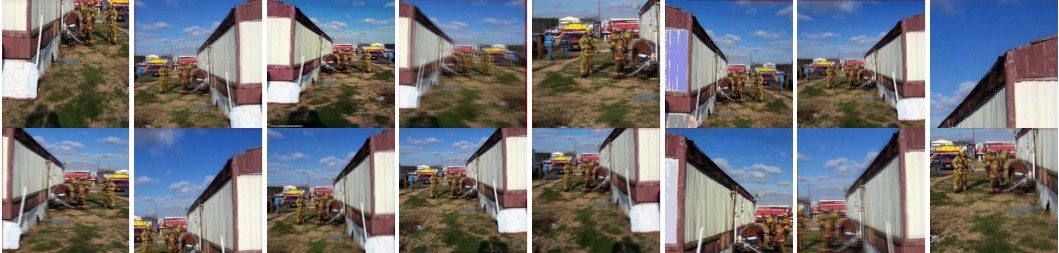

Figure A44: $K = 16$ sample outputs from AugMix [25] (which yields the results of random sampling from 13 transformations that include both spatial and color distortions).

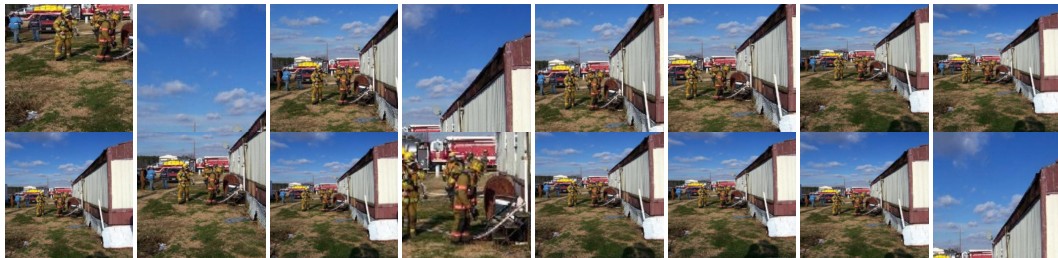

Figure A45: $K = 16$ sample outputs from `RandomResizedCrop` (RRC), which basically randomly zooms into an arbitrary region in the input image.

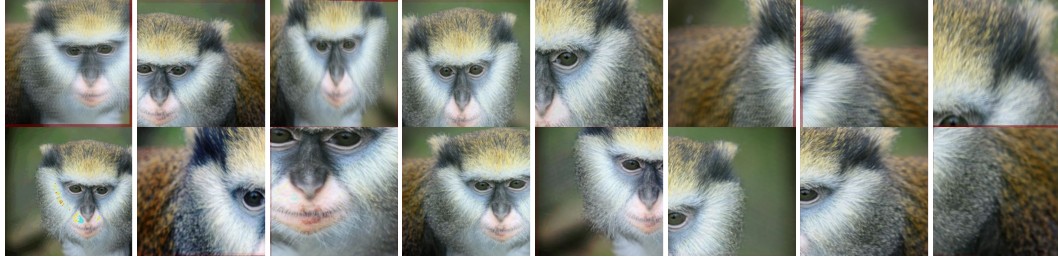

Figure A46: $K = 16$ sample outputs from AugMix [25] (which yields the results of random sampling from 13 transformations that include both spatial and color distortions).

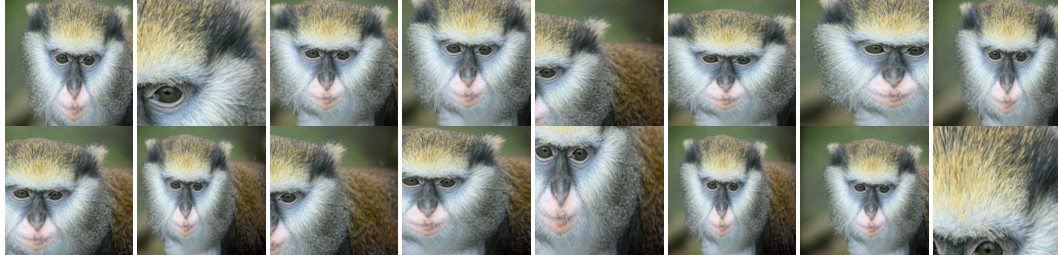

Figure A47: $K = 16$ sample outputs from `RandomResizedCrop` (RRC), which basically randomly zooms into an arbitrary region in the input image.

# E   ImageNet-Hard

In this section, we provide details about the ImageNet-hard dataset.

## E.1   Distribution

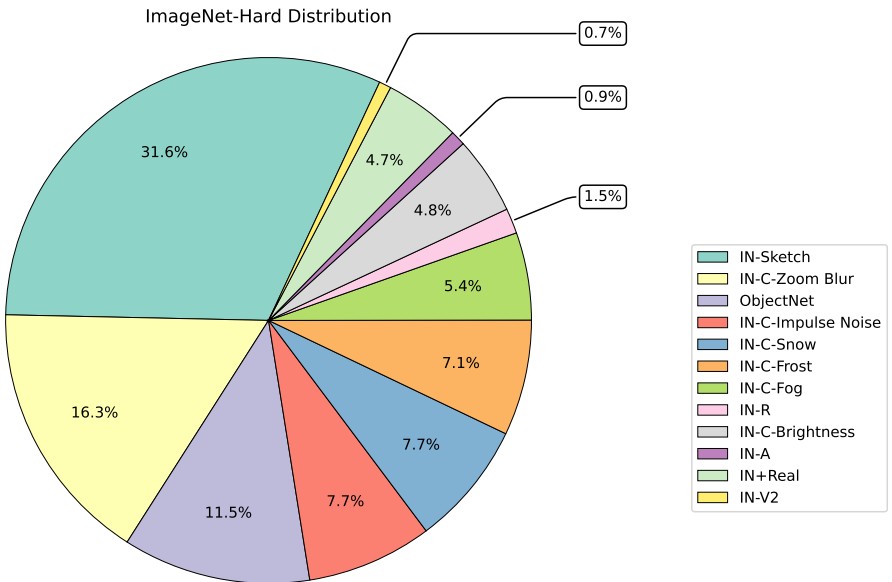

Figure A48: The distribution of the dataset within the ImageNet-Hard Dataset.

## E.2   Samples images

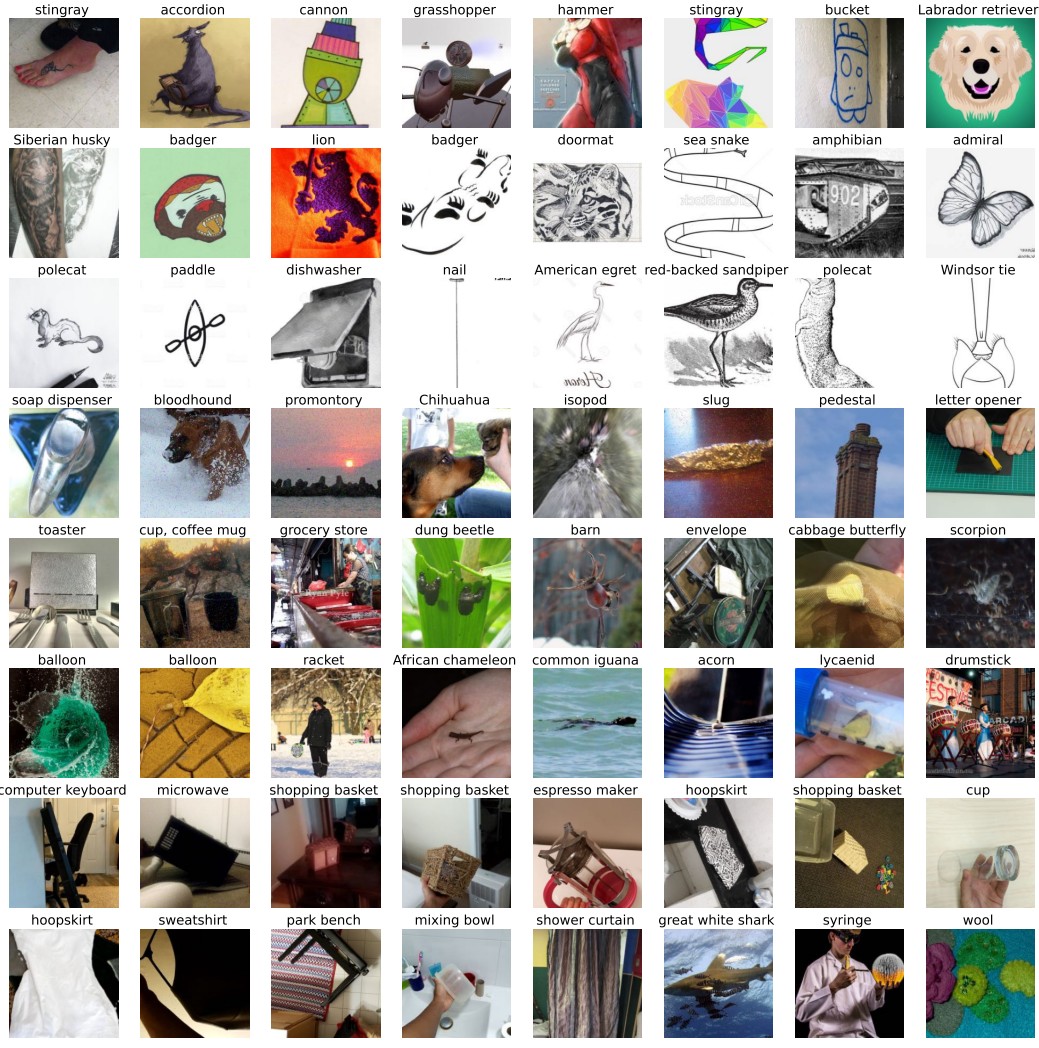

Figure A49: Sample images from ImageNet-Hard dataset with groundtruth labels.

## E.3 Analysis of wrong predictions

We used `gpt-3.5-turbo` to categorize each misprediction made by EfficientNet-L2 into two classes: plausible and implausible, based on the semantic distance between the groundtruth label and the predicted label.

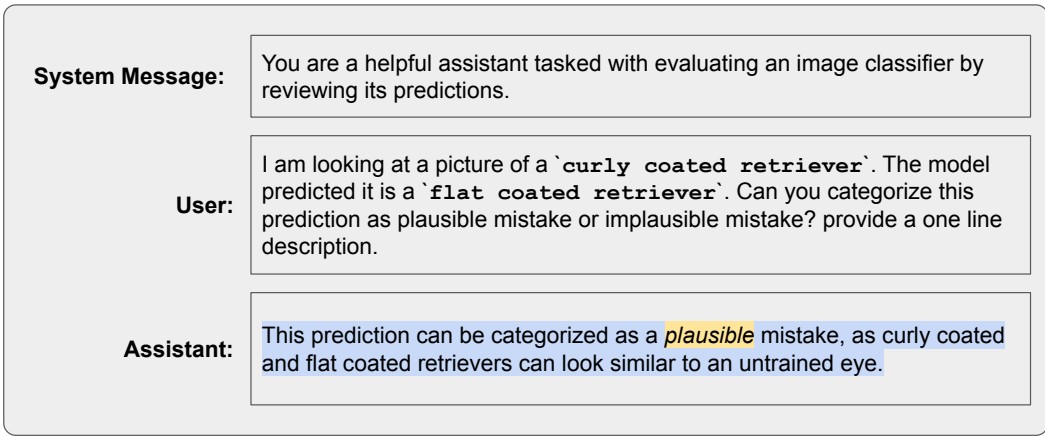

Figure A50: Sample prompt and response of `gpt-3.5-turbo` for a plausible classification. The text in the **Assistant** block is the generated response.

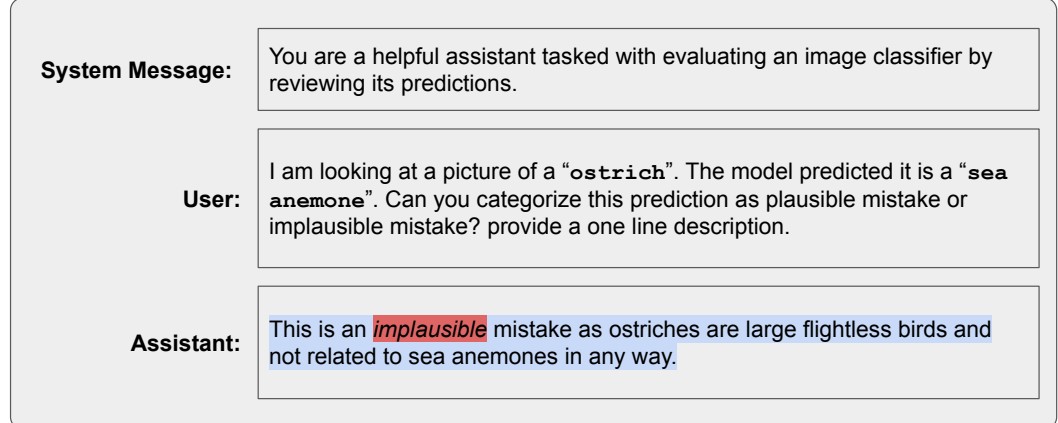

Figure A51: Sample prompt and response of `gpt-3.5-turbo` for an implausible classification. The text in the **Assistant** block is the generated response.

### E.4 Confusing classes

In this section, we present a selection of examples highlighting the errors made by our highest-performing model, EfficientNet-L2.

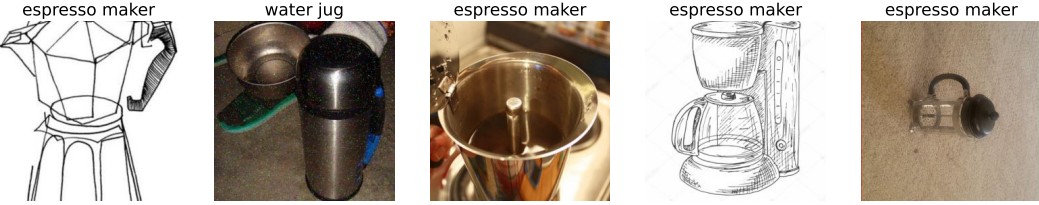

Figure A52: Images misclassified into `coffemaker` by EfficientNet-L2

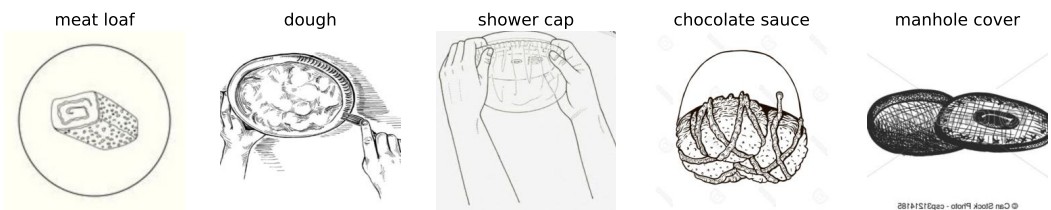

Figure A53: Images misclassified into `strainer` by EfficientNet-L2

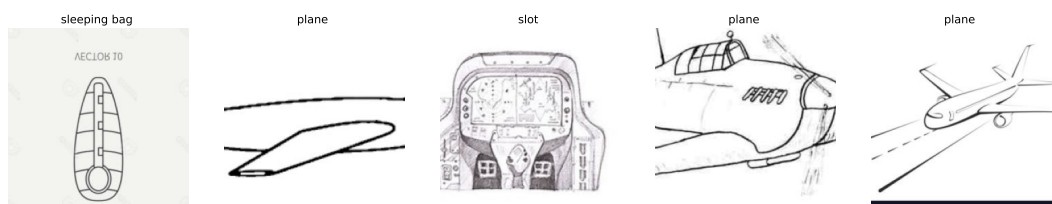

Figure A54: Images misclassified into `space shuttle` by EfficientNet-L2

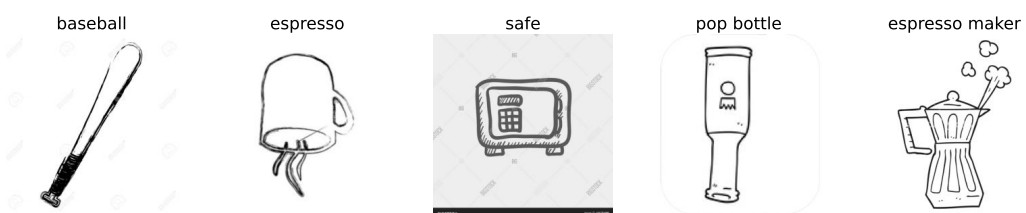

Figure A55: Images misclassified into `safety pin` by EfficientNet-L2

## E.5 Common and rare misclassification

This section shows some sample misclassification for EfficientNet-L2 and OpenCLIP's ViT-bigG-14 classifiers.

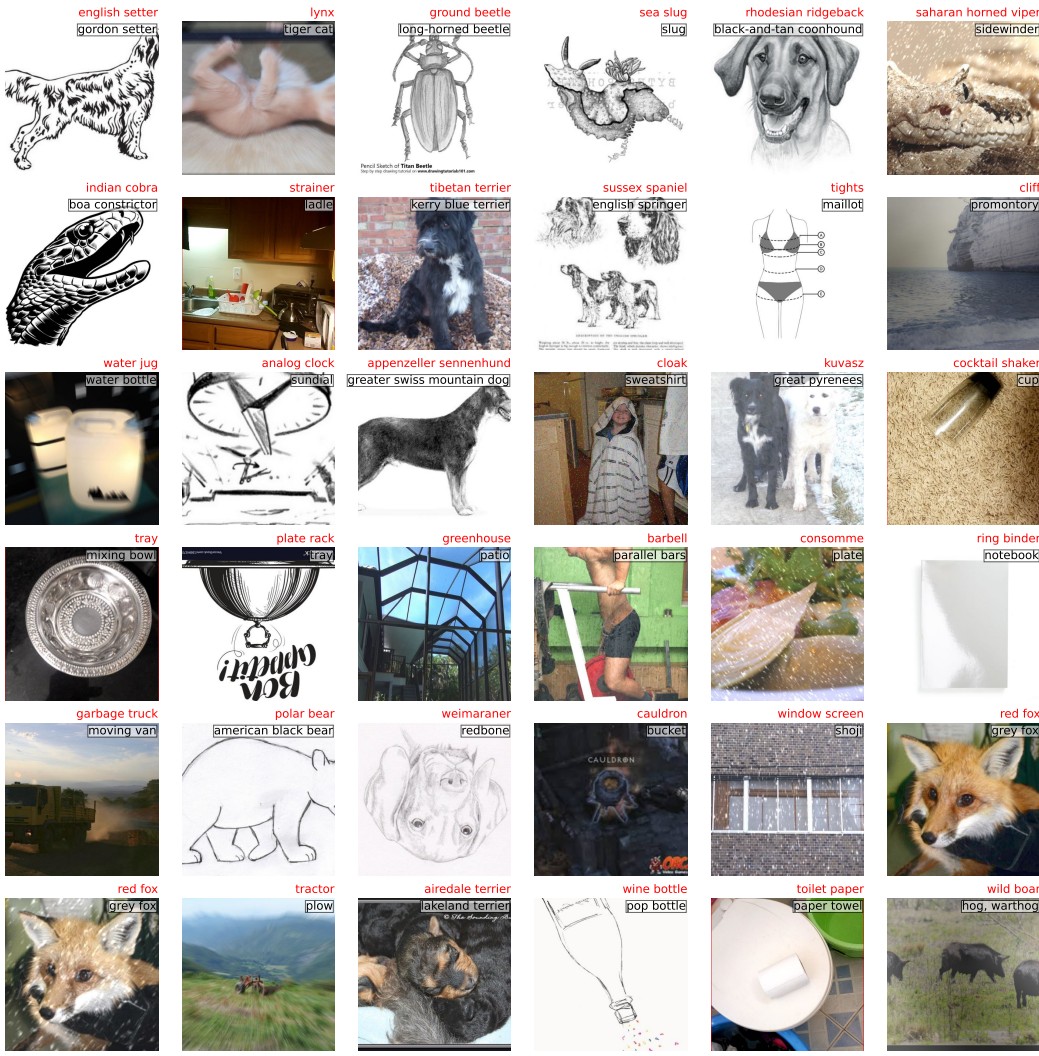

Figure A56: Examples of misclassifications by EfficientNet-L2 under the *Common* category.

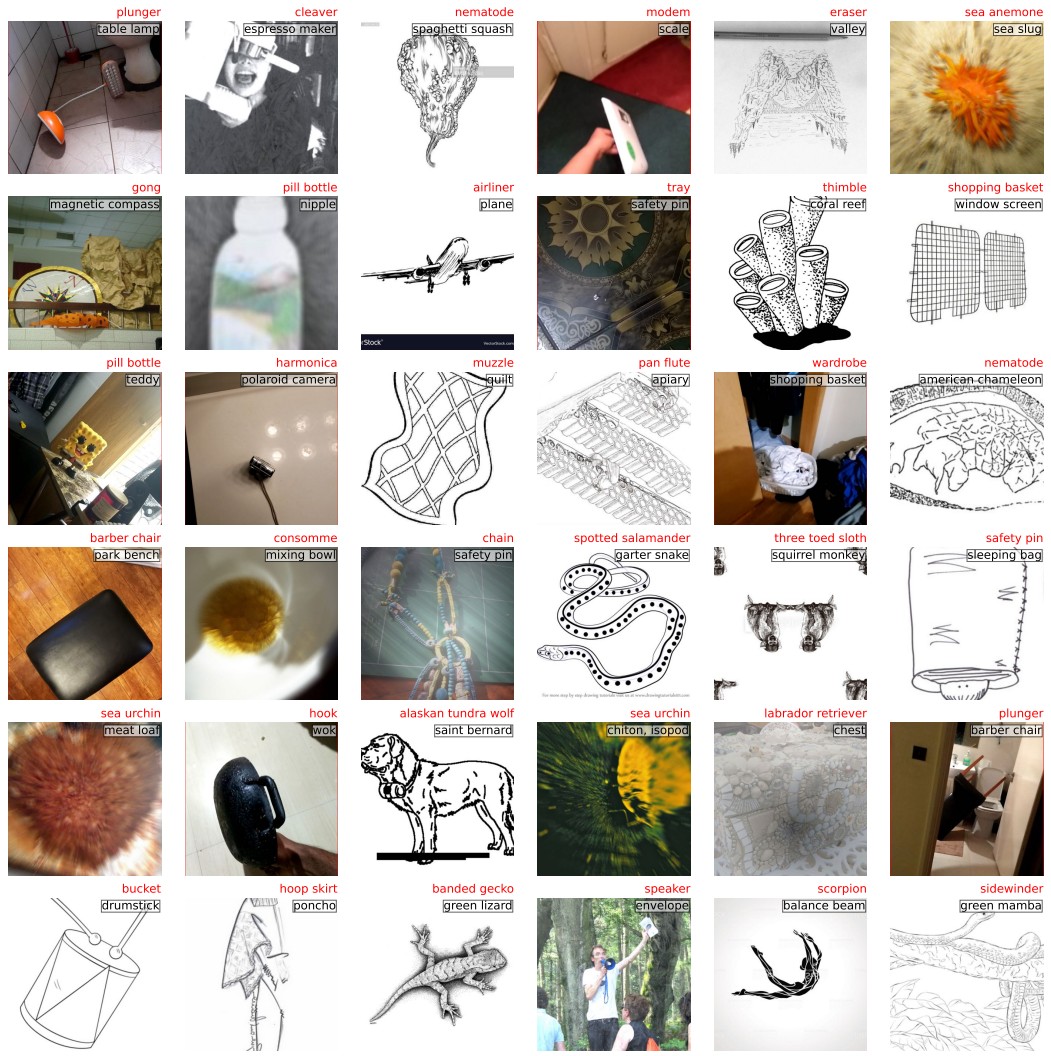

Figure A57: Examples of misclassifications by EfficientNet-L2 under the *Rare* category.

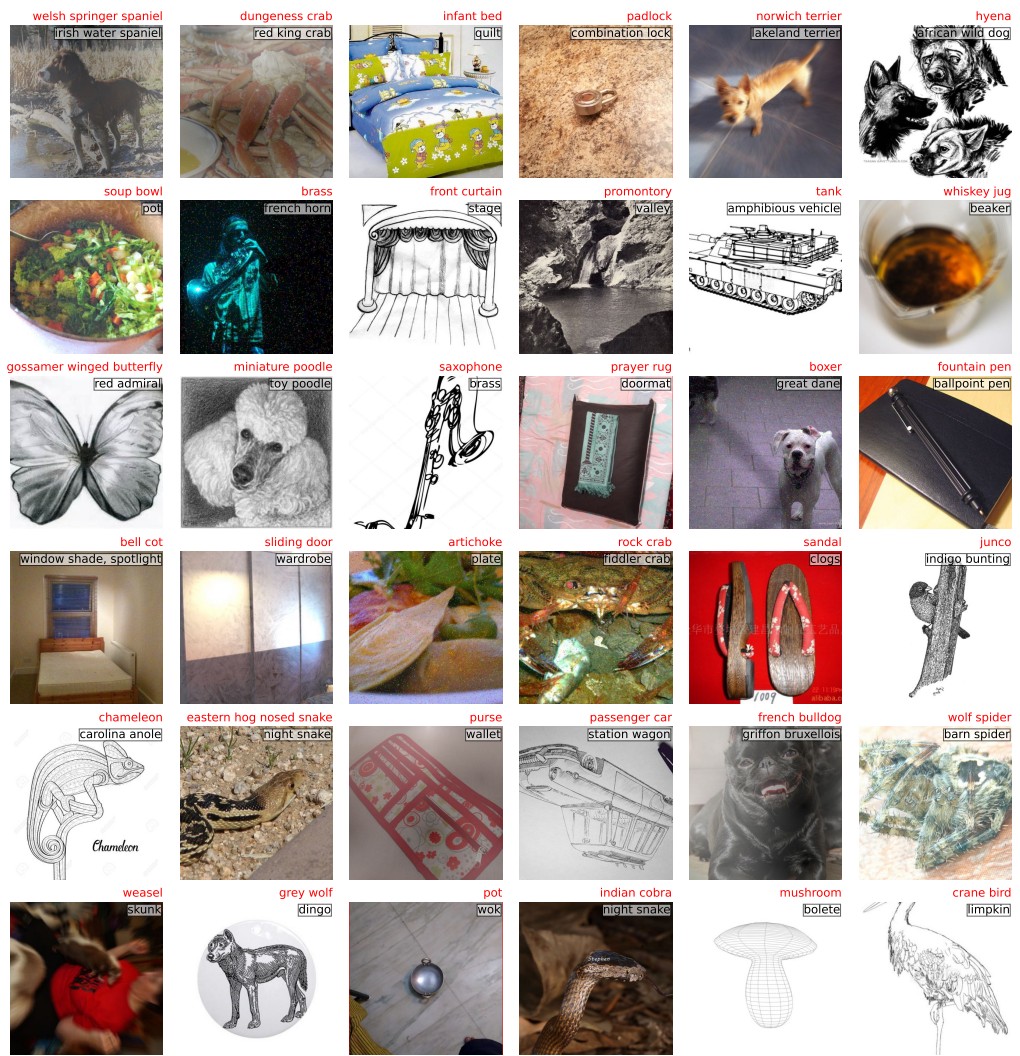

Figure A58: Examples of misclassifications by OpenCLIP under the *Common* category.

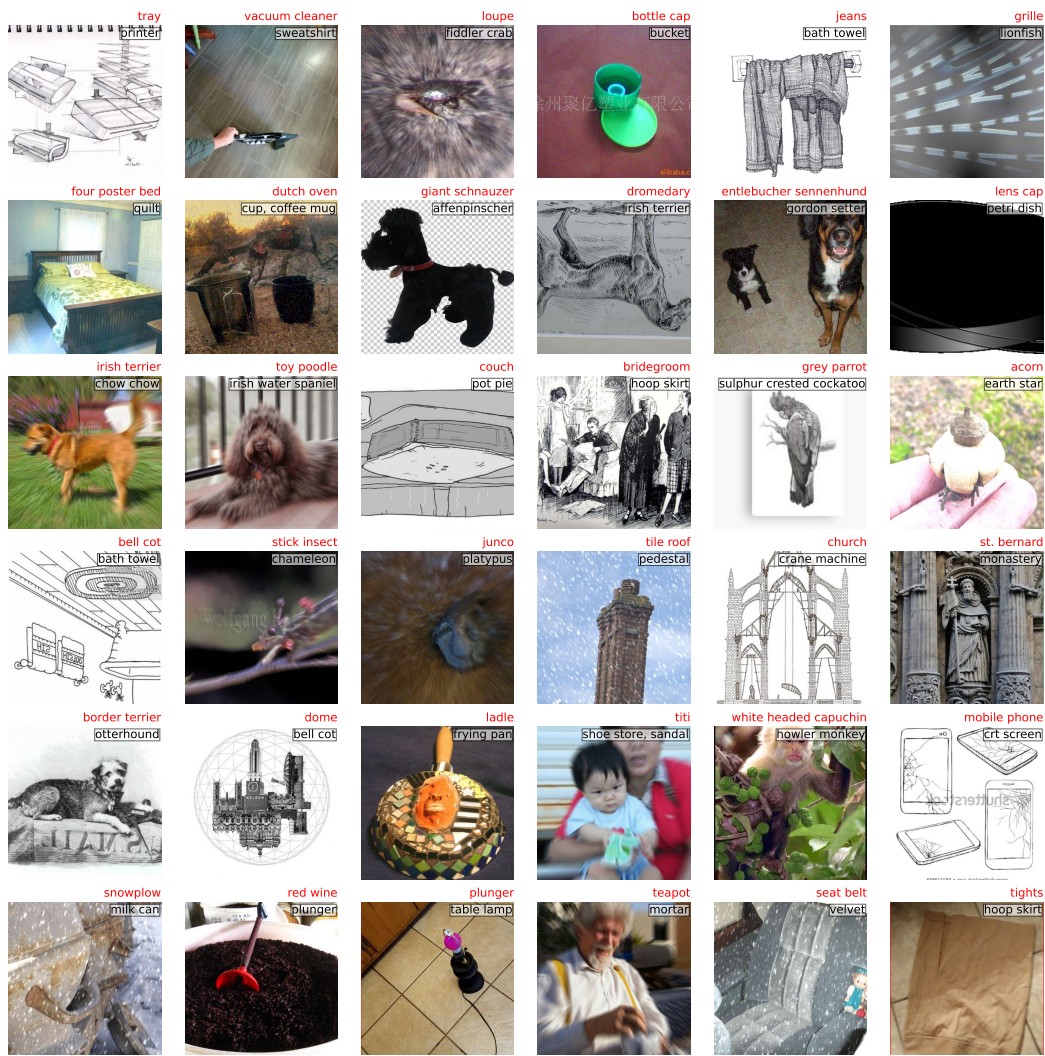

Figure A59: Examples of misclassifications by OpenCLIP under the *Rare* category.

## E.6 Evaluating OpenCLIP models' performance on ImageNet-Hard

All the models in this section are downloaded and used from the *OpenCLIP* library version 2.20.0.

Table A11: Zero-shot performance of OpenCLIP on ImageNet-Hard (%)

| Model | Pre-trained Dataset | Top-1 Accuracy |
|---|---|---|
| RN50 | `yfcc15m` | 0.80 |
| RN50 | `cc12m` | 1.18 |
| RN50-quickgelu | `yfcc15m` | 0.75 |
| RN50-quickgelu | `cc12m` | 1.08 |
| RN101 | `yfcc15m` | 0.65 |
| RN101-quickgelu | `yfcc15m` | 0.62 |
| ViT-B/32 | `laion400m_e31` | 5.34 |
| ViT-B/32 | `laion400m_e32` | 5.41 |
| ViT-B/32 | `laion2b_e16` | 5.66 |
| ViT-B/32 | `laion2b_s34b_b79k` | 6.13 |
| ViT-B/32 | `datacomp_m_s128m_b4k` | 2.79 |
| ViT-B/32 | `commonpool_m_clip_s128m_b4k` | 2.50 |
| ViT-B/32 | `commonpool_m_laion_s128m_b4k` | 2.41 |
| ViT-B/32 | `commonpool_m_image_s128m_b4k` | 2.72 |
| ViT-B/32 | `commonpool_m_text_s128m_b4k` | 2.46 |
| ViT-B/32 | `commonpool_m_basic_s128m_b4k` | 2.23 |
| ViT-B/32 | `commonpool_m_s128m_b4k` | 1.73 |
| ViT-B/32 | `datacomp_s_s13m_b4k` | 0.61 |
| ViT-B/32 | `commonpool_s_clip_s13m_b4k` | 0.84 |
| ViT-B/32 | `commonpool_s_laion_s13m_b4k` | 0.66 |
| ViT-B/32 | `commonpool_s_image_s13m_b4k` | 0.61 |
| ViT-B/32 | `commonpool_s_text_s13m_b4k` | 0.77 |
| ViT-B/32 | `commonpool_s_basic_s13m_b4k` | 0.75 |
| ViT-B/32 | `commonpool_s_s13m_b4k` | 0.43 |
| ViT-B/32-quickgelu | `laion400m_e31` | 5.34 |
| ViT-B/32-quickgelu | `laion400m_e32` | 5.28 |
| ViT-B/16 | `laion400m_e31` | 6.31 |
| ViT-B/16 | `laion400m_e32` | 6.46 |
| ViT-B/16 | `laion2b_s34b_b88k` | 7.18 |
| ViT-B/16 | `datacomp_l_s1b_b8k` | 5.98 |
| ViT-B/16 | `commonpool_l_clip_s1b_b8k` | 4.92 |
| ViT-B/16 | `commonpool_l_laion_s1b_b8k` | 4.44 |
| ViT-B/16 | `commonpool_l_image_s1b_b8k` | 4.75 |
| ViT-B/16 | `commonpool_l_text_s1b_b8k` | 5.63 |
| ViT-B/16 | `commonpool_l_basic_s1b_b8k` | 4.44 |
| ViT-B/16 | `commonpool_l_s1b_b8k` | 3.83 |
| ViT-B/16-plus-240 | `laion400m_e31` | 6.65 |
| ViT-B/16-plus-240 | `laion400m_e32` | 6.69 |
| ViT-L/14 | `laion400m_e31` | 8.83 |
| ViT-L/14 | `laion400m_e32` | 8.72 |
| ViT-L/14 | `laion2b_s32b_b82k` | 10.13 |
| ViT-L/14 | `datacomp_xl_s13b_b90k` | 15.60 |
| ViT-L/14 | `commonpool_xl_clip_s13b_b90k` | 11.58 |
| ViT-L/14 | `commonpool_xl_laion_s13b_b90k` | 11.42 |
| ViT-L/14 | `commonpool_xl_s13b_b90k` | 12.44 |
| ViT-H/14 | `laion2b_s32b_b79k` | 13.01 |
| ViT-g/14 | `laion2b_s12b_b42k` | 11.47 |
| ViT-g/14 | `laion2b_s34b_b88k` | 14.03 |
| ViT-bigG-14 | `laion2b_s39b_b160k` | 15.93 |
| roberta-ViT-B/32 | `laion2b_s12b_b32k` | 5.21 |
| xlm-roberta-base-ViT-B/32 | `laion5b_s13b_b90k` | 5.72 |
| xlm-roberta-large-ViT-H/14 | `frozen_laion5b_s13b_b90k` | 12.95 |

| | | | |
|---|---|---|---:|
| convnext_base | `laion400m_s13b_b51k` | | 4.74 |
| convnext_base_w | `laion2b_s13b_b82k` | | 6.09 |
| convnext_base_w | `laion2b_s13b_b82k_augreg` | | 7.25 |
| convnext_base_w | `laion_aesthetic_s13b_b82k` | | 5.57 |
| convnext_base_w_320 | `laion_aesthetic_s13b_b82k` | | 5.50 |
| convnext_base_w_320 | `laion_aesthetic_s13b_b82k_augreg` | | 7.14 |
| convnext_large_d | `laion2b_s26b_b102k_augreg` | | 10.39 |
| convnext_large_d_320 | `laion2b_s29b_b131k_ft` | | 10.69 |
| convnext_large_d_320 | `laion2b_s29b_b131k_ft_soup` | | 11.20 |
| convnext_xxlarge | `laion2b_s34b_b82k_augreg` | | 14.27 |
| convnext_xxlarge | `laion2b_s34b_b82k_augreg_rewind` | | 14.23 |
| convnext_xxlarge | `laion2b_s34b_b82k_augreg_soup` | | 14.68 |
| coca_ViT-B/32 | `laion2b_s13b_b90k` | | 5.83 |
| coca_ViT-B/32 | `mscoco_finetuned_laion2b_s13b_b90k` | | 0.20 |
| coca_ViT-L/14 | `laion2b_s13b_b90k` | | 10.79 |
| coca_ViT-L/14 | `mscoco_finetuned_laion2b_s13b_b90k` | | 9.28 |

Table A12: Zero-shot performance of CommonPool and DataComp models on ImageNet-Hard (%)

| Scale | Model | Pretrained | Top-1 Accuracy |
|---|---|---|---:|
| xlarge | ViT-L/14 | `datacomp_xl_s13b_b90k` | 15.60 |
| | ViT-L/14 | `commonpool_xl_clip_s13b_b90k` | 11.58 |
| | ViT-L/14 | `commonpool_xl_laion_s13b_b90k` | 11.42 |
| | ViT-L/14 | `commonpool_xl_s13b_b90k` | 12.44 |
| large | ViT-B/16 | `datacomp_l_s1b_b8k` | 5.98 |
| | ViT-B/16 | `commonpool_l_clip_s1b_b8k` | 4.92 |
| | ViT-B/16 | `commonpool_l_laion_s1b_b8k` | 4.44 |
| | ViT-B/16 | `commonpool_l_image_s1b_b8k` | 4.75 |
| | ViT-B/16 | `commonpool_l_text_s1b_b8k` | 5.63 |
| | ViT-B/16 | `commonpool_l_basic_s1b_b8k` | 4.44 |
| | ViT-B/16 | `commonpool_l_s1b_b8k` | 3.83 |
| medium | ViT-B/32 | `datacomp_m_s128m_b4k` | 2.79 |
| | ViT-B/32 | `commonpool_m_clip_s128m_b4k` | 2.50 |
| | ViT-B/32 | `commonpool_m_laion_s128m_b4k` | 2.41 |
| | ViT-B/32 | `commonpool_m_image_s128m_b4k` | 2.72 |
| | ViT-B/32 | `commonpool_m_text_s128m_b4k` | 2.46 |
| | ViT-B/32 | `commonpool_m_basic_s128m_b4k` | 2.23 |
| | ViT-B/32 | `commonpool_m_s128m_b4k` | 1.73 |
| small | ViT-B/32 | `datacomp_s_s13m_b4k` | 0.61 |
| | ViT-B/32 | `commonpool_s_clip_s13m_b4k` | 0.84 |
| | ViT-B/32 | `commonpool_s_laion_s13m_b4k` | 0.66 |
| | ViT-B/32 | `commonpool_s_image_s13m_b4k` | 0.61 |
| | ViT-B/32 | `commonpool_s_text_s13m_b4k` | 0.77 |
| | ViT-B/32 | `commonpool_s_basic_s13m_b4k` | 0.75 |
| | ViT-B/32 | `commonpool_s_s13m_b4k` | 0.43 |

## E.7 Evaluating classifiers on ImageNet-Hard-4K

Table A13: Top-1 accuracy (%) on ImageNet-Hard-4K. Most models obtain a lower accuracy compared to their corresponding accuracy on ImageNet-Hard.

| Classifier | Accuracy | Classifier | Accuracy | Classifier | Accuracy |
|---|---|---|---|---|---|
| AlexNet | 7.08 (-0.16) | ViT-B/32 | 18.12 (-0.40) | CLIP-ViT-L/14@224px | 1.81 (-0.05) |
| VGG-16 | 11.32 (-0.68) | EfficientNet-B0@224px | 12.94 (-3.63) | CLIP-ViT-L/14@336px | 1.88 (-0.14) |
| ResNet-18 | 10.42 (-0.44) | EfficientNet-B7@600px | 18.67 (-4.53) | OpenCLIP-ViT-bigG-14 | 14.33 (-1.60) |
| ResNet-50 | 13.93 (-0.81) | EfficientNet-L2@800px | 28.42 (-10.58) | OpenCLIP-ViT-L-14 | 13.04 (-2.56) |

## E.8 Obviously ill-posed samples from ImageNet-Sketch

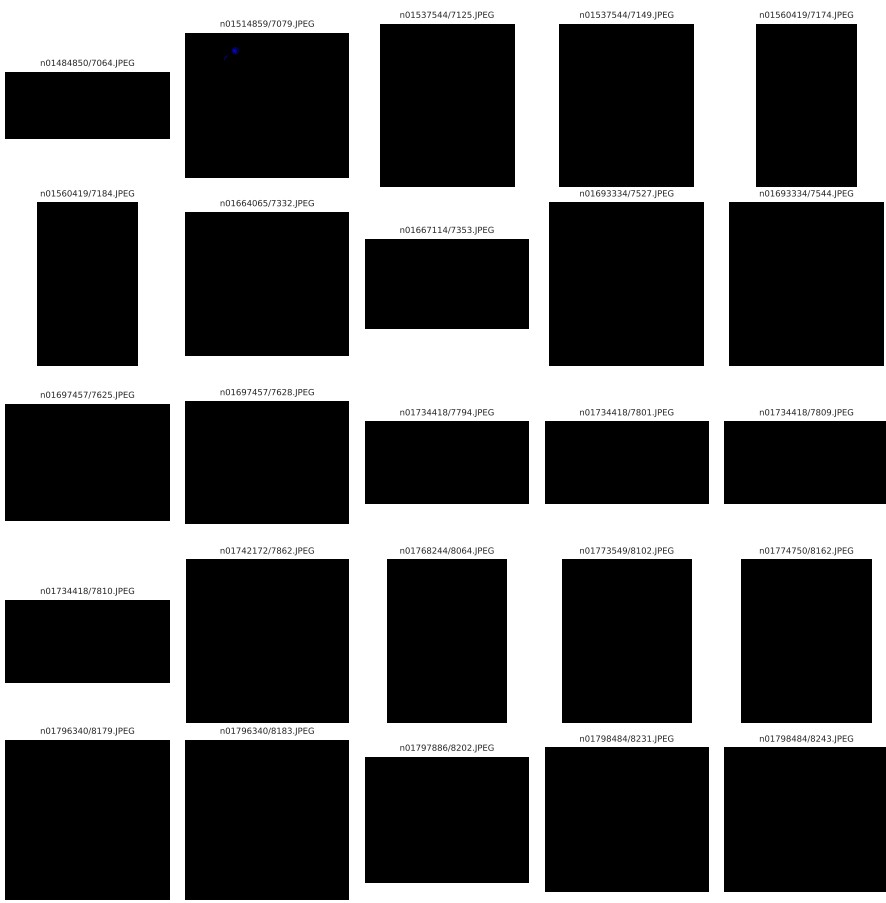

Figure A60: Sample images from ImageNet-Sketch that are completely black.

# F Additional Details

## F.1 Additional results for performance of classifiers using maximum possible accuracy

In this section, we present the delta values for Tab. 1, which represent the difference in relation to the 1-crop accuracy for each cell.

Table A14: On in-distribution data (IN & ReaL) there exists a substantial improvement when models are provided with an optimal zoom, either selected from 36 (b) or 324 pre-defined zoom crops (c). In contrast, OOD benchmarks still pose a significant challenge to IN-trained models even with optimal zooming (i.e., all upper-bound accuracy scores $< 80\%$). Improvements are respected to the standard 1-crop accuracy.

| | IN | ReaL | IN+ReaL | IN-A | IN-R | IN-S | ON |
|---|---|---|---|---|---|---|---|
| (a) *Standard top-1 accuracy based on N = 1 crop* | | | | | | | |
| AlexNet | 56.16 | 62.67 | 61.76 | 1.75 | 21.10 | 10.05 | 14.23 |
| VGG-16 | 71.37 | 78.90 | 78.52 | 2.69 | 26.98 | 16.78 | 28.32 |
| ResNet-18 | 69.45 | 76.94 | 76.47 | 1.37 | 32.14 | 19.41 | 27.59 |
| ResNet-50 | 75.75 | 82.63 | 82.97 | 0.21 | 35.39 | 22.91 | 36.18 |
| ViT-B/32 | 75.75 | 81.89 | 82.59 | 9.64 | 41.29 | 26.83 | 30.89 |
| CLIP-ViT-L/14 | 75.03 | 80.68 | 81.95 | 71.28 | 87.74 | 58.23 | 66.32 |
| (b) *Upper-bound accuracy using N = 36 crops* | | | | | | | |
| AlexNet | 85.19 (+29.03) | 90.30 (+27.63) | 89.74 (+27.98) | 31.37 (+29.62) | 47.04 (+25.94) | 24.40 (+14.35) | 49.17 (+34.94) |
| VGG-16 | 92.30 (+20.93) | 96.08 (+17.18) | 95.81 (+17.29) | 46.69 (+44.00) | 52.86 (+25.88) | 34.34 (+17.56) | 62.94 (+34.62) |
| ResNet-18 | 92.08 (+22.63) | 95.97 (+19.03) | 95.73 (+19.26) | 47.48 (+46.11) | 58.85 (+26.71) | 37.91 (+18.50) | 63.08 (+35.49) |
| ResNet-50 | 94.46 (+18.71) | 97.36 (+14.73) | 97.40 (+14.43) | 55.68 (+55.47) | 61.42 (+26.03) | 41.71 (+18.80) | 69.60 (+33.42) |
| ViT-B/32 | 95.05 (+19.30) | 97.61 (+15.72) | 97.88 (+15.29) | 68.43 (+58.79) | 68.77 (+27.48) | 49.10 (+22.27) | 70.30 (+39.41) |
| CLIP-ViT-L/14 | 94.19 (+19.16) | 97.32 (+16.64) | 97.56 (+15.61) | 97.16 (+25.88) | 98.60 (+10.86) | 83.77 (+25.54) | 89.59 (+23.27) |
| (c) *Upper-bound accuracy using N = 324 crops* | | | | | | | |
| AlexNet | 90.03 (+33.87) | 93.85 (+31.18) | 93.48 (+31.72) | 42.23 (+40.48) | 55.52 (+34.42) | 29.53 (+19.48) | 59.65 (+45.42) |
| VGG-16 | 95.30 (+23.93) | 97.90 (+19.00) | 97.66 (+19.14) | 58.27 (+55.58) | 60.88 (+33.90) | 39.90 (+23.12) | 71.85 (+43.53) |
| ResNet-18 | 95.15 (+25.70) | 97.76 (+20.82) | 97.55 (+21.08) | 58.87 (+57.50) | 66.89 (+34.75) | 43.68 (+24.27) | 71.44 (+43.85) |
| ResNet-50 | 96.78 (+21.03) | 98.62 (+15.99) | 98.57 (+15.60) | 66.68 (+66.47) | 68.84 (+33.45) | 47.64 (+24.73) | 76.83 (+40.65) |
| ViT-B/32 | **97.19** (+21.44) | **98.75** (+16.86) | **98.91** (+16.32) | 78.03 (+68.39) | 75.58 (+34.29) | 55.99 (+29.16) | 79.28 (+48.39) |
| CLIP-ViT-L/14 | 96.78 (+21.75) | 98.69 (+18.01) | 98.80 (+16.85) | **98.45** (+27.17) | **99.20** (+11.46) | **89.00** (+30.77) | **93.13** (+26.81) |

## F.2 Comparing modern architectures with regard to their maximum possible accuracy.

In this section, we repeated the experiment presented in Tab. 1 in order to include a broader range of architectures and conduct a rank analysis. Specifically, we added MaxViT [72], Swin Transformer [43], ConvNext [44], EfficientNet [69], and MobileNetV3 [28].

Table A15: Repeating the experiment in Tab. 1 for various architectures, please note that the small variations of ResNet-50 and ViT-B/32 models arise from the use of different CUDA and PyTorch versions.

(a) *Standard top-1 accuracy based on $N = 1$ crop*

| Classifier | ImageNet | ImageNet-ReaL | ImageNet-A |
|---|---|---|---|
| MobileNetV3-Small | 67.42 | 74.47 | 2.73 |
| ResNet-50 | 75.74 | 82.63 | 0.17 |
| EfficientNet-B0 | 77.69 | 84.13 | 7.01 |
| ViT-B/32 | 75.75 | 81.89 | 9.51 |
| ConvNext-Base | 83.49 | 88.19 | 33.99 |
| Swin-B | 83.37 | 87.80 | 34.87 |
| MaxViT-T | 83.49 | 88.05 | 36.00 |

(b) *Upper-bound accuracy using $N = \mathbf{36}$ crops*

| | | | |
|---|---|---|---|
| MobileNetV3-Small | 91.79 | 95.63 | 44.45 |
| ResNet-50 | 94.47 | 97.35 | 54.97 |
| EfficientNet-B0 | 94.87 | 97.49 | 59.33 |
| ViT-B/32 | 95.04 | 97.60 | 68.33 |
| ConvNext-Base | 95.78 | 97.68 | 82.25 |
| Swin-B | 95.95 | 97.75 | 82.81 |
| MaxViT-T | 96.11 | 97.99 | 83.33 |

(c) *Upper-bound accuracy using $N = \mathbf{324}$ crops*

| | | | |
|---|---|---|---|
| MobileNetV3-Small | 95.20 | 97.66 | 57.53 |
| ResNet-50 | 96.77 | 98.63 | 66.15 |
| EfficientNet-B0 | 96.87 | 98.60 | 69.20 |
| ViT-B/32 | 97.19 | 98.75 | 77.77 |
| ConvNext-Base | 97.16 | 98.53 | 87.16 |
| Swin-B | 97.39 | 98.69 | 88.15 |
| MaxViT-T | 97.45 | 98.83 | 88.24 |

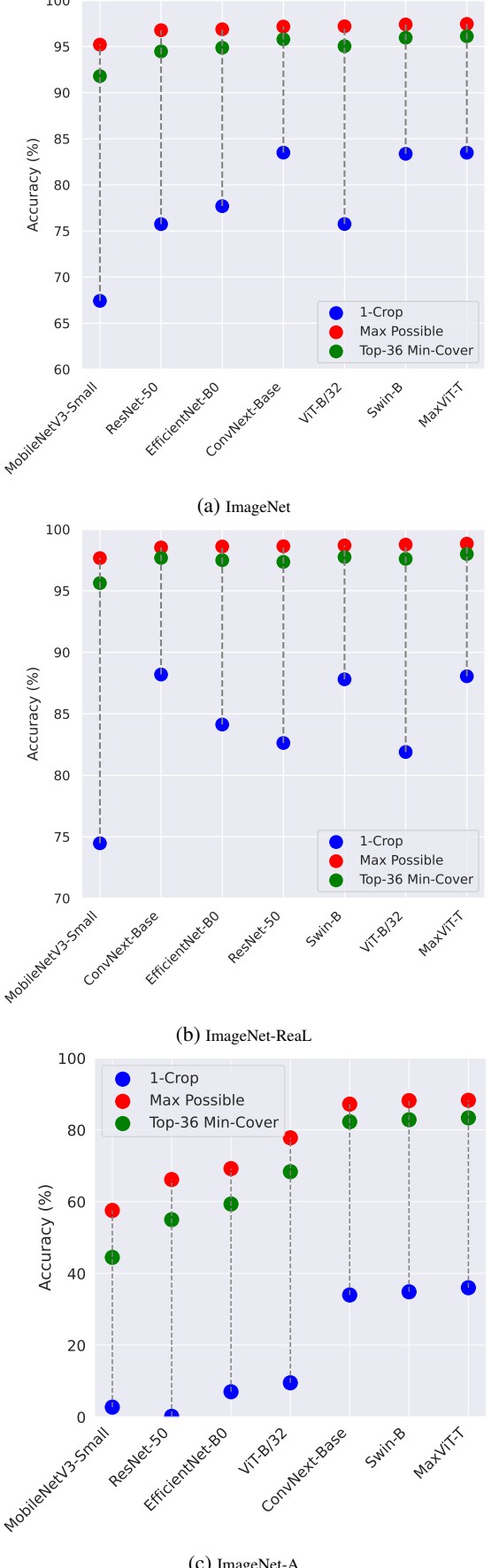

(a) ImageNet

(b) ImageNet-ReaL

(c) ImageNet-A

Figure A61: Comparing modern architectures using maximum possible accuracy

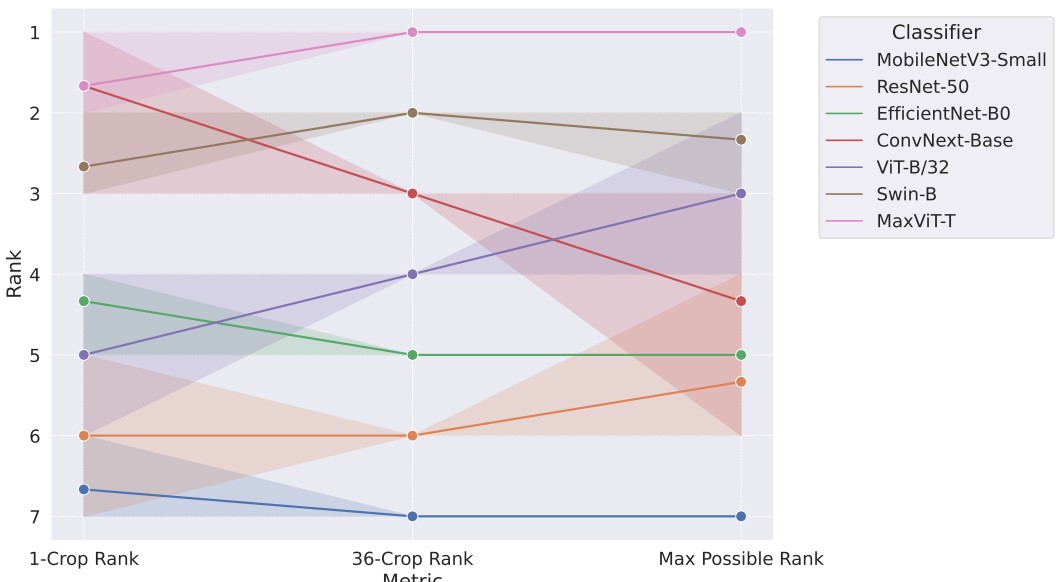

Figure A62: Average rank analysis of various architectures on ImageNet, ImageNet-ReaL, and ImageNet-A using different metrics. Max-ViT consistently emerges as the top performer across all experiments. Despite its high 1-crop accuracy, ConvNext's rank declined when more crops were included. Conversely, ViT-B/32 exhibited a reverse trend; while its 1-crop rank was not robust, the inclusion of more crops elevated its accuracy beyond that of ConvNext.

### F.3 Error-analysis for MEMO results

In this section, we repeated the MEMO experiment using various random seeds to evaluate the consistency of the results. Our findings demonstrate a consistent trend between MEMO and RRC, indicating that this relationship holds steady regardless of variations in individual runs.

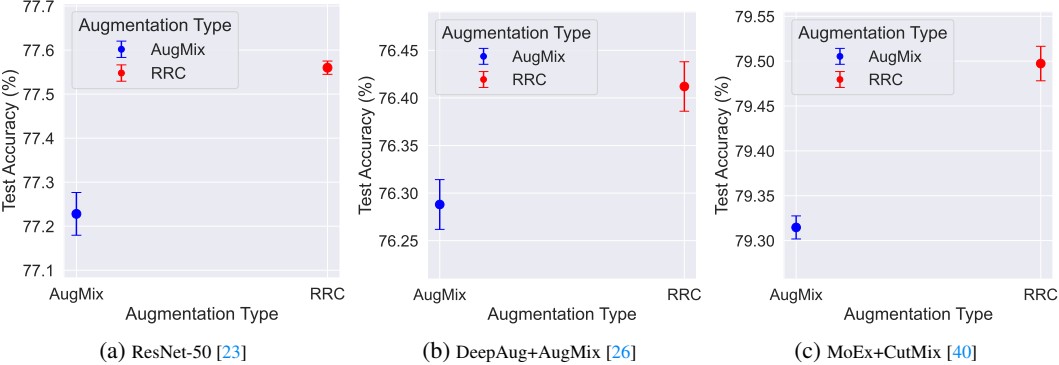

(a) ResNet-50 [23]  (b) DeepAug+AugMix [26]  (c) MoEx+CutMix [40]

Figure A63: Error bars representing the outcomes of the MEMO experiment on ImageNet, conducted with three distinct random seeds.

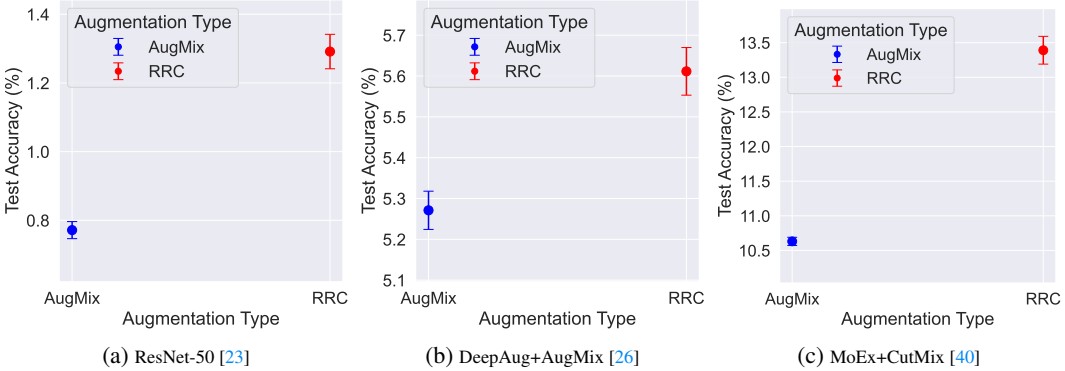

(a) ResNet-50 [23]  (b) DeepAug+AugMix [26]  (c) MoEx+CutMix [40]

Figure A64: Error bars representing the outcomes of the MEMO experiment on ImageNet-A, conducted with three distinct random seeds.

### F.4 Grad-CAM visualizations for MEMO + RRC

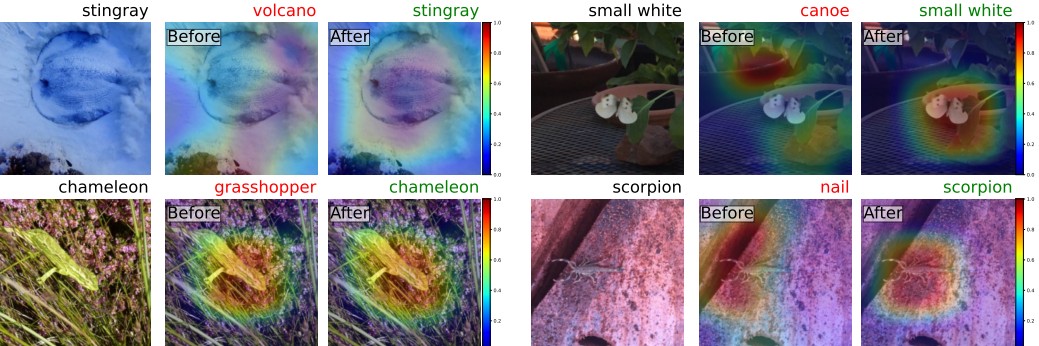

Figure A65: Additional Grad-CAM visualization for the final convolutional layer of a ResNet-50 before and after MEMO + RRC update.

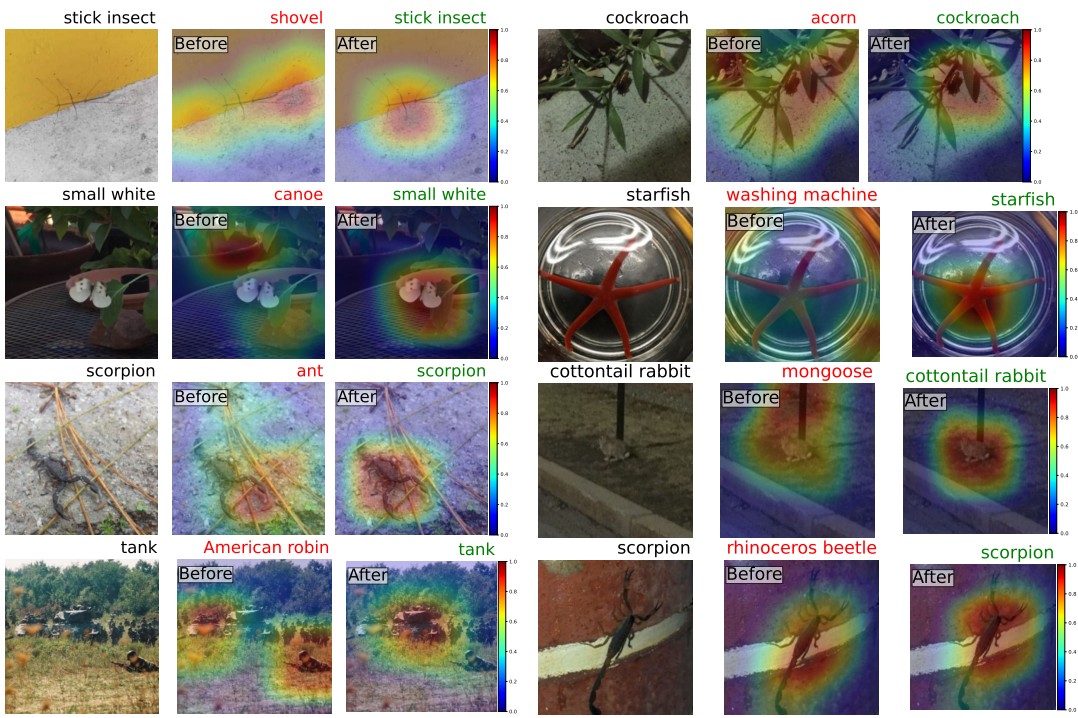

Figure A66: Additional Grad-CAM visualization for the final convolutional layer of a ResNet-50 before and after MEMO + RRC update.

# G  Datasheet for ImageNet-Hard

## G.1  Motivation

The questions in this section are primarily intended to encourage dataset creators to clearly articulate their reasons for creating the dataset and to promote transparency about funding interests. The latter may be particularly relevant for datasets created for research purposes.

- **For what purpose was the dataset created?** Was there a specific task in mind? Was there a specific gap that needed to be filled? Please provide a description.

  The ImageNet-Hard is a new benchmark to test the robustness of state-of-the-art image classifiers. It comprises an array of challenging images collected from *six* validation datasets of ImageNet. This dataset challenges state-of-the-art image classification models because even by perfectly localizing the key objects, the state-of-the-art classifiers still fail to correctly recognize.

- **Who created the dataset (e.g., which team, research group) and on behalf of which entity (e.g., company, institution, organization)?**

  The dataset was created in collaboration efforts between the University of Alberta, Canada, and Auburn University, USA; mostly by, Mohammad Reza Taesiri and Anh Nguyen.

- **Who funded the creation of the dataset?** If there is an associated grant, please provide the name of the grantor and the grant name and number.

  Anh Nguyen was supported by NSF Grant No. 2145767, and donations from NaphCare Foundation, and Adobe Research

- **Any other comments?**

  No.

## G.2  Composition

Dataset creators should read through *these questions* prior to any data collection and then provide answers once *data* collection is complete. Most of the questions *in this section* are intended to provide dataset consumers with the information they need to make informed decisions about using the dataset for their chosen tasks. Some of the questions are *designed to elicit* information about compliance with the EU's General Data Protection Regulation (GDPR) or comparable regulations in other jurisdictions.

*Questions that apply only to datasets that relate to people are grouped together at the end of the section. We recommend taking a broad interpretation of whether a dataset relates to people. For example, any dataset containing text that was written by people relates to people.*

- **What do the instances that comprise the dataset represent (e.g., documents, photos, people, countries)?** Are there multiple types of instances (e.g., movies, users, and ratings; people and interactions between them; nodes and edges)? Please provide a description.

  Each instance of the ImageNet-Hard dataset corresponds to an image and at least one groundtruth label that will be used to assess image classifiers.

- **How many instances are there in total (of each type, if appropriate)?**

  There are 10,980 images in this dataset.

- **Does the dataset contain all possible instances or is it a sample (not necessarily random) of instances from a larger set?** If the dataset is a sample, then what is the larger set? Is the sample representative of the larger set (e.g., geographic coverage)? If so, please describe how this representativeness was validated/verified. If it is not representative of the larger set, please describe why not (e.g., to cover a more diverse range of instances, because instances were withheld or unavailable).

  ImageNet-Hard is a combination of various publicly available datasets. We tried multiple refinement steps to make sure to get the best possible samples for the intended purpose. Then, it is not representative of any larger sets but a selective combination of multiple sets.

- **What data does each instance consist of?** "Raw" data (e.g., unprocessed text or images) or features? In either case, please provide a description.

  The dataset contains both raw and processed images. The processed images come from ImageNet-C. Details can be found in Sec. 4.4.

- **Is there a label or target associated with each instance?** If so, please provide a description.

  Yes. Each sample has the label that is the folder name the image belongs to. Basically, we follow the structure of the ImageNet paper [59].

- **Is any information missing from individual instances?** If so, please provide a description, explaining why this information is missing (e.g., because it was unavailable). This does not include intentionally removed information, but might include, e.g., redacted text.

  No.

- **Are relationships between individual instances made explicit (e.g., users' movie ratings, social network links)?** If so, please describe how these relationships are made explicit.

  No. The individual instances has no relationships.

- **Are there recommended data splits (e.g., training, development/validation, testing)?** If so, please provide a description of these splits, explaining the rationale behind them.

  No. This dataset is created for the testing purposes.

- **Are there any errors, sources of noise, or redundancies in the dataset?** If so, please provide a description.

  To the best of our knowledge, No. We tried our best efforts to filter any errors, sources of noise, or redundancies to create the ImageNet-Hard dataset.

- **Is the dataset self-contained, or does it link to or otherwise rely on external resources (e.g., websites, tweets, other datasets)?** If it links to or relies on external resources, a) are there guarantees that they will exist, and remain constant, over time; b) are there official archival versions of the complete dataset (i.e., including the external resources as they existed at the time the dataset was created); c) are there any restrictions (e.g., licenses, fees) associated with any of the external resources that might apply to a *dataset consumer*? Please provide descriptions of all external resources and any restrictions associated with them, as well as links or other access points, as appropriate.

  Yes. It does link and inherits from existing image datasets and was detailed in Sec. 4.4.

- **Does the dataset contain data that might be considered confidential (e.g., data that is protected by legal privilege or by doctor–patient confidentiality, data that includes the content of individuals' non-public communications)?** If so, please provide a description.

  No.

- **Does the dataset contain data that, if viewed directly, might be offensive, insulting, threatening, or might otherwise cause anxiety?** If so, please describe why.

  No.

*If the dataset does not* relate to people, you may skip the remaining questions in this section.

- **Does the dataset identify any subpopulations (e.g., by age, gender)?** If so, please describe how these subpopulations are identified and provide a description of their respective distributions within the dataset.

  N/A.

- **Is it possible to identify individuals (i.e., one or more natural persons), either directly or indirectly (i.e., in combination with other data) from the dataset?** If so, please describe how.

  N/A.

- **Does the dataset contain data that might be considered sensitive in any way (e.g., data that reveals race or ethnic origins, sexual orientations, religious beliefs, political opinions or union memberships, or locations; financial or health data; biometric or genetic data; forms of government identification, such as social security numbers; criminal history)?** If so, please provide a description.

  N/A.

- **Any other comments?**

  No.

## G.3 Collection Process

As with the *questions in the* previous section, dataset creators should read through these questions prior to any data collection to flag potential issues and then provide answers once collection is complete. *In addition to the goals outlined in the previous section, the questions in this section are designed to elicit information that may help researchers and practitioners to create alternative datasets with similar characteristics. Again, questions that apply only to datasets that relate to people are grouped together at the end of the section.*

- **How was the data associated with each instance acquired?** Was the data directly observable (e.g., raw text, movie ratings), reported by subjects (e.g., survey responses), or indirectly inferred/derived from other data (e.g., part-of-speech tags, model-based guesses for age or language)? If the data was reported by subjects or indirectly inferred/derived from other data, was the data validated/verified? If so, please describe how.

  The dataset is linked from other 6 datasets. Please find the contribution of original daatasets in Appendix E.1)

- **What mechanisms or procedures were used to collect the data (e.g., hardware apparatuses or sensors, manual human curation, software programs, software APIs)?** How were these mechanisms or procedures validated?

  We used both algorithm and human efforts to collect the data. Algorithms were used to choose hard samples from various datasets. We then used two human groups and their agreement to make sure the high quality of the process. Details for the human validation in Sec. 4.4. Finally, we removed samples that have debatable labels (e.g. sunglass vs. sunglasses).

- **If the dataset is a sample from a larger set, what was the sampling strategy (e.g., deterministic, probabilistic with specific sampling probabilities)?**

  No, the dataset was not a subset of a larger set.

- **Who was involved in the data collection process (e.g., students, crowdworkers, contractors) and how were they compensated (e.g., how much were crowdworkers paid)?**

  In the data collection process, we involved students who voluntarily participated.

- **Over what timeframe was the data collected?** Does this timeframe match the creation timeframe of the data associated with the instances (e.g., recent crawl of old news articles)? If not, please describe the timeframe in which the data associated with the instances was created.

  Feedback data was collected from April 20 2023 – May 4 2023.

- **Were any ethical review processes conducted (e.g., by an institutional review board)?** If so, please provide a description of these review processes, including the outcomes, as well as a link or other access point to any supporting documentation.

  N/A. In this study, humans are not the subjects. Their voluntary feedback, however, is used to filter out incorrectly labelled samples from the original 6 datasets.

*If the dataset does not relate to people, you may skip the remaining questions in this section.*

- **Did you collect the data from the individuals in question directly, or obtain it via third parties or other sources (e.g., websites)?**

  We only involved individuals in the label verification step (i.e. 3133 samples). The answers from individuals directly affect if one of those 3133 samples will be kept or not.

- **Were the individuals in question notified about the data collection?** If so, please describe (or show with screenshots or other information) how notice was provided, and provide a link or other access point to, or otherwise reproduce, the exact language of the notification itself.

  N/A. In this study, humans are not the subjects. Their voluntary feedback, however, is used to filter out incorrectly labelled samples from the original 6 datasets.

- **Did the individuals in question consent to the collection and use of their data?** If so, please describe (or show with screenshots or other information) how consent was requested and provided, and provide a link or other access point to, or otherwise reproduce, the exact language to which the individuals consented.

  N/A. In this study, humans are not the subjects. Their voluntary feedback, however, is used to filter out incorrectly labelled samples from the original 6 datasets.

- **If consent was obtained, were the consenting individuals provided with a mechanism to revoke their consent in the future or for certain uses?** If so, please provide a description, as well as a link or other access point to the mechanism (if appropriate).

  N/A. In this study, humans are not the subjects. Their voluntary feedback, however, is used to filter out incorrectly labelled samples from the original 6 datasets.

- **Has an analysis of the potential impact of the dataset and its use on data subjects (e.g., a data protection impact analysis) been conducted?** If so, please provide a description of this analysis, including the outcomes, as well as a link or other access point to any supporting documentation.

  N/A. In this study, humans are not the subjects. Their voluntary feedback, however, is used to filter out incorrectly labelled samples from the original 6 datasets.

- **Any other comments?**

  No.

## G.4 Preprocessing/cleaning/labeling

Dataset creators should read through these questions prior to any preprocessing, cleaning, or labeling and then provide answers once these tasks are complete. The questions in this section are intended to provide dataset consumers with the information they need to determine whether the "raw" data has been processed in ways that are compatible with their chosen tasks. For example, text that has been converted into a "bag-of-words" is not suitable for tasks involving word order.

- **Was any preprocessing/cleaning/labeling of the data done (e.g., discretization or bucketing, tokenization, part-of-speech tagging, SIFT feature extraction, removal of instances, processing of missing values)?** If so, please provide a description. If not, you may skip the remaining questions in this section.

  Yes, we did cleaning up the data. We removed 370 images associated with the labels `sunglass`, `sunglasses`, `tub`, `bathtub`, `cradle`, `bassinet`, `projectile`, and `missile`, *i.e.*, the classes that often contain similar images that belong to more than one class. Also, the agreement human setup in Sec. 4.4 helps us remove bad samples.

- **Was the "raw" data saved in addition to the preprocessed/cleaned/labeled data (e.g., to support unanticipated future uses)?** If so, please provide a link or other access point to the "raw" data.

  No, we did not save. However, the "raw" data (i.e. six image datasets) can be found on the Internet.

- **Is the software *that was* used to preprocess/clean/label the data available?** If so, please provide a link or other access point.

  Yes, we provided the source code for the algorithms that can be found at here.

- **Any other comments?**

  No.

## G.5 Uses

The questions in this section are intended to encourage dataset creators to reflect on the tasks for which the dataset should and should not be used. By explicitly highlighting these tasks, dataset creators can help dataset consumers to make informed decisions, thereby avoiding potential risks or harms.

- **Has the dataset been used for any tasks already?** If so, please provide a description.

  The dataset was used for image classification in this paper.

- **Is there a repository that links to any or all papers or systems that use the dataset?** If so, please provide a link or other access point.

  The papers that use the dataset could be found at this paperwithcode repository.

- **What (other) tasks could the dataset be used for?**

  Studying the effects of upscaling on image classification.

- **Is there anything about the composition of the dataset or the way it was collected and preprocessed/cleaned/labeled that might impact future uses?** For example, is there anything that a *dataset consumer* might need to know to avoid uses that could result in unfair treatment of individuals or groups (e.g., stereotyping, quality of service issues) or other risks or harms (e.g., *legal risks,* financial harms)? If so, please provide a description. Is there anything a *dataset consumer* could do to mitigate these risks or harms?

  No.

- **Are there tasks for which the dataset should not be used?** If so, please provide a description.

  No.

- **Any other comments?**

  No.

## G.6   Distribution

Dataset creators should provide answers to these questions prior to distributing the dataset either internally within the entity on behalf of which the dataset was created or externally to third parties.

- **Will the dataset be distributed to third parties outside of the entity (e.g., company, institution, organization) on behalf of which the dataset was created?** If so, please provide a description.

  The dataset will be shared with the common public.

- **How will the dataset will be distributed (e.g., tarball on website, API, GitHub)?** Does the dataset have a digital object identifier (DOI)?

  It has a GitHub repository and Hugging Face page. It has no DOI.

- **When will the dataset be distributed?**

  Until June 5 2023, the dataset had been already distributed.

- **Will the dataset be distributed under a copyright or other intellectual property (IP) license, and/or under applicable terms of use (ToU)?** If so, please describe this license and/or ToU, and provide a link or other access point to, or otherwise reproduce, any relevant licensing terms or ToU, as well as any fees associated with these restrictions.

  No.

- **Have any third parties imposed IP-based or other restrictions on the data associated with the instances?** If so, please describe these restrictions, and provide a link or other access point to, or otherwise reproduce, any relevant licensing terms, as well as any fees associated with these restrictions.

  There are no such restrictions.

- **Do any export controls or other regulatory restrictions apply to the dataset or to individual instances?** If so, please describe these restrictions, and provide a link or other access point to, or otherwise reproduce, any supporting documentation.

  N/A.

- **Any other comments?**

  No.

## G.7   Maintenance

As with the questions in the previous section, dataset creators should provide answers to these questions prior to distributing the dataset. The questions *in this section* are intended to encourage dataset creators to plan for dataset maintenance and communicate this plan to dataset consumers.

- **Who *will be* supporting/hosting/maintaining the dataset?**

  The authors of the paper will be supporting/hosting/maintaining the dataset.

- **How can the owner/curator/manager of the dataset be contacted (e.g., email address)?**

  Please reach out to mtaesiri@gmail.com and anh.ng8@gmail.com.

- **Is there an erratum?** If so, please provide a link or other access point.

  No, there is no erratum.

- **Will the dataset be updated (e.g., to correct labeling errors, add new instances, delete instances)?** If so, please describe how often, by whom, and how updates will be communicated to *dataset consumers* (e.g., mailing list, GitHub)?

  Yes, we anticipate either correcting the labels or removing some images entirely to reduce the noise in the dataset. We manage the versioning through the Hugging Face Dataset platform.

- **If the dataset relates to people, are there applicable limits on the retention of the data associated with the instances (e.g., were the individuals in question told that their data would be retained for a fixed period of time and then deleted)?** If so, please describe these limits and explain how they will be enforced.

  N/A

- **Will older versions of the dataset continue to be supported/hosted/maintained?** If so, please describe how. If not, please describe how its obsolescence will be communicated to *dataset consumers*.

  No, the older versions will not be updated. Users are encouraged to use the latest version.

- **If others want to extend/augment/build on/contribute to the dataset, is there a mechanism for them to do so?** If so, please provide a description. Will these contributions be validated/verified? If so, please describe how. If not, why not? Is there a process for communicating/distributing these contributions to *dataset consumers*? If so, please provide a description.

  Please reach out to mtaesiri@gmail.com or anhng8@gmail.com.

- **Any other comments?**

  No.

