# OpenReview forum: "ImageNet-Hard: The Hardest Images Remaining from a Study of the Power of Zoom and Spatial Biases in Image Classification"
_NeurIPS.cc/2023/Track/Datasets_and_Benchmarks — NeurIPS 2023 Datasets and Benchmarks Poster_

### Official Review · Reviewer_pbFv · 2023-07-15
**Proposing a challenging dataset by tackling the shape bias in ImageNet is interesting but there are several issues that need improvements.**

**Rating:** 7
**Confidence:** 4
**Correctness:** Yes
**Clarity:** Yes

**Strengths:**

- The analysis on shape bias with ImageNet is interesting. Previous studies mainly focused on the texture/color bias for improving the image classification performance of ImageNet. The authors made a decent approach with a different perspective for the analysis.
- Based on the analysis, I believe that the proposed ImageNet-Hard would be another impactful benchmark that future researchers may not easily conquer.
- The authors reported the extensive experiments and details regarding the zoom-in transforms in the supplementary. Such details enable the future researchers to re-implement the paper with minimal errors.

**Additional Feedback:**

None.

**Documentation:**

Yes, most of the details are included in the supplementary.

**Limitations:**

Yes. For the limiation, the authors stated that the dataset includes some noisy labels.

**Opportunities For Improvement:**

- In the abstract, the authors hypothesized that image classifiers can reach high accuracy by first zooming the most discriminative region and then extract features for the predictions. I understand that zoom-in transforms improve image classification. However, I do not think that such a fact directly indicates that the model first zooms the most discriminative region and extracts relevant features inside the model. In other words, it may be true or it may not. The authors did not demonstrate this point with details after the abstract. Further analysis on such a part or removing such a claim is highly recommended.
- Regarding the terminology of test-time augmentation (TTA), I think it may be confusing for the ones who are familiar with test-time adaptation for TTA. I think TTA is widely used for test-time adaptation, not test-time augmentation.
- The “random” baseline in Table 1 is not specified in detail. Is it a randomly initialized model? It was hard to understand Table 1 since the details on “random” were not specified.
- Regarding the construction of ImageNet-Hard, I have the following two questions. First, why do groups A and B have different numbers of students? Is it intentionally designed to have different numbers of students for cross validation? Second, in Lines 275, 276, the authors stated that they discarded labels which overlap with each other. How did the authors decide if classes overlap with each other or not? In fact, Figure 4 (a) contains the results of Common classes. To the best of my knowledge, these classes are grouped as Common classes since they seem similar but they are still contained in the dataset. Why did the authors not exclude the Common classes while discarding the similar ones that were mentioned in Lines 275, 276. How are they different and how did the authors identify them?

**Relation To Prior Work:**

Yes

**Summary And Contributions:**

In this work, the authors proposed ImageNet-Hard dataset, which includes images that models fail to classify even with diverse zoom-in transforms. The main observation of this proposed benchmark is that simply tuning the zoom-in augmentation significantly improves the performance of ImageNet-related test sets. The authors additionally demonstrated that zoom-in augmentations improve the performance of the current state-of-the-art test-time augmentation method named MEMO.

---

> ### Author Response · Authors · 2023-08-09
> **Response to Reviewer pbFv - Part 1/2**
>
> Thank you for your valuable questions! In light of your feedback, we have improved the manuscript.
> We think our answers have addressed your concerns. If you have any follow-up questions, we'd be happy to address them as well!
>
> > In the abstract, the authors hypothesized that image classifiers can reach high accuracy by first zooming the most discriminative region and then extract features for the predictions. I understand that zoom-in transforms improve image classification. However, I do not think that such a fact directly indicates that the model first zooms the most discriminative region and extracts relevant features inside the model. In other words, it may be true or it may not. The authors did not demonstrate this point with details after the abstract. Further analysis on such a part or removing such a claim is highly recommended.
>
> Thank you for the great point!
>
> First, we acknowledge that understanding the decision-making process of an existing classifier is an active Explainable AI (XAI) research topic.
> So far, while there is no single, definitive method to use, Class Activation Map (CAM) and GradCAM are among the most popular methods to visualize which image areas a classifier looks at when make a certain prediction.
> We have added `Fig. 4` to the main text and `Sec. F4` in Appendix to show how a ResNet-50 (after finetuning using MEMO + RRC) has better accuracy and also localizes objects better (via GradCAM).
>
> Second, we have revised the hypothesis paragraph in the Abstract and Introduction to clarify our point. The paragraph now reads as the following:
>
> _When processing an image, a model may implicitly zoom in or out (defined in `Sec. 3`) to the most discriminative image region ignoring the rest of the image (`Fig. 1a`), and then extract that localized region's features to predict image labels.
> We hypothesize that the improved image classification may largely be due to the networks accurately zooming to the discriminative areas (e.g., junco and magpie birds in `Fig. 1`) rather than more accurately describing them (i.e., generating better features)._
>
> In this paper, we show supporting evidence for our hypothesis.
> For example, using the same old AlexNet feature extractor, one can reach a high accuracy (using 36 zoom crops) of almost `90%` on ImageNet.
> In other words, we are able to show that the zooming hypothesis is a **viable** mechanism to improve accuracy.
> Yet, future XAI research is needed to fully disentangle the two hypotheses (better _zooming_ to regions/objects vs. better _features_ of regions/objects).
>
>
> > Regarding the terminology of test-time augmentation (TTA), I think it may be confusing for the ones who are familiar with test-time adaptation for TTA. I think TTA is widely used for test-time adaptation, not test-time augmentation.
>
> Thank you for your feedback! We acknowledge that the abbreviation "TTA" can refer to both Test-Time Adaptation and Augmentation, which might confuse readers. We used this jargon following the convention established in previously published papers, such as MEMO (NeurIPS 2022).
>
> > The “random” baseline in Table 1 is not specified in detail. Is it a randomly initialized model? It was hard to understand Table 1 since the details on “random” were not specified.
>
> Yes. The purpose of this **random** baseline is to provide a measurement for a hypothetical classifier that randomly selects labels from the target label distribution. In a 1000-way classification, for example, a given crop has 1/1000 chance to be labeled correctly by a random classifier. So for a total of 324 crops, the upper-bound accuracy would be $324 \times 1/1000 $ = 32.4%.
>
> Similarly, for the 200-way image classification benchmark of ImageNet-A, the upper-bound accuracy would be $324 \times 1/200$, which is larger than 100% so we set it at 100%.
> The purpose of having a random baseline is to contrast well-known classifiers with the random classification behaviors and better interpret the upper-bound accuracy. For example, on ImageNet, all classifiers' upper-bound accuracy is way above the random baseline (32.4%).
> In contrast, on ImageNet-A, they are way below the random baseline (100%), suggesting that most classifiers do not perform randomly but instead are biased towards the content inside the crops.
>
> > Why do groups A and B have different numbers of students? Is it intentionally designed to have different numbers of students for cross validation?
>
> We organized the volunteer participants into two distinct groups, depending on their availability and the level of effort they were willing to contribute to the labeling process. Given the substantial effort required to label all images (approximately 13K samples), one group committed to handling the entire dataset, while the other agreed to label at least 50 images per person.  The cross-validation setup we used was designed to maximize the agreement between pairs of participants based on their participation.

---

> ### Author Response · Authors · 2023-08-09
> **Response to Reviewer pbFv - Part 2/2**
>
> > Second, in Lines 275, 276, the authors stated that they discarded labels which overlap with each other. How did the authors decide if classes overlap with each other or not? In fact, Figure 4 (a) contains the results of Common classes. To the best of my knowledge, these classes are grouped as Common classes since they seem similar but they are still contained in the dataset. Why did the authors not exclude the Common classes while discarding the similar ones that were mentioned in Lines 275, 276. How are they different and how did the authors identify them?
>
> Our decision to remove classes such as `projectile` and `missile` came after recognizing a lack of discernible distinction between images within these categories, a pattern that was consistent even within the training set samples. In many instances, the categories of `projectile` and `missile` appeared very similar, making them virtually indistinguishable in some cases and even in the training set. This decision was made without regard to any specific classifier. The analysis conducted concerning `Rare` and `Common` misclassifications focused specifically on errors made by `EfficientNet-L2` classifier. Since these analyses are dependent on the classifier, different models may yield varying results. Additionally, related data for both types of misclassification is [available](https://github.com/taesiri/ZoomIsAllYouNeed/tree/main/src/ImageNet_Hard/gpt-response) for future reference and  usecases.

---

> > ### Comment · Reviewer_pbFv · 2023-08-25
> > **Most of my issues are addressed and happy to go forward for this paper**
> >
> > I deeply appreciate the authors' feedback and I believe that most of my concerns are addressed, and this paper deserves to be accepted.
> > I raised my rating from 6 to 7.

---

### Official Review · Reviewer_twuE · 2023-07-20
**A hard ImageNet benchmark to challenge models even with zooming-in**

**Rating:** 5
**Confidence:** 3
**Correctness:** 1. A claim about "the model implicitl…
**Clarity:** Yes, the paper is well-written.

**Strengths:**

- The introduces an interesting test-time augmentation technique based zooming operations.
- The paper is well-written.

**Additional Feedback:**

"augmented images" on line 90 should be "crops" or "patches" to be more specific, if my understanding is correct.

**Documentation:**

Yes.

**Ethics:**

No.

**Limitations:**

1. A paper's hypothesis that "the model implicitly zooms in to the most salient region ignoring the rest" (Line 23-24) is not validated. The experiments only explicitly feed and train the models on the zoomed-in images/crops, which does not substantiate the claim. To validate this hypothesis, the paper needs to let the model learn an entire image and conduct experiments to test if the model indeed only zooms in a salient region of an image.

2. A major finding in this paper is that the model's accuracy improves when using a good zooming strategy. However, this finding has been studied in several works [1*,2*,18, 33, 69] and is not a surprising fact.

3. Another finding in this paper is that ImageNet-A and ObjectNet has a center bias. This finding is also trivial, as the image center always contains the main object of interest. Indeed, the community has usually used center-cropped as their standard data augmentation for the image classification problem.

4. The zooming strategy can be non-optimal. It considers a discrete zoom-crop space with 324 configurations. There could exist a case where the target object is not completely contained in one of the crop. Even the experiments consider 36 scales, how can  we ensure that the parts of the object are not cut off after cropping.  As such, Fig. 2 shows that the failed examples are from  the crops with incomplete objects.
The proposed zoom strategy may not lead to the optimal accuracy. This can invalidate conclusions about "the type of image that cannot be correctly labeled", "the model's accuracy on zoomed-cropped images", and the "selection of hard examples for the proposed ImageNet-Hard dataset".

5. To verify whether the proposed ImageNet-Hard is challenging with even zooming operations, the paper should evaluate zooming-based image classification methods such as [18, 33, 69]. This will set a competitive benchmark for the community to develop more advanced learning-to-zoom techniques.

5. From Table 1, the 100% accuracy from Random baseline on IN-A, IN-S and ON is strange and needs an explanation.

6. To verify whether
[1*] Learning to Zoom: a Saliency-Based Sampling Layer for Neural Networks , ECCV2018.
[2*] Zoom to Learn, Learn to Zoom, CVPR2019.

**Opportunities For Improvement:**

See Limitations section.

**Relation To Prior Work:**

The paper should discuss how a major findings about "zooming increases model performance" is different from the discoveries of recent studies [1*,2*,18, 33, 69].

[1*] Learning to Zoom: a Saliency-Based Sampling Layer for Neural Networks , ECCV2018.
[2*] Zoom to Learn, Learn to Zoom, CVPR2019.

**Summary And Contributions:**

The paper studies the effects of zooming on image classification model's accuracy. It presents a finding about a center bias on ImageNet-A and ObjectNet. The paper proposes a test-time augmentation technique to select confident model predictions on different zoomed-in crops. Finally, a new ImageNet-Hard benchmark is presented to challenge image classifiers even when applying zooming effects.

---

> ### Author Response · Authors · 2023-08-09
> **Response to Review twuE - Part 1/3**
>
> Thank you for your valuable questions! In light of your feedback, we have improved the manuscript.
>
> We think our answers have addressed your concerns. If you have any follow-up questions, we'd be happy to address them as well!
>
>
> > 1. A paper's hypothesis that "the model implicitly zooms in to the most salient region ignoring the rest" (Line 23-24) is not validated.
>
> Thank you for the great point!
>
> First, we acknowledge that understanding the decision-making process of an existing, **pre-trained** classifier is an active Explainable AI (XAI) research topic.
> So far, while there is no single, definitive method to use, Class Activation Map (CAM) and GradCAM are among the most popular methods to visualize which image areas a classifier uses when making a certain prediction.
> We have added `Fig. 4` to the main text and `Sec. F4` in Appendix to show how a ResNet-50 (after finetuning using MEMO + RRC) has better accuracy and also localizes objects better (via GradCAM).
>
> Second, we have revised the hypothesis paragraph in the Introduction to clarify our point. The paragraph now reads as:
>
> _When processing an image, a model may implicitly zoom in or out (defined in `Sec. 3`) to the most **discriminative** image region, ignoring the rest of the image (`Fig. 1a`), and then extract that localized region's features to predict image labels.
> We hypothesize that the improved image classification may largely be due to the networks accurately zooming to the discriminative areas (e.g., junco and magpie birds in `Fig. 1`) rather than more accurately describing them (i.e., generating better features)._
>
> In this paper, we show supporting evidence for this hypothesis.
> For example, using the same old AlexNet feature extractor, one can reach a high accuracy (using 36 zoom crops) of almost `90%` on ImageNet.
> In other words, we are able to show that the zooming hypothesis is a **viable** mechanism to improve accuracy.
> Yet, future XAI research is needed to fully disentangle the two hypotheses (better _zooming_ to regions/objects vs. better _features_ of regions/objects).
>
>
> > 2. A major finding in this paper is that the model's accuracy improves when using a good zooming strategy. However, this finding has been studied in several works [1*,2*,18, 33, 69] and is not a surprising fact.
>
> Thank you for the references! Please let us clarify our novelty.
>
> - First, none of the works in [1*,2*,18, 33, 69] studies the effects of zoom-**in and out** on ImageNet and ImageNet-scale OOD benchmarks as our work does. They only focus on fine-grained image classification (e.g., bird classification) where zoom-in is known to help.
> - Second, we have examined the positional bias using a technique that we found to be effective, allowing us to compare different datasets.
> - Third, we are the first to study the upper-bound accuracy of well-known classifiers when optimal zooming is given. We find surprising findings that even a 10-year-old AlexNet can classify up to 90% of images given the correct framing of the input image.
>
> Contrary to the method described in [1*], our study provides a comprehensive analysis of the implications of utilizing zooming techniques for image classification tasks. We focus particularly on the ImageNet and ImageNet-scale datasets, diverging from the approach in (1), which introduces a method to distort an image for fine-grained classification. The study detailed in [2*] concentrates on improving image super-resolution, especially within the context of computational zoom, which is not related to our study.
>
>
> > 3. Another finding in this paper is that ImageNet-A and ObjectNet has a center bias. This finding is also trivial, as the image center always contains the main object of interest.
>
> If this was true for all the datasets, then we should have found a similarly strong center bias in all 6 benchmarks tested. Yet, this was not the case (only ImageNet-A and ObjectNet were found to contain a strong center bias).
>
> > The community has usually used center-cropped as their standard data augmentation for image classification.
>
> Center-crop is a common image transformation for classifiers, while center-bias is a property of a **dataset**. Not all images from various datasets have their objects at the center of the image (for example, https://zoom.taesiri.ai/static/gifs/animation-2.gif).
>
> Our study quantifies properties like center bias in OOD datasets and offers insights into how simple changes, like upscaling images, can affect accuracy across various classifiers. Our findings have significant implications for both train/test time augmentation and for dataset construction. From a dataset perspective, our approach does not rely on bounding box information, allowing us to evaluate positional bias easily on any dataset. For example, on ImageNet-A, using insight from the center-bias analysis, the accuracy could be increased from 0% to 15% by merely changing the initial zoom scale (Please refer to `Sec. B9` and `Fig. A14` in the appendix).

---

> > ### Comment · Reviewer_twuE · 2023-08-20
> >
> > I appreciate the authors' efforts in clarifying the concerns.
> > I agree on the responses addressing Issue 1 and 2.
> >
> >
> > Regarding Issue 3, the "stronger" center bias of ImageNet-A and ObjectNet does not bring much practical values to the community.
> > Figures in B.2 indicate that other datasets such as ImageNet-Real, ImageNet-R and ImageNet-Sketch also exhibit center bias, i.e., the accuracy on the center crop is always the highest. Even though the center bias is not as strong, it still indicates that "cropping the object at the center will lead to the a higher accuracy".
> > Yet, knowing that ImageNet-A and ObjectNet has a _stronger_ center bias doesn't add further values. Computer vision practitioners still perform center crop to optimize the accuracy regardless of ImageNet-A with a stronger center bias or ImageNet-Real with a lower center bias.

---

> > > ### Author Response · Authors · 2023-08-23
> > >
> > > > Regarding Issue 3, the "stronger" center bias of ImageNet-A and ObjectNet does not bring much practical values to the community. Figures in B.2 indicate that other datasets such as ImageNet-Real, ImageNet-R and ImageNet-Sketch also exhibit center bias, i.e., the accuracy on the center crop is always the highest. Even though the center bias is not as strong, it still indicates that "cropping the object at the center will lead to the a higher accuracy". Yet, knowing that ImageNet-A and ObjectNet has a stronger center bias doesn't add further values. Computer vision practitioners still perform center crop to optimize the accuracy regardless of ImageNet-A with a stronger center bias or ImageNet-Real with a lower center bias.
> > >
> > >
> > >
> > > We thank the reviewer for their effort and scrutiny put into the review process and appreciate their feedback.
> > >
> > > ImageNet-A and ObjectNet are both well-established benchmarks in the computer vision community. They position themselves as challenging benchmarks that test the generalizability and robustness of image classifiers. According to Google Scholar, ImageNet-A and ObjectNet have been cited 800 and 378 times, respectively, as of the time of writing this response. However, we demonstrated that these two datasets present challenges for another reason: positional bias and the scale of the object of interest in the original image (of course there is a compound effect here). These findings have significant implications for future efforts in dataset construction. We proposed a general, bounding-box-free approach that is valuable for measuring such metrics across various datasets.
> > >
> > > Particularly, for ImageNet-A, we demonstrated that it's possible to achieve an accuracy range of `16.1-29.7%` using various ResNet-50 classifiers (please refer to `Sec. C3`, `Tab. A10`) with a simple trick (upscale the image then center crop). These numbers surpass those of state-of-the-art methods like Robust ViT (RVT) [R1] (`14.4-28.5%`).
> > >
> > > **A side note about our contribution and scope of NeurIPS Track Dataset and Benchmark**: Our paper closely aligns with the data-centric machine learning research (DMLR) focus of the NeurIPS Track Dataset and Benchmark, particularly the segment emphasizing "**[audits of existing datasets and identifying significant problems within them](https://neurips.cc/Conferences/2023/CallForDatasetsBenchmarks).**" Through our  examination of well-known datasets such as ImageNet-A and ObjectNet, we showed pronounced positional biases, a finding that has not been highlighted before. Additionally, our approach in creating a new dataset stands as a novel contribution to the current methodologies in dataset construction.
> > >
> > >
> > > ### References:
> > > ```
> > > [R1] Mao, Xiaofeng, et al. "Towards robust vision transformer." Proceedings of the IEEE/CVF conference on Computer Vision and Pattern Recognition. 2022.
> > > ```

---

> ### Author Response · Authors · 2023-08-09
> **Response to Review twuE - Part 2/3**
>
> > 4. The zooming strategy can be non-optimal. It considers a discrete zoom-crop space with 324 configurations. There could exist a case where the target object is not completely contained in one of the crop. Even the experiments consider 36 scales, how can we ensure that the parts of the object are not cut off after cropping. As such, Fig. 2 shows that the failed examples are from the crops with incomplete objects. The proposed zoom strategy may not lead to the optimal accuracy. This can invalidate conclusions about "the type of image that cannot be correctly labeled", "the model's accuracy on zoomed-cropped images", and the "selection of hard examples for the proposed ImageNet-Hard dataset".
>
> We agree that some zoom crops may contain incomplete objects. However, we do not understand why exactly this strategy must be optimal w.r.t. some objective.
> The point is exactly the opposite---with even such a naive zooming strategy, we find AlexNet's upper-bound accuracy can reach nearly `90%` already.
>
> Please note that we also include the zoom-**out** group and have also included a sample in the middle row of `Fig. 1(a)`, where we demonstrate that zooming out is necessary to properly frame the object and view it correctly. However, generally, classifiers can detect objects from incomplete crops or even solely from background information (please refer to [R1] and `Sec. C1` in the appendix). When we consider 36 crops, we identify the most effective set containing 36 zoom transforms that yield the highest accuracy, using a greedy search algorithm.
>
> **Regarding Fig. 2:** This figure illustrates that when we utilize all 324 crops and all IN-trained classifiers, there exists a set of approximately 200 images that remain unclassifiable. We manually investigated this set to report its properties.
>
> **Regarding the sampling technique,** as mentioned in `Sec. 4.4` of the paper, there are several methods for constructing an image dataset. Existing ImageNet-scale benchmarks adhere to one of three construction methods:
>
> - Perturbing real images with the intention of increasing their difficulty (e.g., ImageNet-C or DamageNet).
> - Collecting real images based on model misclassifications (e.g., ImageNet-A).
> - Implementing a highly controlled data collection process (e.g., ImageNet-S and ObjectNet).
>
> Nevertheless, none of these benchmarks explicitly challenge models to recognize a properly framed object within an image. Therefore, we propose a new method and test it with 324 crops. This is a versatile approach that can also be implemented with any number of crops, whether it's 36 or 1,000. The novelty of this method lies in its introduction of a new way to construct a dataset.
>
> The rationale for using CLIP and 324 crops is derived from our empirical findings, suggesting that 324 attempts using a CLIP classifier can achieve nearly perfect accuracy, almost reaching 100%, though never exactly 100%.
>
> Further in the study, we also demonstrate how to enhance this subset by utilizing responses from the GPT-3.5 model to filter out *Common* and *Rare* misclassifications - instances where the target and predicted classes are considerably distant or very close. This method offers a generic approach that the community can adopt to construct new datasets.
>
>
> > 5. To verify whether the proposed ImageNet-Hard is challenging with even zooming operations, the paper should evaluate zooming-based image classification methods such as [18, 33, 69]. This will set a competitive benchmark for the community to develop more advanced learning-to-zoom techniques.
>
>
> Thank you for your feedback and suggestions! Indeed it is valuable to evaluate the performance of zooming-based methods  [18, 33, 69] on ImageNet-Hard. The methods in [18, 33, 69] are primarily designed for fine-grained classification and may not be applicable at the scale of ImageNet. For instance, the Spatial Transformer Network [R2] has only been tested on datasets like MNIST [R3] and CUB [R4].
>
> Furthermore, we have revised the title of `Sec. 4.4` to eliminate confusion regarding the claim of the impracticality of zooming on ImageNet-Hard. The ImageNet-Hard dataset is constructed using a novel sampling technique that we proposed. This technique focuses on the difficulties classifiers encounter in localizing or framing objects. For ImageNet-Hard, we directly employed results from our empirical analysis, demonstrating that while CLIP can achieve very high accuracy with zooming, it never reaches 100%.

---

> > ### Comment · Reviewer_twuE · 2023-08-20
> >
> > Regarding Issue 4, the ability to search for an optimal zoom-crop is important.
> > In the proposed discrete zoom-crop strategy, it can miss an optimal zoom-crop (including more complete view of an object), yielding the accurate classification of such as image example. Hence, we cannot conclude that "an image type that cannot be correctly labeled with an optimal zooming strategy" if we have not extracted an optimal zoom-crop from that image. It can only be concluded that those are images that cannot be classified with a discrete zooming strategy. Similarly, the finding about "selection of hard examples for the proposed ImageNet-Hard dataset" is also biased to the discrete zooming strategy.
> >
> > This links with Issue 5. Learning-to-zoom methods [18, 69], especially [69], learn an optimal zoom strategy (optimize via a loss function) to look at an arbitrary image regions for maximizing the accuracy. Such methods can figure out a more optimal zoom-crop for those "selected hard examples" in the ImageNet-Hard dataset, where the discrete zooming strategy could miss.
> > Hence, results from [18, 69] are needed to (1) present a competitive benchmarks for future research, and (2) verify if a discrete zooming strategy can already find out a sufficient zoom-crop to detect hard examples for ImageNet-Hard.

---

> > > ### Author Response · Authors · 2023-08-23
> > >
> > > > Regarding Issue 4, the ability to search for an optimal zoom-crop is important....
> > >
> > > We think there is some confusion regarding the phrase "optimal zoom" and whether a crop $C$ exists that fully contains an object in the image. **We show that at least 99% of the time, such a crop exists.**
> > >
> > > Suppose we have an object placed in an image region with size $(W, H)$, located at $(A, B)$ in the original image's coordinates. It is easy to calculate the coordinates and sizes of the object after scaling, as well as the anchor point location and boundaries of all 9 crops. Using this setup, we ran an analysis on 10,000 images with dimensions similar to ImageNet images (please refer to our response to reviewer `JuE9` [here](https://openreview.net/forum?id=Cf2c9Pk9yF&noteId=lsOTBU2Eca)), containing objects of random sizes and coordinates within the image. We found that **`99.4%` of the time, there was at least one crop that encompassed the entire object/patch**. You can view the mathematical equations, the code, and run the analysis using this [Google Colab](https://colab.research.google.com/drive/1hF6A3iwWUyWuO2WUn06uCaE2tDi7H7VC?usp=sharing). This confirms that the likelihood of occurrences where no crop contains the object is quite low.
> > >
> > > ---
> > >
> > > >  This links with Issue 5. Learning-to-zoom methods [18, 69], especially [69], learn an optimal zoom strategy (optimize via a loss function) to look at an arbitrary image regions for maximizing the accuracy. Such methods...
> > >
> > > We would like to highlight some critical points about dataset construction and ImageNet-Hard dataset:
> > >
> > > - **Transparency in Dataset Construction:** The importance of a transparent method for dataset construction. If we use a black box to filter out series of images, we would end up with an inexplicable selection process.
> > >
> > > - **Flexibility of ImageNet-Hard Construction**: ImageNet-Hard is an instance of a dataset construction recipe. That is, you can choose any classifier, any zoom level configuration (either discrete or automated method), and then filter massive datasets to retain challenging images.
> > >
> > > - **Performance Benchmark**: `Tab. 3` in the paper clearly shows ImageNet-Hard is already a challenging dataset for many classifiers, **Even vision-language models find ImageNet-Hard challenging**.
> > >
> > >
> > > **Regarding RA-CNN [18] and PatchDrop [69]**
> > >
> > > We have attempted our best to leverage your suggested models; however, unfortunately, we think these models do not help our study or increase the difficulty of ImageNet-Hard as we hoped. Here is why:
> > >
> > > - **RA-CNN** [18]: This is a multi-scale recurrent neural network that progressively zooms into finer discriminative regions for fine-grained image recognition. The process this model uses to sample the next region is similar to a saliency-based _Resize and Crop_.
> > >
> > > - **PatchDrop** [69]: This method **selectively samples high-resolution patches** from the full-resolution image with the key motivation of **reducing computation while maintaining accuracy**. This **method does not zoom in on arbitrary image locations**; rather, it samples image patches of size $56 \times 56$ from a full-resolution image. The process that the policy network uses is similar to _Masked Image Modeling_.
> > >
> > >
> > > Neither of these methods is compatible with the dataset and classifier configuration we used. **RA-CNN does not work on ImageNet**, and the **weights for PatchDrop are not available** (the codebase is 3 years old).
> > >
> > >
> > > We attempted to replicate [18] and [69] without training an optimal _policy_ network to verify ImageNet-Hard will remain challenging.  We measured the _maximum possible accuracy_ in a manner similar to how each of these methods operates. Specifically, we used CLIP to repeat the experiment in `Sec. 1` to calculate the _maximum possible accuracy_ using both _RandomResizedCrop_ (RRC) and _Random Image Masking_ (Masking) to **simulate** [18] and [69] respectively. Since the maximum possible accuracy provides an upper bound accuracy for a given classifier, this metric can reflect the performance of *an ideal* classifier of these kinds.
> > >
> > > Our results on 1,000 random images from ImageNet-Hard show that the maximum possible accuracy using either of **these methods can only reach ~23% (Masking) and ~28% (RRC)**. We used 324 different attempts for each method (324 different mask configurations and 324 different crops) and calculated the maximum possible accuracy. This is in clear contrast with the results we have in `Tab. 1` for various classifiers, showing that CLIP can easily reach `>90%`. This **solidifies** the argument that dataset construction such as we did for ImageNet-Hard is valid and results in a challenging dataset, **even given a hypothetical optimal zoom method**. You can find the code for this analysis on [Google Colab](https://colab.research.google.com/drive/1n_IAeLaND7xYQf10s7Yl-z3_1mimcIcx?usp=sharing).

---

> > > > ### Comment · Reviewer_twuE · 2023-08-28
> > > >
> > > > Thanks the authors for clarifying my concerns about zoom-crop strategies and experiments with zooming networks. I raised my rating from 4 to 5. I still have a minor concern about center bias, which is not a novel finding; thus, I couldn't raise my rating higher.

---

> ### Author Response · Authors · 2023-08-09
> **Response to Review twuE - Part 3/3**
>
> > 6. From Table 1, the 100% accuracy from Random baseline on IN-A, IN-S and ON is strange and needs an explanation.
>
> The purpose of this **random** baseline is to provide a measurement for a hypothetical classifier that **randomly selects** labels from the target label distribution. In a 1000-way classification, for example, a perfect random classifier would pick one label randomly from 1000 different possible labels (1/1000 chance of success on each try). This concept becomes relevant when dealing with a large number of image crops and a smaller number of target classes.
>
> Take ImageNet-A, which involves 200-way classification. For each crop, the random chance to hit the correct label is $1/200$.
> For a total of 324 crops, the upper-bound accuracy would be $324 \times 1 / 200 $ so we use 100%.
> Please note that all tested classifiers perform very differently from a random classifier (i.e., they have an upper-bound accuracy much lower than 100%).
>
>
> > "augmented images" on line 90 should be "crops" or "patches" to be more specific, if my understanding is correct.
>
> Thank you for your feedback!
>
> We have incorporated your suggestion into our writing to be more clear.
>
>
> ## References
>
> ```
> [R1] Xiao, Kai, et al. "Noise or signal: The role of image backgrounds in object recognition." arXiv preprint arXiv:2006.09994 (2020).
>
> [R2] Jaderberg, Max, Karen Simonyan, and Andrew Zisserman. "Spatial transformer networks." Advances in neural information processing systems 28 (2015).
>
> [R3] LeCun, Yann. "The MNIST database of handwritten digits." http://yann. lecun. com/exdb/mnist/ (1998).
>
> [R4] Wah, Catherine, et al. "The caltech-ucsd birds-200-2011 dataset." (2011).
> ```

---

> > ### Comment · Reviewer_twuE · 2023-08-20
> >
> > Thanks for your clarification. This observation is not obvious. The authors could consider clarifying clearer in a concise way in the paper, or explaining this in the Supplementary Materials.

---

> > > ### Author Response · Authors · 2023-08-23
> > >
> > > > Thanks for your clarification. This observation is not obvious. The authors could consider clarifying clearer in a concise way in the paper, or explaining this in the Supplementary Materials.
> > >
> > >
> > > Thank you for your comment, we will update the manuscript to provide more details about this.

---

### Official Review · Reviewer_3Lfp · 2023-07-22
**Interesting paper that re-examines classification abilities of existing models by adjusting what parts of images that look at**

**Rating:** 7
**Confidence:** 4
**Clarity:** The paper is well written.

**Strengths:**

- The paper is well-motivated.
- The findings are novel and the paper offers several interesting insights on both model behavior as well as dataset biases.
- The results are significant for re-interpreting existing benchmarks (e.g. ObjectNet) and for the construction of future ones.

**Additional Feedback:**

The abstract proposes a hypothesis that “one way for image classifiers to reach high accuracy is to first zoom to the most discriminative region in the image” but my impression is that this hypothesis is still not sufficiently answered in the paper. The zooming experiments only provide an upper bound of what performance models can achieve given the right input scale, but don’t really tell us how much models are relying on better features versus better zoom in their classification. Do the authors have any thoughts on how to better disentangle these?

**Correctness:**

The claims made in the paper are sound. Most claims are well supported with empirically evidence (see “Opportunities For Improvement” for more details).

**Documentation:**

The paper uses publicly available datasets, and sufficient details are provided to recreate the benchmark proposed (ImageNet-Hard).

**Limitations:**

The authors have adequately addressed the limitation of their work.

**Opportunities For Improvement:**

- It would be good to report % change in accuracy (e.g. by using 324 crops instead of 1 crop) in table 1.
- The authors mention that for MEMO, replacing AugMix with their zoom-in functions improves performance by 0.28 on average, and by 1.10 on IN-A. Since this augmentation technique involves fine-tuning a pre-trained classifier, I believe reporting error bars (and number of random seeds) would make the improvement more convincing.
- The findings of the paper are interesting, especially from a dataset and model understanding perspective. To improve the relevance of the results in practice, the paper should also demonstrate how the ranking/ order of performance for a wide range of architectures would change given the new zooming operation.
- Regarding the question of whether “image classifiers’ evolution over the past ten years is about mastering where and at what scale to zoom”, a prior work (Do Vision Transformers See Like Convolutional Neural Networks?) has touched on how modern classifiers solve classification differently from older architectures. The authors may want to discuss this work in “Discussion” as well.

**Relation To Prior Work:**

Related work is thoroughly covered.

**Summary And Contributions:**

This work proposes a zooming hypothesis: image classifiers attain high performance by first (implicitly) zooming into certain regions of the image. Using existing models, the authors demonstrate substantial gains on ImageNet just by adjusting the framing of the input images. By examining which zooming setting leads to the most performance gains on different datasets, the paper finds that ImageNet-A and ObjectNet exhibit a strong center bias. Finally, the paper proposes a test-time augmentation method that can offer performance gains by making models zoom in before making predictions. The images from ImageNet-derived benchmarks that OpenAI’s CLIP fails to classify even after zooming operations then form the ImageNet-Hard benchmark.

---

> ### Author Response · Authors · 2023-08-09
> **Response to Reviewer 3Lfp - Part 1/3**
>
> Thank you for your valuable, constructive feedback! We have revised the manuscript in light of your comments.
>
> Please find our inline answers below. We appreciate any follow-up requests and feedback to make our work better.
>
>
> > It would be good to report % change in accuracy (e.g. by using 324 crops instead of 1 crop) in table 1.
>
> Thank you for your suggestion!
>
> We added a new table (`Tab. A14`) in the appendix to show improvements (percentage points) for both the 36 and 324 crop size. Due to certain model dataset configurations (such as ResNet-50 on ImageNet) achieving an accuracy of near zero, the percentage of change (%) can reach more than 31,000. As a result, we are only reporting the _absolute difference_ (i.e., delta in % accuracy) between the baseline and new values.
>
> > The authors mention that for MEMO, replacing AugMix with their zoom-in functions improves performance by 0.28 on average, and by 1.10 on IN-A. Since this augmentation technique involves fine-tuning a pre-trained classifier, I believe reporting error bars (and number of random seeds) would make the improvement more convincing.
>
> Thank you very much for your feedback and suggestions!
>
> We added `Sec. F3` to the appendix that offers more detailed information about the performance of MEMO on both ImageNet and ImageNet-A. Due to the computational power required to run AugMix experiments, we have repeated the experiments for ImageNet and ImageNet-A, providing error bars for all three ResNet-50 models. The results remain consistent and demonstrate that `RandomResizedCrop`, despite being only one transform, outperforms AugMix (which is a combination of multiple transforms):
>
>
> ### Results for ImageNet
>
> Using 3 random seeds
>
> | Model Type     | Aug. Type | $\mu$| $\sigma$ |
> |----------------|----------|--------------------|-------------------|
> | ResNet-50      | AugMix   | 77.23              | 0.05              |
> | ResNet-50      | RRC (**ours**)      | **77.56**          | 0.02              |
> | DeepAug+AugMix | AugMix   | 76.29              | 0.03              |
> | DeepAug+AugMix | RRC (**ours**)      | **76.41**          | 0.03              |
> | MoEx+CutMix    | AugMix   | 79.31              | 0.01              |
> | MoEx+CutMix    | RRC (**ours**)     | **79.50**          | 0.02              |
>
>
> ### Results for ImageNet-A
>
> Using 3 random seeds
>
> | Model Type     | Aug. Type | $\mu$| $\sigma$ |
> |----------------|----------|--------------------|-------------------|
> | ResNet-50      | AugMix   | 0.77               | 0.02              |
> | ResNet-50      | RRC (**ours**)      | **1.29**           | 0.05              |
> | DeepAug+AugMix | AugMix   | 5.27               | 0.05              |
> | DeepAug+AugMix | RRC (**ours**)     | **5.61**           | 0.06              |
> | MoEx+CutMix    | AugMix   | 10.63              | 0.06              |
> | MoEx+CutMix    | RRC (**ours**)  | **13.39**          | 0.20              |

---

> ### Author Response · Authors · 2023-08-09
> **Response to Reviewer 3Lfp - Part 2/3**
>
> > The findings of the paper are interesting, especially from a dataset and model understanding perspective. To improve the relevance of the results in practice, the paper should also demonstrate how the ranking/ order of performance for a wide range of architectures would change given the new zooming operation.
>
>
> Thank you for your suggestions!
>
> We included a new section in the appendix (`Sec. F2`) to provide results for modern architectures, including ConvNeXT [R1], Swin Transformer [R2], and Max-ViT [R3]. Due to the needed computational resources, we conducted the study on ImageNet, ImageNet-Real, and ImageNet-A. Our ranking analysis shows Max-ViT as the consistent top performer across all experiments. We also performed a rank analysis in `Fig. A62`, and its results show that ConvNeXT, despite having high 1-crop accuracy, did not maintain its rank when more crops were included. ViT-B/32 displayed a reverse trend; although its 1-crop rank was not strong, the addition of more crops helped it surpass the accuracy of ConvNeXT.
>
> Additionally, in the paper, we introduced ImageNet-Hard as a new benchmark. We report the accuracy of a wide range of OpenCLIP models on ImageNet-Hard in `Sec. E6` of the appendix. These results also highlight the utility of ImageNet-Hard in ranking different models, as it correctly ranks ViT-bigG-14 and DataComp-ViT-L/14 [R4] close to each other and shows the effectiveness of the data filtering process proposed in DataComp.
>
>
> ### Ranking Results
>
> Average rankings of different classifiers on ImageNet, ImageNet-ReaL, and ImageNet-A are as follows:
>
> | Classifier        | 1-Crop Rank | 36-Crop Rank | Max Possible Rank |
> |-------------------|-------------|--------------|-------------------|
> | MaxViT-T          | 1.67        | 1.00         | 1.00              |
> | ConvNext-Base     | 1.67        | 3.00         | 4.33              |
> | Swin-B            | 2.67        | 2.00         | 2.33              |
> | EfficientNet-B0   | 4.33        | 5.00         | 5.00              |
> | MobileNetV3-Small | 6.67        | 7.00         | 7.00              |
> | ResNet-50         | 6.00        | 6.00         | 5.33              |
> | ViT-B/32          | 5.00        | 4.00         | 3.00              |
>
>
> ### Accuracy Results
>
>
> | (a) Standard top-1 accuracy based on $N = 1$ crop | ImageNet | ImageNet-ReaL | ImageNet-A |
> |-----------------|-----------|----------------|------------|
> | **Classifier**  | ImageNet  | ImageNet-ReaL  | ImageNet-A |
> | MobileNetV3-Small | 67.42     | 74.47          | 2.73       |
> | ResNet-50        | 75.74     | 82.63          | 0.17       |
> | EfficientNet-B0  | 77.69     | 84.13          | 7.01       |
> | ViT-B/32         | 75.75     | 81.89          | 9.51       |
> | ConvNext-Base    | 83.49     | 88.19          | 33.99      |
> | Swin-B           | 83.37     | 87.80          | 34.87      |
> | MaxViT-T         | 83.49     | 88.05          | 36.00      |
>
> | (b) Upper-bound accuracy using $N = 36$ crops | ImageNet | ImageNet-ReaL | ImageNet-A |
> |-----------------|-----------|----------------|------------|
> | MobileNetV3-Small | 91.79     | 95.63          | 44.45      |
> | ResNet-50        | 94.47     | 97.35          | 54.97      |
> | EfficientNet-B0  | 94.87     | 97.49          | 59.33      |
> | ViT-B/32         | 95.04     | 97.60          | 68.33      |
> | ConvNext-Base    | 95.78     | 97.68          | 82.25      |
> | Swin-B           | 95.95     | 97.75          | 82.81      |
> | MaxViT-T         | 96.11     | 97.99          | 83.33      |
>
> | (c) Upper-bound accuracy using $N = 324$ crops | ImageNet | ImageNet-ReaL | ImageNet-A |
> |-----------------|-----------|----------------|------------|
> | MobileNetV3-Small | 95.20     | 97.66          | 57.53      |
> | ResNet-50        | 96.77     | 98.63          | 66.15      |
> | EfficientNet-B0  | 96.87     | 98.60          | 69.20      |
> | ViT-B/32         | 97.19     | 98.75          | 77.77      |
> | ConvNext-Base    | 97.16     | 98.53          | 87.16      |
> | Swin-B           | 97.39     | 98.69          | 88.15      |
> | MaxViT-T         | 97.45     | 98.83          | 88.24      |

---

> ### Author Response · Authors · 2023-08-09
> **Response to Reviewer 3Lfp - Part 3/3**
>
> > Regarding the question of whether “image classifiers’ evolution over the past ten years is about mastering where and at what scale to zoom”, a prior work (Do Vision Transformers See Like Convolutional Neural Networks?) has touched on how modern classifiers solve classification differently from older architectures. The authors may want to discuss this work in “Discussion” as well.
>
> Thanks for suggesting us this interesting work!
>
> Both this paper and ours explore similar hypotheses about the advancement of representation learning within deep networks, presenting corroborative evidence. In particular, our focus on the implicit zooming mechanisms employed by deep classifiers on input images aligns with the investigation by Raghu et al. [R5] into how representation learning has transitioned from CNNs to ViTs.
>
> In our revised manuscript, we have included a discussion regarding this research, illuminating the connections and insights it provides in relation to our work.
>
>
> > The abstract proposes a hypothesis that “one way for image classifiers to reach high accuracy is to first zoom to the most discriminative region in the image” but my impression is that this hypothesis is still not sufficiently answered in the paper. The zooming experiments only provide an upper bound of what performance models can achieve given the right input scale, but don’t really tell us how much models are relying on better features versus better zoom in their classification. Do the authors have any thoughts on how to better disentangle these?
>
> Thank you for the great point!
>
> First, we acknowledge that understanding the decision-making process of an existing classifier is an active Explainable AI (XAI) research topic.
> So far, while there is no single, definitive method to use, Class Activation Map (CAM) and GradCAM are among the most popular methods to visualize which image areas a classifier looks at when make a certain prediction.
> We have added `Fig. 4` to the main text and `Sec. F4` in Appendix to show how a ResNet-50 (after finetuning using MEMO + RRC) has better accuracy and also **localizes objects better** (visualized via GradCAM). That is, MEMO+RRC ResNet-50 indeed changes to focus more on objects.
>
> Second, we have revised the hypothesis paragraph in the Introduction to clarify our point. The paragraph now reads as the following:
>
> _When processing an image, a model may implicitly zoom in or out (defined in `Sec. 3`) to the most discriminative image region ignoring the rest of the image (`Fig. 1a`), and then extracts that localized region's features to predict image labels.
> We hypothesize that the improved image classification may largely be due to the networks accurately zooming to the discriminative areas (e.g., junco and magpie birds in `Fig. 1`) rather than more accurately describing them (i.e., generating better features)._
>
> In this paper, we show supporting evidence for our hypothesis.
> For example, using the same old AlexNet feature extractor, one can reach a high accuracy (using 36 zoom crops) of almost `90%` on ImageNet.
> In other words, we are able to show that the zooming hypothesis is a **viable** mechanism to improve accuracy.
> Yet, future XAI research is needed to fully disentangle the two hypothesis (better _zooming_ to regions/objects vs. better _features_ of regions/objects).
>
> ## References
>
> ```
> [R1] Liu, Zhuang, et al. "A convnet for the 2020s." Proceedings of the IEEE/CVF conference on computer vision and pattern recognition. 2022.
>
> [R2] Liu, Ze, et al. "Swin transformer: Hierarchical vision transformer using shifted windows." Proceedings of the IEEE/CVF international conference on computer vision. 2021.
>
> [R3] Tu, Zhengzhong, et al. "Maxvit: Multi-axis vision transformer." European conference on computer vision. Cham: Springer Nature Switzerland, 2022.
>
> [R4] Gadre, Samir Yitzhak, et al. "DataComp: In search of the next generation of multimodal datasets." arXiv preprint arXiv:2304.14108 (2023).
>
> [R5] Raghu, Maithra, et al. "Do vision transformers see like convolutional neural networks?." Advances in Neural Information Processing Systems 34 (2021): 12116-12128.
> ```

---

### Official Review · Reviewer_j8Bw · 2023-07-23
**Poses an interesting+potentially insightful hypothesis, but is not entirely convincing**

**Rating:** 7
**Confidence:** 4
**Correctness:** Yes
**Clarity:** Yes

**Strengths:**

– Proposes and studies a simple, intuitive, and interesting hypothesis that could have important implications

– The experiments performed to validate the hypothesis are generally well-designed and easy to follow. The findings from the oracle experiment in Table 1, especially that even AlexNet can obtain high accuracies given the right framing, was surprising and insightful.

– Includes a comprehensive appendix with details and additional results which is a helpful resource

– The huggingface page for the dataset seems well-documented and easy to use

– The paper is well-written and easy to follow

**Additional Feedback:**

Well-written paper that studies an interesting problem, but I believe could benefit from a few additional clarifications / reframing of claim (see weaknesses above). I would be happy to reconsider my rating based on the author response to my concerns.

Suggestion:
The min-set cover algorithm used to find a minimal and optimal subset of transforms seemed interesting but is not defined in sufficient detail even in the appendix: I think it would be useful to describe it in more detail, and to discuss if it could potentially have broader applications in finding a minimal subset of transforms to apply for a given dataset and model?

**Documentation:**

Yes

**Ethics:**

No concerns

**Limitations:**

Yes

**Opportunities For Improvement:**

– My largest concern is around framing: while zooming is certainly an intuitive and succinct way to frame what models might be learning, it seems possibly oversimplified, for two reasons:

i) L23: “A model first implicitly zooms in or out .. to the most salient region in an image ignoring the rest of the image and then extracts that localized region’s features to predict image labels. That is, we hypothesize that the improved image classification may also be due to the networks accurately”-> In my opinion, the experiments conducted show that this is a possible reason for improved performance – since correct framing is shown to indeed improve performance – but does not provide sufficient evidence to make a causal claim (which is what it currently reads as): the other prevailing hypotheses – eg. that learning representational invariances via augmentation forces the model to learn commonalities corresponding to the category – also still seems equally plausible to me.

ii) It is also well known that models rely on context to make predictions (shortcut learning), which in some cases may be spurious but in other cases reasonable. Therefore, it is not clear to me that zooming is necessarily the right inductive bias for all applications. A discussion of this, perhaps with examples of cases where it may not be ideal, would strengthen the paper

– I have some concerns about the MEMO+RRC extension (Table 2):

i) The performance of MEMO+RRC against MEMO(+Augmix) is only slightly better, and it is unclear if the gains are statistically significant. Since Augmix already includes spatial transforms, is the main finding here that those are more important than the other transforms it includes?

ii) It does not seem obvious to me that the performance gains of MEMO+RRC are due to learning an implicit zooming operation, rather than simply useful learning representation invariance to diverse viewpoints / contexts / scale etc., as the paper appears to claim.

iii) Minor note: it would be useful to review MEMO in the paper as the current description is somewhat brief and vague, and I needed to refer to the original paper to understand it.

– That ImageNet has a center bias is well-known, and so it not entirely surprising that ImageNet-A, being its subset, also contains a center bias (though the experiment conducted in experiment 4.2 is compelling). Contextualizing these results against ImageNet would be helpful: for example, is the center bias in ImageNet-A significantly higher than for ImageNet as a whole?

– Minor: HardImageNet is already an existing ImageNet subset (https://openreview.net/forum?id=76w7bsdViZf) , so it might be helpful to use a different name for the proposed benchmark to avoid confusion, perhaps one that captures the focus on spatial bias?

**Relation To Prior Work:**

Yes

**Summary And Contributions:**

Studies the role of spatial biases in image classification by investigating and validating the hypothesis that models implicitly learn to zoom into the discriminative area of the image. Shows several off-the-shelf networks to perform substantially better on ImageNet given the right input framing, in addition to establishing the existence of a strong center bias in ImageNet-A and ObjectNet. Finally, an extension to an existing test-time adaptation method is proposed that makes use only of zooming operations.

---

> ### Author Response · Authors · 2023-08-09
> **Response to Reviewer j8Bw - Part 1/2**
>
> Thank you for your valuable, constructive feedback, which has made the latest revision stronger! Please find our inline answers below.
>
> We think our answers have addressed your concerns. If you have follow-up questions or requests, we'd be very happy to address them!
>
> > i) L23: “A model first implicitly zooms in or out .. to the most salient region in an image ignoring the rest of the image and then extracts that localized region’s features to predict image labels. That is, we hypothesize that the improved image classification may also be due to the networks accurately”-> In my opinion, the experiments conducted show that this is a possible reason for improved performance – since correct framing is shown to indeed improve performance – but does not provide sufficient evidence to make a causal claim (which is what it currently reads as): the other prevailing hypotheses – eg. that learning representational invariances via augmentation forces the model to learn commonalities corresponding to the category – also still seems equally plausible to me.
>
> Thank you for the great point!
>
> First, we acknowledge that understanding the decision-making process of an existing classifier is an active Explainable AI (XAI) research topic.
> So far, while there is no single, definitive method to use, Class Activation Map (CAM) and GradCAM are among the most popular methods to visualize which image areas a classifier uses to make a certain prediction.
> We have added `Fig. 4` to the main text and `Sec. F4` in Appendix to show how a ResNet-50 (after finetuning using MEMO + RRC) has better accuracy and also localizes objects better (via GradCAM).
>
> Second, we have revised the hypothesis paragraph in the Introduction to clarify our point. The paragraph now reads as the following:
>
> _When processing an image, a model may implicitly zoom in or out (defined in `Sec. 3`) to the most discriminative image region ignoring the rest of the image (`Fig. 1a`), and then extract that localized region's features to predict image labels.
> We hypothesize that the improved image classification may largely be due to the networks accurately zooming to the discriminative areas (e.g., junco and magpie birds in `Fig. 1`) rather than more accurately describing them (i.e., generating better features of these two birds)._
>
> In this paper, we show supporting evidence for our hypothesis.
> For example, using the same old AlexNet feature extractor, one can reach a high accuracy (using 36 zoom crops) of almost `90%` on ImageNet.
> In other words, we are able to show that the zooming hypothesis is a **viable** mechanism to improve accuracy.
> Yet, future XAI research is needed to fully disentangle the two hypotheses (better _zooming_ to regions/objects vs. better _features_ of regions/objects).
>
>
> > ii) It is also well known that models rely on context to make predictions (shortcut learning), which in some cases may be spurious but in other cases reasonable. Therefore, it is not clear to me that zooming is necessarily the right inductive bias for all applications. A discussion of this, perhaps with examples of cases where it may not be ideal, would strengthen the paper
>
> Thank you for the great point!
>
> In `Sec. C1` of the appendix, we conducted a study to explore the role of zooming for classification using only foreground (FGSet) and background (BGSet) elements on ImageNet. We created two sets: FGSet, where only the main object is visible, and BGSet, where only the background is visible. We then measured the performance of classifiers using a standard 1-crop method and also reported the maximum possible accuracy using zoom. Our results show that while the BGSet achieves lower accuracy, both FGSet and BGSet exhibit a substantial increase in accuracy with zoom (average of 25pp). This suggests that zooming plays a significant role in classifying both the context and the main object, emphasizing the importance of zooming.
>
>
> > i) The performance of MEMO+RRC against MEMO(+Augmix) is only slightly better, and it is unclear if the gains are statistically significant.
>
> Thank you for your question!
> Upon your request, we have repeated the experiments three times using different random seeds and the conclusion remains unchanged---MEMO + RRC outperforms the default MEMO (+ AugMix). So, yes, the gains are **statistically significant**.

---

> ### Author Response · Authors · 2023-08-09
> **Response to Reviewer j8Bw - Part 2/2**
>
> > Since Augmix already includes spatial transforms, is the main finding here that those are more important than the other transforms it includes?
>
> The default AugMix combines several transformations during test time. It begins by generating multiple augmentations of the initial image, then blends these augmentations together to produce the final output. We reduced this set to only one essential transform, which is `RandomResizedCrop`. Our results show that this reduction is very successful, and in almost all cases, it maintains or even surpasses the **accuracy** of AugMix by a few points across various datasets. More importantly, the result of this reduction leads to a `1.5x` to `2x` times improvement in **speed**, a factor that is crucial for algorithms in the test-time adaptation and test-time augmentation field.
>
>
> > ii) It does not seem obvious to me that the performance gains of MEMO+RRC are due to learning an implicit zooming operation, rather than simply useful learning representation invariance to diverse viewpoints / contexts / scale etc., as the paper appears to claim.
>
> Thank you for your feedback!
>
> First, MEMO only finetunes a network for one epoch; therefore, it tends to not drastically alter the representation learned by the network. In light of your feedback, we have generated Grad-CAM visualizations for the last convolutional layers of different ResNet-50 models fine-tuned using MEMO+RCC and compared them with the original baseline ResNet-50.
> We find that, after fine-tuning, the networks often **localize objects better**, which corresponds with their improved accuracy (see `Fig. 4`).
>
>
> > iii) Minor note: it would be useful to review MEMO in the paper as the current description is somewhat brief and vague, and I needed to refer to the original paper to understand it.
>
> Thank you for your feedback! We have revised the main text to include more details on MEMO.
>
> > That ImageNet has a center bias is well-known, and so it is not entirely surprising that ImageNet-A, being its subset, also contains a center bias (though the experiment conducted in experiment 4.2 is compelling). Contextualizing these results against ImageNet would be helpful: for example, is the center bias in ImageNet-A significantly higher than for ImageNet as a whole?
>
> Thank you for your question!
>
> First, we wish to clarify that ImageNet-A (along with ImageNet-R, ImageNet-Sketch, and ObjectNet) are NOT subsets of ImageNet, but rather **out-of-distribution** sets. These datasets are acquired separately many years after ImageNet and with objects to challenge the classifiers on rare images. In the case of ImageNet-A, this dataset has been constructed based on iNaturalist and Flickr using a process called adversarial filtering.
>
> Regarding contextualization with ImageNet, we have revised `Fig. 3` to include ImageNet, and additionally, we have added $\delta$ values between off-center crops and the center crop to illustrate the sharp contrast between ImageNet-A, ObjectNet, and other datasets.
>
> Our results from the experiment in `Sec. 4.2` suggest the presence of a strong center bias in ImageNet-A and ObjectNet. To demonstrate that bias is strong, we show that, during inference, simply upscaling every image (and then center cropping) in these two datasets, results in improved image classification accuracy for the same _pre-trained_ classifiers. The details can be found in `Fig. A14` and `Fig.  A17` in `Sec. B9`.
>
>
> > Minor: HardImageNet is already an existing ImageNet subset (https://openreview.net/forum?id=76w7bsdViZf) , so it might be helpful to use a different name for the proposed benchmark to avoid confusion, perhaps one that captures the focus on spatial bias?
>
> Thank you for bringing this to our attention! We are aware of this dataset, and we are considering renaming ours. If you have any naming suggestions, we'd really appreciate it.
>
>
> > Suggestion: The min-set cover algorithm used to find a minimal and optimal subset of transforms seemed interesting but is not defined in sufficient detail even in the appendix: I think it would be useful to describe it in more detail, and to discuss if it could potentially have broader applications in finding a minimal subset of transforms to apply for a given dataset and model?
>
> In light of your comments, we revised the text to provide more details and explanations of the minimum set algorithm. Both the main text and the appendix now include additional information about the necessary steps.

---

> > ### Comment · Reviewer_j8Bw · 2023-08-24
> > **Thanks for the response**
> >
> > The author response has addressed most of my concerns. In particular, I'm satisfied with the reframing (zooming in being a possible explanation, and softly validated using techniques like GradCAM), and the additional clarifications around center bias. The experiment included in C.1 is also compelling: it would be good to reference it in the main paper and contextualize its findings wrt the other claims made in the paper. I will raise my rating.

---

### Official Review · Reviewer_JuE9 · 2023-07-24
**imagenethard-review**

**Rating:** 7
**Confidence:** 3
**Correctness:** Looks like it.
**Clarity:** Yes

**Strengths:**

- The proposed test time augmentation is computationally faster than MEMO, another commonly used technique

**Additional Feedback:**

NA

**Documentation:**

Yes

**Limitations:**

See above.

**Opportunities For Improvement:**

1. The novelty of this work is not very clear to me - the issue around ImageNet suffering from center bias has been established in the literature for a couple of years now. The second issue around smaller objects not being easy to classify is also something that is established in the literature and is the reason behind the proposal of Feature pyramid networks: https://arxiv.org/abs/1612.03144


2. The description through the figure caption in Fig.1 is not quite clear. Why is the third row evaluated using a different model, VIT? Why isn't it made explicit that the same image is being broken down into 9 patches in (b)? Also, what is the original image resolution of the figure shown in (b)? How are you accounting for lower-resolution images where you are evaluating each patch? The patch would be of much lower visual resolution and scaling it to 224X224 might result in images of lower resolution.

3. Is there any pattern that is identifies in the top-36 zooms that seemed to be most helpful? Eg: Are they all centering the objects in the image?

4. How were the different zoom versions rated?

5.

**Relation To Prior Work:**

Moderately so. I did not find relevant references to the prior works which already proved about center bias issue wrt ImageNet.

**Summary And Contributions:**

- This work uncovers positional biases in various datasets, especially a strong center bias in two popular datasets: ImageNet-A and ObjectNet.

- A test time augmentation proposed in this work improves classification accuracy by forcing models to explicitly perform zoom-in operations before making predictions.

- A new dataset, ImageNet-Hard, a new benchmark that challenges SOTA classifiers including large vision-language models even when optimal zooming is allowed.

---

> ### Author Response · Authors · 2023-08-09
> **Response to Reviewer JuE9 - Part 1/2**
>
> Thank you for your valuable, constructive feedback! We have updated the paper in light of your comments.
> Please find our inline answers below.
>
> > The novelty of this work is not very clear to me - the issue around ImageNet suffering from center bias has been established in the literature for a couple of years now. The second issue around smaller objects not being easy to classify is also something that is established in the literature and is the reason behind the proposal of Feature pyramid networks: https://arxiv.org/abs/1612.03144
>
>
> Thank you for your question!
>
> The main objective of our study is to investigate the effect of zoom on image classification performance. Based on the study, we introduce the ImageNet-Hard dataset. One key outcome of our method is a new way to measure the positional bias of different image classification benchmarks, regardless of the availability of bounding box information. To our knowledge, our work is **the first** to:
> - Establish an upper bound accuracy of well-known classifiers on many OOD benchmarks when zooming is allowed. For example, we are the first to find out that even the same old AlexNet feature extractor can theoretically achieve over 90% accuracy on ImageNet (given the right zoom crop).
> - Find out the positional biases in *non*-ImageNet benchmarks (e.g. ImageNet-A and ObjectNet)
> - Create ImageNet-Hard, a benchmark that contains the hardest remaining images (from a set of 6 benchmarks). The latest vision-language models only perform poorly on ImageNet-Hard (1 to 2%).
>
> Moreover, the findings we produced are used to directly improve a state-of-the-art MEMO test-time augmentation method (MEMO + RRC).
>
> For further novelty comparison w.r.t. other papers, we kindly ask the reviewer to provide a specific reference, which we are very happy to contrast with our work. We believe we have adequately cited all relevant publications in the related work section, but if something has been missed or if there is any ambiguity, we are more than willing to revise that section and offer additional details.
>
>
> > The description through the figure caption in Fig.1 is not quite clear. Why is the third row evaluated using a different model, VIT? Why isn't it made explicit that the same image is being broken down into 9 patches in (b)? Also, what is the original image resolution of the figure shown in (b)? How are you accounting for lower-resolution images where you are evaluating each patch? The patch would be of much lower visual resolution and scaling it to `224x224` might result in images of lower resolution.
>
> We apologize for the confusion that may have arisen from combining two plots side by side in Figure 1 (we put these plots together to save space). In Subfigure 1(a), the effectiveness of zooming in and out is illustrated across three distinct scenarios using two different classifiers: ResNet-50 and ViT. This subfigure demonstrates that these networks, utilizing the same learned representations, can classify images from various datasets. The key lies in properly framing the object either to (top row) see it clearly, (middle row) capture it completely, or (bottom row) reduce the clutter. On the other hand, Subfigure 1(b) illustrates the overall process of extracting patches from the original image. This process involves defining sample anchor points on the image, scaling the image up and down, and slicing it using a `CenterCrop` operation centered at each anchor point.
>
>
> Regarding the cropping process, it is important to note that the scaling operation is performed before the `CenterCrop`, which subsequently extracts a `224x224` region from the scaled image. Most images in the ImageNet dataset (~84%) have sizes between  `333x333` and `500x500` pixels. For scaling images up and down, we used PyTorch's default scaling behavior, and at extreme zoom scales, the resulting image might appear blurry. Additionally, it is worth mentioning that we have incorporated both zooming in and out groups that aid in framing the object across various scenarios, such as low-resolution images.
>
> The original size of the `snail` image (Fig. 1b) is `375 × 500` pixels.
>
>
> > Is there any pattern in the top-36 zooms that seemed to be most helpful? Eg: Are they all centering the objects in the image?
>
> Thank you for your question!
>
> In our analysis of top-performing transforms, we discovered distinct patterns that highlight the unique characteristics of each dataset. Notably, a trend spanning all datasets and classifiers is the strong emphasis on the center, with approximately `26%` of all the top 36 transforms consistently belonging to the center crop at varying scales. Particularly, ObjectNet shows a stronger preference for centered patches, with an average of 35.65% of the patches belonging to the center across varying scales.
>
> Based on your question, we have included this result and relevant plots in a new section (`Sec. B10`, page 33) in the appendix.

---

> ### Author Response · Authors · 2023-08-09
> **Response to Reviewer JuE9 - Part 2/2**
>
> > How were the different zoom versions rated?
>
> Thank you for your question!
>
> According to our results, there is a dataset-dependent trend among datasets. ImageNet-A and ObjectNet favor higher zoom levels up to a certain threshold, while ImageNet-R and ImageNet-Sketch favor zooming out. To summarize:
>
> - For ImageNet-A, upscaling every image in the dataset to 448 and performing a center-crop can lead to a 5-15pp improvement in accuracy across various classifiers.
> - ObjectNet is exhibiting a similar trend, with consistent improvement of 2-8pp.
> - Various classifiers benefit from a slight zoom out on both ImageNet-R and ImageNet-Sketch, and this improvement can reach up to 2pp.
>
> Please refer to `Sec. B9` in the appendix for details and plots.
>
> > Moderately so. I did not find relevant references to the prior works which already proved about center bias issue wrt ImageNet.
>
> Thank you very much for your valuable feedback and comments!
>
> We believe we have cited all relevant publications related to image classification tasks in the "Related Work" section. However, if something has been missed or if there is any ambiguity, we are more than willing to revise that section and provide additional details.
>
> ## References
>
> ```
> [R1] Tan, Mingxing, and Quoc Le. "Efficientnetv2: Smaller models and faster training." International conference on machine learning.  PMLR, 2021.
> ```

---

> > ### Comment · Reviewer_JuE9 · 2023-08-29
> > **Follow up on authors comments**
> >
> > Thank you for answering my questions. After going through responses to my questions and that of other reviewers, I am happy to bump up my rating.

---

### Decision · Program_Chairs · 2023-09-22

**Decision:**

Accept (Poster)

**Comment:**

Reviewers sentiment is generally positive on this paper. I think this dataset and analysis can be useful to the community. It addresses an important fundamental area in computer vision which is scene and object recognition. I also support accepting this submission.